# Early evolutionary branching across spatial domains predisposes to clonal replacement under chemotherapy in neuroblastoma

Jenny Karlsson [1], Hiroaki Yasui[1,2], Adriana Mañas [3], Natalie Andersson [1], Karin Hansson [3], Kristina Aaltonen [3], Caroline Jansson[1], Geoffroy Durand[1], Naveen Ravi[1], Michele Ferro[1], Minjun Yang [1], Subhayan Chattopadhyay[1], Kajsa Paulsson [1], Diana Spierings [4], Floris Foijer [4], Anders Valind [1,5], Daniel Bexell [3] & David Gisselsson [1,6] ✉

Neuroblastoma (NB) is one of the most lethal childhood cancers due to its propensity to become treatment resistant. By spatial mapping of subclone geographies before and after chemotherapy across 89 tumor regions from 12 NBs, we find that densely packed territories of closely related subclones present at diagnosis are replaced under effective treatment by islands of distantly related survivor subclones, originating from a different most recent ancestor compared to lineages dominating before treatment. Conversely, in tumors that progressed under treatment, ancestors of subclones dominating later in disease are present already at diagnosis. Chemotherapy treated xenografts and cell culture models replicate these two contrasting scenarios and show branching evolution to be a constant feature of proliferating NB cells. Phylogenies based on whole genome sequencing of 505 individual NB cells indicate that a rich repertoire of parallel subclones emerges already with the first oncogenic mutations and lays the foundation for clonal replacement under treatment.

Neuroblastoma (NB) is the most common pediatric solid malignancy outside the central nervous system. Although the pathogenesis of NB has been thoroughly investigated, the five-year overall survival of high-risk NB remains around 50%[1]. The main cause of death in NB is treatment refractory disease[2]. Treatment resistance occasionally manifests as progressive metastatic disease already under first-line multimodal chemotherapy. More commonly, a treatment-resistant metastatic relapse emerges after an initial response to therapy and surgical removal of the primary tumor. Relapse is associated with a dismal prognostic outlook[3,4], with few if any standard therapeutic options that reliably promise success. Therefore, molecularly targeted therapy is

increasingly being considered and there is now a broad range of clinical trials, of which many are coupled to the detection of a specific genetic profile in the patient's tumor[5,6]. A major challenge when matching treatment to molecular profile is the fact that the NB genome is neither uniform nor stable within a patient[7]. Early studies comparing primary NBs to matched relapses, demonstrated significant differences in genomic profiles over time[8–10], while more recent studies have shown substantial genetic diversity also in primary tumors[11] with ongoing evolution during and following chemotherapy[12], and through the metastatic process[13]. However, little is known about exactly how NB cells surviving chemotherapy in the primary tumor are related to

[1]Division of Clinical Genetics, Department of Laboratory Medicine, Lund University, Lund, Sweden. [2]Department of Obstetrics and Gynecology, Nagoya University Graduate School of Medicine, Nagoya, Japan. [3]Division of Translational Cancer Research, Department of Laboratory Medicine, Lund University, Lund, Sweden. [4]European Research Institute for the Biology of Ageing (ERIBA), University of Groningen, University Medical Center Groningen, Groningen, The Netherlands. [5]Department of Pediatrics, Skåne University Hospital, Lund, Sweden. [6]Department of Pathology, Office of Medical Services, Region Skåne, Lund, Sweden. ✉e-mail: david.gisselsson_nord@med.lu.se

the lineages dominating prior to treatment. In the present study we perform high-resolution spatial mapping of subclones having survived chemotherapy in NB and investigate their phylogenetic relationships to clones detected at diagnosis. We find that subclones surviving effective chemotherapy are typically collaterally related ("cousins") to those present in the treatment-naïve biopsy, while those found after progression under failed treatment are usually linear descendents ("sons" and "daughters") of the subclones found prior to treatment.

## Results

### Spatiotemporal profiling under chemotherapy reveals two contrasting evolutionary patterns

NB is largely a copy-number aberration (CNA) driven disease[14–21]. We have reported a software tool (DEVOLUTION) that allows integration of point mutations and CNA data, where the latter is curated according to constraints of chromosomal evolution[22]. Please see specification in Methods for Data and Code availability. To perform high-resolution mapping of subclone territories across anatomic tumor space, we applied DEVOLUTION to mutational data from archived paraffin embedded formalin-fixed NB tissue (Fig. 1a–e; "Methods"). This allowed a comprehensive comparison between the genomes of NB cells before and after treatment, including both patients with a remaining, extensively viable tumor and cases with good chemotherapy response where survivor populations were limited to spatially dispersed tumor islets a few millimeters in size. Areas of ≅20 mm² containing >30% tumor cells were subjected to whole genome copy number profiling parallel to whole exome sequencing (WES). Mutations detected by WES were validated by targeted resequencing using a custom-made panel based on the WES findings. From a consecutive patient cohort treated at a single pediatric oncology center over a

period of 20 years, we identified 12 patients from whom viable tumor tissue was available from ≥2 geographically separated tumor regions from ≥2 time points (Fig. 1f, g; Supplementary Data 1a). A similar number of primary tumor samples (median 3.0) per patient from before treatment as from after chemotherapy were analyzed with both CNA profiling and sequencing; fewer samples were from metastatic sites (median 2.0).

Among the 12 patients, two exhbited very high-risk genetic profiles including *MDM2* amplification[23] in one and *NF1* deletion along with *MYCN* amplification[24], in the other. Both patients showed progression of the primary tumor with metastases under treatment (Patients 1–2). Nine other patients showed a significant therapy response (40–97% primary tumor regression including necrosis, fibrosis and extensive maturation; Patients 3–11), while one patient had a single small tumor in the neck and was treated with up-front surgery only and later developed a metastatic relapse (Patient 12). All seven patients whose tumor could be evaluated before and after significant chemotherapy response, showed a strikingly similar phylogenetic relationship between subpopulations detected before and after treatment (Patients 3–9; Fig. 2a and Supplementary Fig. 1, Supplementary Data 2a): some genetic changes found at diagnosis were no longer detected after treatment while other mutations emerged, as populations present prior to chemotherapy were replaced by others, typically arising from a different most recent common ancestor (MRCA). In the phylogenetic trees, this corresponded to one set of collateral branches being replaced by other, parallel branches and hence we denoted this pattern collateral clonal replacement (CCR). CCR occurred irrespective of whether the pre-treatment sample was obtained from the primary tumor (Patients 3–7) or from a metastasis sampled at presentation of disease (Patients 8–9). That CCR could take place gradually over time

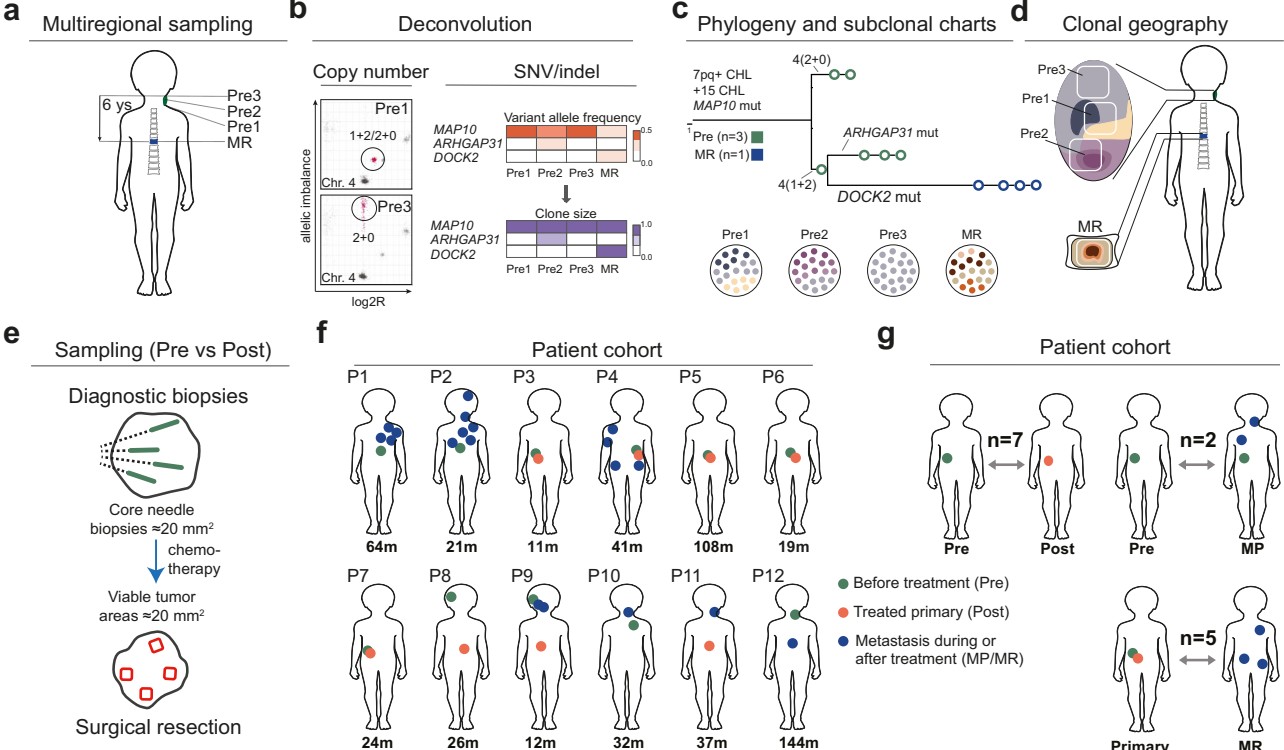

**Fig. 1 | Experimental approach and patient cohort.** Evolutionary analysis, illustrated by Patient 12 with multiregional sampling (**a**) of primary tumor (Pre1-Pre3) and a metastatic relapse (MR), followed by clonal deconvolution (**b**), construction of a maximum likelihood (ML) phylogeny and subclonal compositions (**c**); subclones are backtracked to samples to reconstruct their geographical positions (areas of different colors; **d**). Gains of 7pq and chromosome 15 (both including

chromothripsis like [CHL] events) and *MAP10* mutation are truncal events, a *DOCK2* mutation is confined to MR and a subclonal *ARHGAP31* mutation to Pre2.
**e** Sampling strategy before and after treatment. Patient cohort with sampled tumor sites and age at presentation (m, months; **f**) and the number (n) of spatiotemporal comparisons allowed by the cohort (**g**).

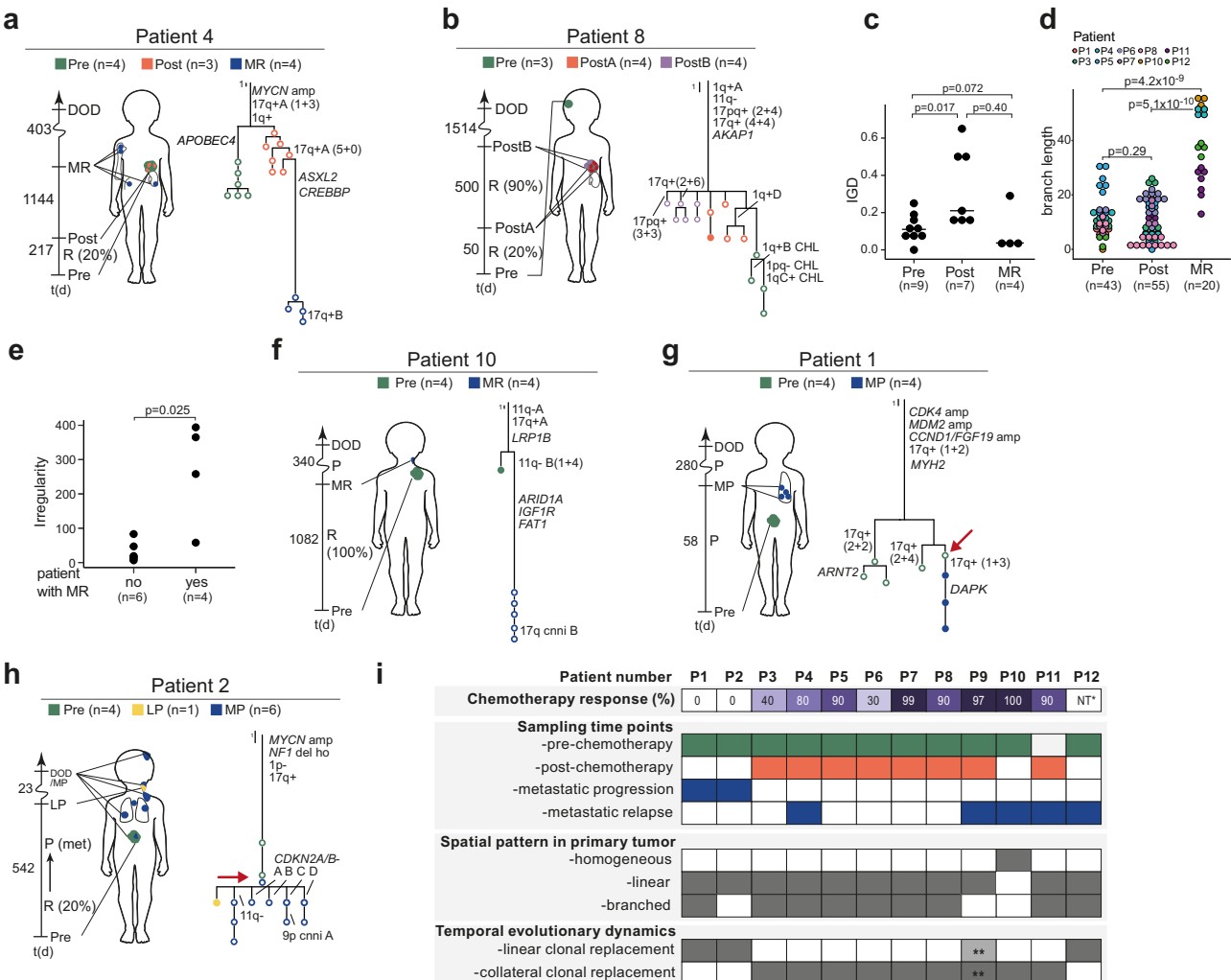

**Fig. 2 | Spatiotemporal evolution of somatic mutations under neuroblastoma treatment. a** ML phylogeny from Patient 4, illustrating collateral clonal replacement during chemotherapy response. Time line, days; R, regression; DOD, dead of disease. The tree is based on CNAs and SNV/indels with n indicating number of samples before treatment (Pre;green), after treatment (Post;red) and at metastatic relapse (MR;blue). Amplification (amp) of *MYCN*, 17q + , and 1q+ are truncal events (scale bar, one event). Capital letters indicates the same chromosome arm affected by >1 events with different breakpoints. Allelic compositions of 17q gain are denoted by numbers in parenthesis. Gene symbols signify non-synonymous SNVs/indels of likely importance to NB pathogenesis. **b** ML tree for Patient 8, based on samples from before (Pre;green), during (Post A;red) and after (Post B;purple) chemotherapy. Summary statistics of trees (*n* = 10) with CNA and SNV/indel data; index of genomic diversity (IGD) (**c**) with time points on x-axis (horizontal line, median value), total branch length (**d**) with patient color code above plot and time points on x-axis and irregularity (**e**) calculated on patients separated based on if a metastatic relapse sample (MR) was present ("yes" or "no" on the x-axis). *P* values were calculated using the Wilcoxon two sided test. Source data are provided as a Source Data file. ML phylogenies from patients with metastatic relapse (MR, **f**) or metastatic progression under treatment (MP, **g**, **h**), annotated as in (**a**) (P, local progression; LP, lymph node progression). In Patient 2 (**h**), parallel evolution of deletions of *CDKN2A/CDKN2B* with different breakpoints in 9p (**a**–**d**) are identified in metastases. Red arrows denote linear evolution at progression from ancestral subclones in primary tumors. **i** Overview across patients P1-P12 of chemotherapy response (%; for details see Supplementary Data 1a), sampling time points, spatial phylogenetic pattern in primary tumors, and temporal evolutionary dynamics. A filled square indicates the presence of a sample or an observation. NT* =Not treated with chemotherapy; up-front surgical removal. **Linear evolution for relapse at metastatic site after non-radical surgery, collateral clonal replacement for primary tumor compared to metastatic relapse.

was illustrated by Patient 8 (Fig. 2b). Here, the tumor was biopsied at diagnosis, after ~50 days of therapy (due to slow response) and then sampled again at resection after ~500 days under treatment. A comparison of these time points revealed a gradual shift towards CCR as populations were replaced. Using an index of genomic diversity (IGD) that was independent of both sample number and the total number of detected mutations (see Methods), we compared the subclonal heterogeneity among samples obtained before and after chemotherapy. In the primary tumor, diversity (IGD, Fig. 2c) was found to be higher after than before chemotherapy, even though the mutational burden (phylogenetic branch lengths) did not differ between these time points (Fig. 2d). CCR in the primary tumor under therapy was thus associated with the emergence of a more diverse clonal landscape.

CCR was also observed for the majority of subclones in the primary tumor when compared to those in a later metastatic relapse (*n* = 5; Patients 4, 9–12). Phylogenies containing clones detected at metastatic relapse exhibited a higher degree of irregularity (see "Methods") than those only based on clones detected in the primary tumor and/or up-front progression (Fig. 2e, f) as the lengths of branches to clones detected in relapse samples were overall longer than those in the primary tumor. In most relapsed cases, the lineage of relapsing subclones diverged from that of the primary tumor at an early stage, with CCR encompassing all subclones in the primary versus the relapse. Minor but significant exceptions from this were Patients 4 and 12. In Patient 4, a regional population corresponding the common ancestor of relapsing subclones was detected in the primary tumor

after therapy (Supplementary Fig. 1d:VII, subclone A), while in Patient 12 it was detected prior to therapy (Supplementary Fig. 1l:VII, subclone H). Notably, these two primary tumors were the ones with least treatment-induced regression; Patient 4 responded largely by tumor cell ganglionic maturation, while Patient 12 was treated with surgery only. This indicated that inferior killing of cancer cells was associated with an aspect of linear evolution from surviving ancestral clones when/if the patient progressed. Indeed, in the cases with local and metastatic progression under chemotherapy (Patients 1–2), a subclone ancestral to the genomes of metastatic populations was ascertained at diagnosis (Fig. 2g–i), supporting features of linear evolution under inferior therapy response.

### Linear evolution under progressive growth of chemotreated patient-derived xenografts

Because the number of patients with progressive disease under therapy was limited, we set up a model system to mimic progressive tumor growth under chemotherapy. We used a well-established NB murine model system of patient derived xenografts (PDXs)[25,26]. PDXs from a *MYCN*-amplified (MNA) NB (*n* = 6) were treated by cisplatin monotherapy, resulting either in stationary disease (*n* = 3) or progression (*n* = 3) under treatment (Fig. 3a and Supplementary Fig. 2a:I) as reported[27]. We also simulated progression after incomplete surgery by partial excision of non-chemotherapy exposed PDXs followed by regrowth to the same volume (*n* = 4). Control PDXs (*n* = 4), grown to the same volume under saline treatment, were used as a reference. We analyzed two biopsies from each tumor (A and B) and phylogenetic analysis revealed a pattern of linear evolution with the cell culture from which PDXs were established acting as founder (Fig. 3b, c; Supplementary Fig. 2a:II–III). Most PDXs progressing under chemotherapy or after surgery showed expansion of a clone with partial gain of 1p, all of 1q and an extra copy of 17q, leading up to clonal sweeps (Supplementary Fig. 2b–d). They also had a significantly higher CNA burden and more private mutations than the cisplatin-stationary and the untreated group, in a fashion that was to some degree time-dependent (Supplementary Fig. 2e); while untreated tumors had the same end-size as tumors progressing under treatment or after resection, they reached this size in shorter time and had fewer anomalies. Whole exome sequencing revealed clonal evolution in PDXs mimicking clinical tumors, including parallel evolution of RAS/MAPK pathway mutations in linear radiations from the mother culture (Fig. 3d; Supplementary Data 2c–e).

To evaluate evolutionary dynamics also at progression under multimodal high-dose chemotherapy, a different PDX model, with upfront resistance to multimodel chemotherapy was then analyzed (Fig. 3e; data obtained from Manas et al.[27]). Here, PDX tumors were subjected to CNA profiling either after free growth (*n* = 3) or after progression under treatment (*n* = 4) with an intense five-drug chemotherapy protocol mimicking the regimen used to treat high-risk NB patients, i.e. rapid COJEC containing cisplatin (C), vincristine (O), carboplatin (J), etoposide (E), and cyclophosphamide (C). Similar to the situation with progression under cisplatin monotherapy, we found that both untreated and treated PDX tumors exhibited linear radiation from founder populations, progressing through clonal sweeps in 1/3 and 3/4 treated and untreated PDXs, respectively (Fig. 3f–g and Supplementary Fig 2f–g). Thus, PDX models with both cisplatin- and COJEC-resistant NB cells independently confirmed the clinical scenario of a pattern of linear evolution from a common ancestor at upfront progression, radiating in different directions at different anatomic sites (different PDX mice).

### Subclone replacement at relapse after effective chemotherapy in patient-derived xenografts

To compare the results from patient cases with effective chemotherapy treatment to a murine model, PDX3 tumors were subjected to COJEC

(Fig. 4a)[27]. Mice were randomized into treatment groups when NB PDX tumors reached approximately 500 mm³. After regression under COJEC to <200 mm³, surgery was performed to remove all visible tumor. The COJEC-treated tumors (9 × 2 samples per tumor) were compared to identically sampled untreated tumors (saline injections; *n* = 9 × 2). Treated and untreated tumors had some common evolutionary features, including expansion in 17/18 tumors of a subclone signified by the same gains in chromosomes 1 and 17 as in the cisplatin-only monotherapy model detailed above (Fig. 4b, c). A comparison of genetic diversity in treated and untreated tumors showed significant differences between the cohorts (Fig. 4d, left and center panels; Supplementary Fig. 3a–b; Supplementary Data 2f, g). Despite treated tumors being on average 10 times smaller in volume than those not treated, all treated tumors (9/9) harbored subclones that were unique to one of the two sampled regions while this was rare (2/9) in untreated tumors (*P* = 0.0023; two-tailed Fisher's exact test). In line with this, the number of subclonal alterations differentiating the two sampled regions as well as the total number of region-unique subclonal alterations was higher in treated than in untreated tumors, while there was little difference in the total number of aberrations between the COJEC-treated and untreated cohort (Fig. 4d, right panel). The scenario was thus similar to the findings in the clinical cohort, where there was no overall increase in the total number of aberrations but still an increased genetic intratumor diversity after chemotherapy treatment.

Because PDX tumor sizes were feared to be too small for sampling prior to treatment without inducing populations bottlenecks, the finding of CCR under therapy from the clinical cohort could not be validated by sampling PDXs before and after therapy as in the patient cohort. However, COJEC-treated PDXs T1 and T5 relapsed locally, enabling comparison of clonal landscapes at two distinct time points, i.e. just after treatment versus after regrowth (Fig. 4e and Supplementary. 3c). The two PDX relapses showed a strikingly similar route of evolution, with additional 17q copies, and successive intragenic deletions in the *PTPRD* and *TCF4* tumor suppressors[27–29]. While not being linked to treatment resistance per se, deletions of these genes have recently been shown to be recurrent features of evolving neuroblastomas in PDX systems[27]. Both PDXs showed genetic aberrations that were unique to each sampling time point, i.e. a pattern of subclonal CCR (Fig. 4f, g). Similar to relapses in the clinical cohort, there was an overall increase in the total number of CNAs after treatment in the PDXs, with the most complex subclones being found at relapse. PDX modeling thus supported the association between effective chemotherapy response and CCR at relapse.

### Recapitulation of collateral clonal replacement under chemotherapy in vitro

To test whether the contrasting scenarios of linear evolution under inferior treatment/progression versus CCR under effective treatment could be reproduced also under tightly controlled in vitro conditions, we used the cisplatin-sensitive cell line IMR-32, derived from a high-risk MNA NB (Supplementary Fig. 4a). In vitro growth for 42 days (11 passages) without treatment resulted in expansion of the major subclone up to fixation and concomitant loss of a parallel subclone (Fig. 5a, Samples A1-A3; Supplementary Fig. 4b; Supplementary Data 2h). Low-dose (0.1 μM) cisplatin treatment initially suppressed cell growth but the cells were proliferating at the end of the experiment. This treatment led to a preserved clonal landscape (Fig. 5a, Samples B1-B3). In contrast, high-dose (1.0 μM) cisplatin exposure with a 94% reduction in population size, followed by regrowth for 42 days, resulted in CCR, with elimination of a major subclone and the expansion of clones with novel CNAs in 2/3 parallel experiments (Fig. 5a, Samples C1-C3). Reseeding the same number of IMR-32 cells (6% of original) as the viable population remaining after cisplatin treatment, followed by expansion to the same final population size as the high-dose cisplatin experiment did not change the clonal landscape (Fig. 5b and

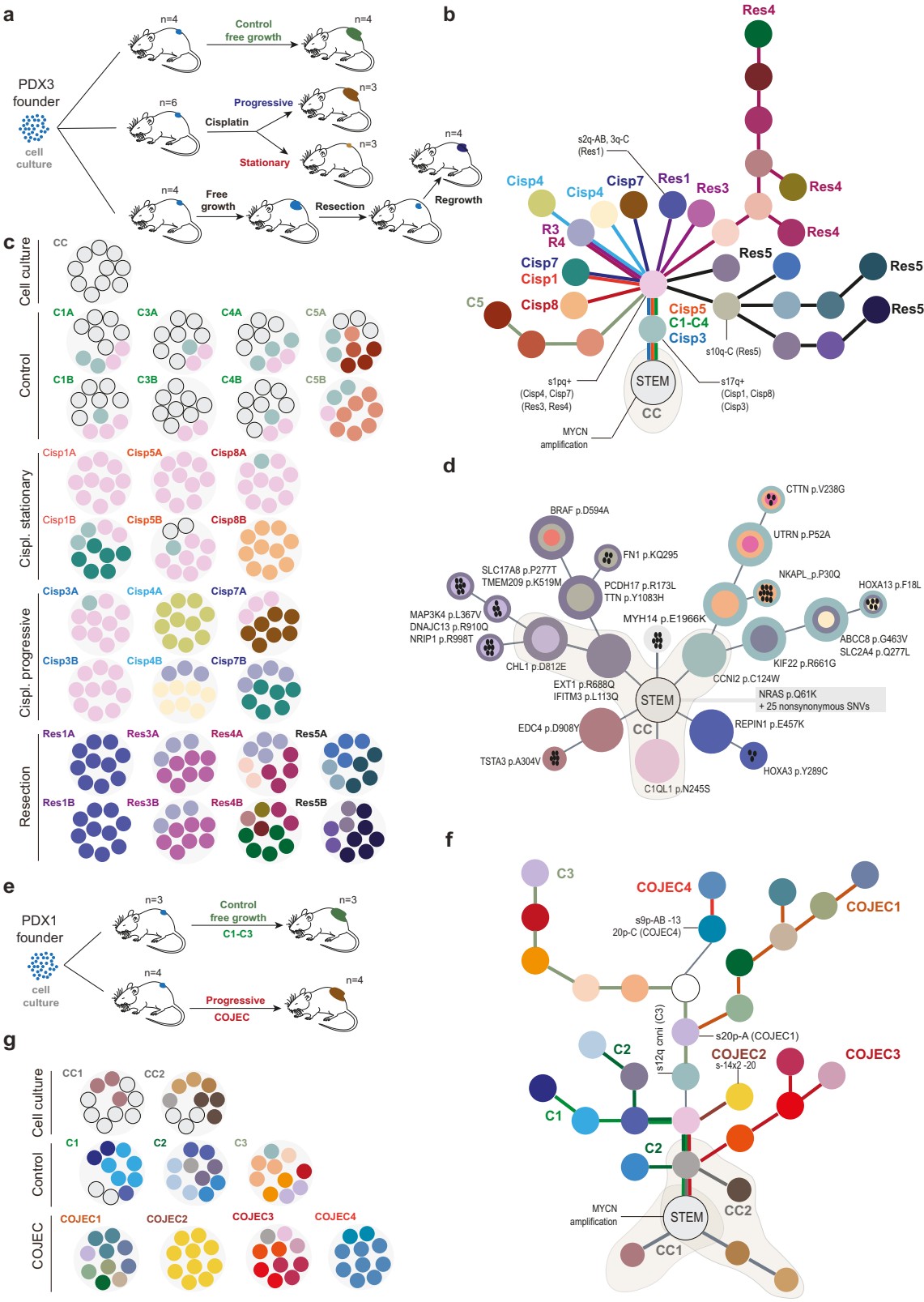

Supplementary Fig. 4c), nor was the landscape largely changed by a series of three mechanical reseeding bottlenecks of around 50% (Fig. 5c, Samples MB1-MB3; Supplementary Fig. 4d), while it again exhibited CCR after three consecutive cisplatin-bottlenecks each killing around 50% of cells. These results again corroborated that effective chemotherapy treatment is linked to a pattern of CCR and indicated that this effect is not merely contingent on the population bottlenecks to which effective treatment is associated (Supplementary Fig. 4e).

We then turned to the SK-N-SH line, well known for containing two morphological cell types, of which one is more resistant to chemotherapy than the other[30,31]. These respective cell types have recently been shown to harbor expression signatures corresponding to mesenchymal (resistant) and adrenergic (chemosensitive) cell states, respectively[31]. While the two cell populations have been reported to have largely similar copy number profiles[31], in-depth phylogenetic analysis of SK-N-SH under chemotherapy treatment had not been

**Fig. 3 | Subclonal evolution under progressive growth in chemotreated neuroblastoma PDXs. a** Genetically identical populations of *MYCN*-amplified NB PDX cells from a partly cisplatin sensitive tumor (PDX3) were injected in mice to form cohorts of untreated tumors (n = 4) and cisplatin-treated tumors (n = 6). Cisplatin treatment resulted in either stationary (n = 3) or progressive (n = 3) disease. A fourth group of PDXs (n = 4) was first allowed free growth, followed by partial resection and regrowth. Phylogenetic ideogram (**b**) and subclone composition (**c**) emerging in PDX3 (**a**) based on CNA profiles. From each PDX tumor, two biopsies were analyzed, large circles (**A** and **B**) in (**c**), in which each small circle corresponds to 10% of tumor cells. Clonal sweeps are annotated in (**b**) by "s" followed by the CNA denoting the sweep and in which PDXs it was observed; each CNA is annotated by chromosome number, chromosome arm (q) and structural variant (**A**–**C**). Evolution follows a pattern of linear radiation from the homogenous STEM population detected in the mother culture (CC, gray demarcation). **d** Phylogenetic ideogram based on WES of PDX3 tumors (same as in **a**–**c**), illustrating presence of RAS-MAP kinase pathway mutations (*BRAF* and *MAP34K*) in linear radiations from subclones in the culture (CC, gray demarcation). Mother clones are denoted as monochrome circles close to the stem, daughter clones as smaller rings inside the mother clones, and private mutations as black oval shapes (the number corresponds to the number of nonsynonymous mutations). Mutations predicted to be damaging by Polyphen are displayed. **e** Experimental setup in which PDXs from a *MYCN*-amplified treatment-resistant NB (PDX1), were allowed either free growth or progressed under high-dose multimodal chemotherapy (rapid COJEC protocol). Phylogenetic ideogram (**f**) and subclone composition (**g**) emerging in PDX1 (**e**) based on CNA profiles, annotated as in (**b**, **c**). A similar evolution as in the cisplatin resistant model (**a**–**d**) is observed, where subclones in emerging PDX tumors, irrespective of whether treated or not, are linear radiations from the main population present in the original cultures (CC1-2, gray demarcations). Details are provided in Supplementary Fig. 2b–g. See Mañas et al.[27] for raw data and experimental setup.

performed previously. We treated SK-N-SH cells with cisplatin for 72 h in vitro, followed by 3 days of regrowth, and genetic profiling (Supplementary Fig. 4f:I). Treated cells were reseeded, expanded, and again treated with cisplatin, followed by profiling for CNAs. Untreated control cells were grown in parallel. As expected, after the first treatment, the cells showed a shift to a more epithelioid morphology, corresponding to chemoresistant state transition and there was little evidence of cell death after the second treatment, compared to 80% cell death at first treatment. Already after the first treatment, there was a notable shift in the clonal landscape, with a clonal sweep of a cell population containing gain in 1q (1q+ ) replacing a cell population with only stem aberrations, present in mother cultures. Concomitant to this, there was an expansion of a nested subclone with imbalances in chromosomes 9 and 11 in treated cells (Supplementary Fig. 4f:II and g). In contrast, the population with only stem aberrations was retained in untreated controls. Furthermore, the two subpopulations dominating treated cultures were suppressed in control cells by expansion of a population with 2p+ , which in turn was suppressed close to the detection limit (10%) in treated cells. Hence, even in SK-N-SH cells, with a known capacity to undergo phenotypic state-transition independent of major genetic alterations[31], there were rapid shifts in the genetic landscape with features of CCR at chemotherapy treatment.

**Phylogenetic branches correspond to spatial territories.** The fact that tumors in the clinical cohort were derived from well-annotated paraffin blocks made it possible to trace back the approximate spatial location of each subclone in each tumor. In total 35/122 subclones spanned two or more biopsy locations allowing positioning of shared clones to neighboring areas, while subclones confined to a single biopsy location could not be mapped spatially beyond that location. Applying these principles to the 12 patients in the clinical cohort, we traced phylogenetic branches back to anatomic territories (Fig. 6 and Supplementary Fig. 5). In untreated diagnostic biopsies, these territories displayed a dense patchwork (median 0.36 subclones per mm², median distance between the center points of clonal territories of 1.7 mm), with no obvious anatomic barriers between subclone territories (Fig. 6a–d, i, k; Supplementary Fig. 5a, b, f, l–m). A similar situation with densely variegated clones were observed in biopsies from metastatic relapses (Supplementary Fig. 5f, i–k). Subclones in the primary tumor after chemotherapy were more dispersed (median 0.024 subclones per mm², median distance between the centerpoints of clonal territories of 6.5 mm; p = 0.0087 pre versus post; p = 0.048 post vs. relapse; Mann–Whitney U test, two-tailed; Supplementary Fig. 5l–m). Mutations detected after chemotherapy were typically fixed at a clonal level ( >90% of all tumor cells) in vastly dispersed islands of surviving tumor cells surrounded by necrosis, hemorrhage and other reactive changes (Fig. 6a lower, b right, c right, e–g, j, l). These distinct tumor cell populations typically originated from multiple phylogenetic branches, indicating a high degree of taxonomic diversity well in accordance to increased IGD after chemotherapy. Subclones identified at diagnosis were not re-detected after treatment in any of the viable primary tumor areas left after treatment (all areas sampled in Patients 3, 5–8). This indicated that the scenario of CCR under therapy was not only a result of geographically dispersed sampling but resulted from a true disappearance of subclones under treatment.

**Clonal replacement under treatment uncovers *MYCN* amplicon diversity.** CCR could potentially be explained by ancestral clones surviving therapy and sprouting new branches under treatment, parallel to those detected in the pre-therapy samples. Alternatively, NB phylogenies are highly branched already at an early stage, prior to treatment. To explore genetic diversity in early phases of disease we performed an in-depth analysis (down to 0.05 Mb) of copy number variation around the *MYCN* oncogene in 2p24 in MNA cases, as gain of *MYCN* gene is thought to be one of the earliest mutations in NB[32,33]. In the six MNA cases, an amplified state ( >5 copies) of *MYCN* was present across all biopsied areas. However, spatiotemporal heterogeneity in the architecture of amplicons in or around *MYCN* was also found in all of them (Supplementary Fig. 6). When chromosomal breakpoints of amplicon cassettes were used to infer phylogenetic relationships, all cases had at least one shared breakpoint across all samples, indicating that one or a few breakage(s) in 2p24 first arose to create patient-specific amplicon cassettes, followed by further modification later on. In total, only ~30% of samples (4/14) obtained from the primary tumor before treatment or from metastatic sites growing off-treatment after remission exhibited variant (non-stem) breakpoints around *MYCN*. In contrast, 76% of samples obtained from chemotherapy treated primary tumors exhibited variant breakpoints (13/17; p = 0.00122; Fisher's exact test). This was in accordance with the scenario of increased genomic diversity after treatment found across the entire genome.

**Phylogenetic branching is an early and stable feature of NB.** To further monitor early branching evolution with a focus on the *MYCN* amplicon, we performed low-pass single cell whole genome sequencing (scWGS) of single biopsies from nine NB primary tumors (three MNA tumors), resulting in 505 single cell genomes (Patients S1-S9 in Supplementary Data 1b). A unifying feature among all tumors irrespective of risk group was extensive phylogenetic branching (Supplementary Fig. 7). Branches typically led up to subsets of cells having identical CNA profiles, with the exception of one case (Patient S7), which had a different CNA profile in every cell (Supplementary Fig. 7d). In MNA patients S1 and S2, the earliest branching event consisted of structural variation in the *MYCN* amplicon cassette (Fig. 7a–e and Supplementary Fig. 7a). Both these cases contained cells having low-level *MYCN* gain as the first event in their phylogenetic trees (in Patient S2 co-gained with *MAML3*), corresponding to an ancestral population. Clonal evolution from this stage occurred through structural aberrations characteristic of NB, including deletion of 1p, gain of 17q (both

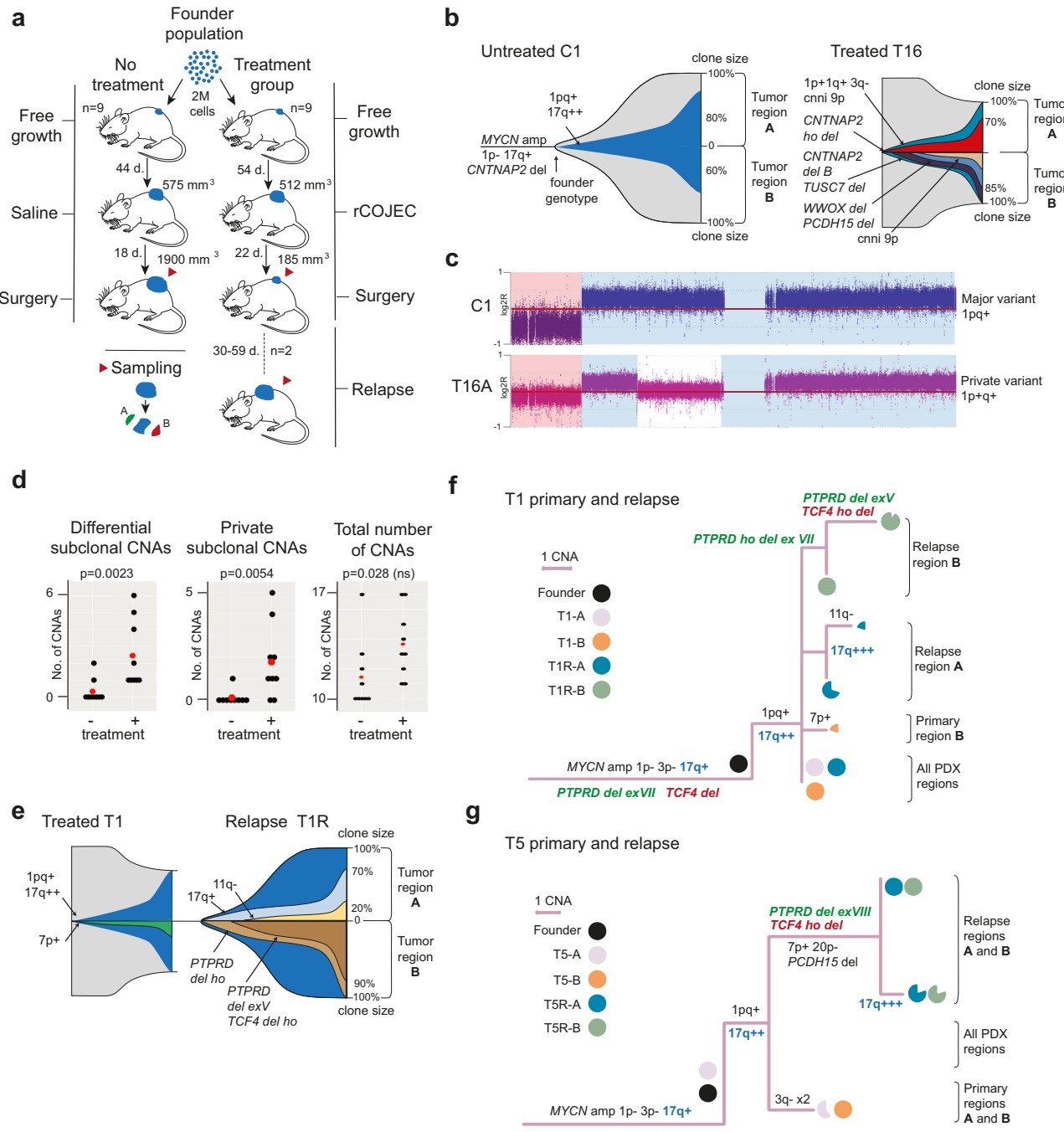

**Fig. 4 | Clonal evolution under effective chemotherapy in vivo. a** Experimental setup of neuroblastoma PDX tumors, red arrowheads indicate the time point for sampling and time is given as mean number of days (d) for each cohort consisting of n animals. **b** Fish plots of clonal evolution in a representative untreated (C1) and treated (T16) PDX tumor, with the upper and lower halves (A and B) representing two regions of the same tumor. Subclones signified by specific sets of CNAs are denoted by color fields, while aberrations in the mother culture are grouped around the stem of the C1 plot. Deletions affecting one gene are denoted by the gene name followed by *del*, while larger CNAs are denoted by chromosome or chromosome arm (p, q) followed by specification of gain (+), loss (−) or copy number neutral allelic imbalance (cnni). **c** SNP array plot of the segmental gain (blue highlight) in chromosome 1 that signifies the most commonly expanding clone in both PDX cohorts (1pq + 17q + +), exemplified by its appearance in untreated C1. In T16, this clone did not expand and there was an alternative

structural rearrangement of chromosome 1. **d** The number of differential subclonal, region-private subclonal CNAs, and the total number of CNAs per tumor in rapid COJEC untreated (−) or treated (+) PDXs. *P* values by Mann–Whitney U test (two-sided) with significance limit adjusted to 0.0055 after Bonferroni correction; ns, not significant. Source data are provided as a Source Data file. **e** Shift in clonal landscapes between primary treated and relapsed T1, showing how the subclone denoted 7p+ is replaced by a set of other clones detailed in (**f**). A and B denote sampled tumor regions. **f**–**g** Maximum likelihood trees showing CCR between primary treated PDX tumors (T1 and T5) and their relapses (T1R and T5R). CNAs are annotated as in (**b**), with chromosome segments targeted by further rearrangement marked by colored letters. Roman numerals denote exons in the *PTPRD* tumor suppressor targeted by sequential deletions. Pies denote the proportion of cells harboring a certain CNA. See Supplementary Fig. 3 for subclonal landscapes of other PDXs.

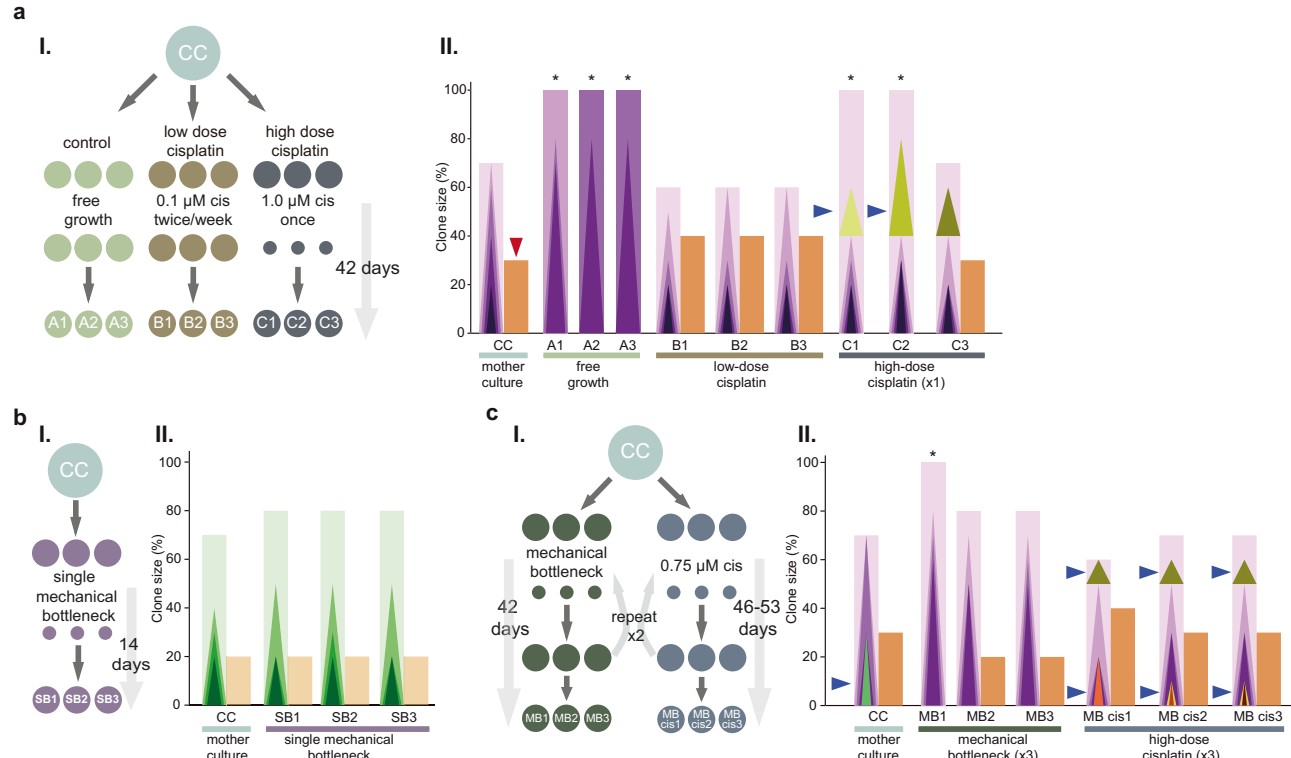

**Fig. 5 | Clonal evolution under chemotherapy treatment in vitro. a–c (I)** Schematic presentation of experimental set ups using cultured IMR-32 cells. All experiments were performed in technical triplicates. CC corresponds to the mother cell culture. A treatment resulting in a substantial decrease in cell numbers is illustrated with smaller circles. The time period in days from the start of the treatments to the collection of the cells is indicated by the arrow to the right in the figure. **(II)** The clonal landscapes were identified with CNA analyses and illustrated as bar charts. The y-axes correspond to clone sizes. Rectangles denote ancestral clones and triangles denote daughter clones. An asterisk (*) indicates the presence of a clonal sweep. Cultures having transitioned through CCR is indicated by blue arrowheads, with the lost parallel clone denoted by a red arrowhead in CC. **a.** IMR-32 cells under free growth (cell cultures A1-A3), continuous low dose cisplatin exposure (B1–B3), or a single high dose of cisplatin (C1–C3). **b** Cells subjected to a single mechanical bottleneck (SB1-3), consisting of restricting cell numbers at passage to a population with the same size as the number of viable cells remaining after the high dose cisplatin treatment. **c** Cell cultures subjected to multiple mechanical bottlenecks (MB1-3) and multiple high-dose cisplatin treatments (MB cis1-3). Details on clonal landscapes are provided in Supplementary Fig. 4a–d with a summary in Supplementary Fig. 4e.

Patients S1 and S2) as well as deletion in 11q (Patient S2 only; Supplementary Fig. 7a:III). To corroborate that MNA could precede 17q gain (the most common CNA in NB), we further evaluated the concurrence of these imbalances by fluorescence in situ hybridization in a subset of MNA NBs (Supplementary Data 1c). The presence of tumor cells with *MYCN* copy number gain while still not having acquired extra copies of 17q was ascertained in 5/6 cases (Fig. 7b and Supplementary Fig. 7f), the exception being S3 in which neither scWGS nor FISH could identify such cells (Supplementary Fig. 7c:I).

Interestingly, in both Patients S1 and S2, clonal evolution through 1p loss/17q gain appeared to be a requirement for further *MYCN* amplification, with significantly higher amplicon numbers in cells having acquired 1p loss/17q gain than in cells with only MNA (Fig. 7f, g and Supplementary Fig. 7a:V). Further branching into distinct subpopulations continued after 1p loss/17q gain in all three MNA cases. To test whether the tendency of branching evolution was a stable feature of NB cells, we turned to three PDX models from MNA NBs[25,26]. We used scWGS to monitor evolution from PDX in vivo generations 1–7, and further through transfer into free-floating three-dimensional tumor organoids analyzed at in vitro passages 7 and 20[27]. This showed that phylogenetic branching was a consistent feature across time in vivo, a feature that re-emerged after single-cell bottlenecks occurring at transition to in vitro conditions (Fig. 7h–j).

**Timing and fitness of chromosomal changes impact NB genome profiles.** The types of CNA associated with branching evolution in

the scWGS data varied across tumors, but a common feature (9/9 cases) was parallel aberrations affecting the same chromosome in different branches of the same phylogeny. For example, in Patient S4, the first branches emerged through distinct parallel deletions in 3p (Supplementary Fig. 7b:I–III), while in Patient S8 chromosome 6 was the substrate of the three earliest branching events, followed by parallel alterations of chromosome 17 and subsequently of chromosome 1 (Supplementary Fig. 7b:IV–V). Similar parallel evolution was detected in Patients S1 (*MYCN* amplicons; Fig. 7d), S2 (chromosome arms 1p, 9q, 17q, 20q; Supplementary Fig. 7a:III–IV), S3 (chromosomes 1, 2, 4, 6; Supplementary Fig. 7c:I–II), S5 (chromosome 5; Supplementary Fig. 7c:III–IV), S6 (chromosome arm 17; Supplementary Fig. 7e:I–II), S7 (chromosome arms 1q and 17q; Supplementary Fig. 7d), and S9 (chromosomes 1 and X; Supplementary Fig. 7e:III–IV). Also, in "numerical only" low-intermediate risk NBs (n = 4), structural rearrangements contributed to parallel rearrangements of the same chromosome. However, there was a difference between clinical-genetic NB subtypes regarding the time point in evolutionary history when different types of CNA occurred (Supplementary Fig. 7g:I). In "numerical only" low-intermediate risk NBs (n = 4), early generation branches (immediately following the stem) contained only whole chromosome aberrations, while chromosome breaks were enriched in subsequent generations. In contrast, in high-risk cases (with and without MNA; n = 5) chromosome breaks were frequent already at the earliest generations and remained so through successive generations.

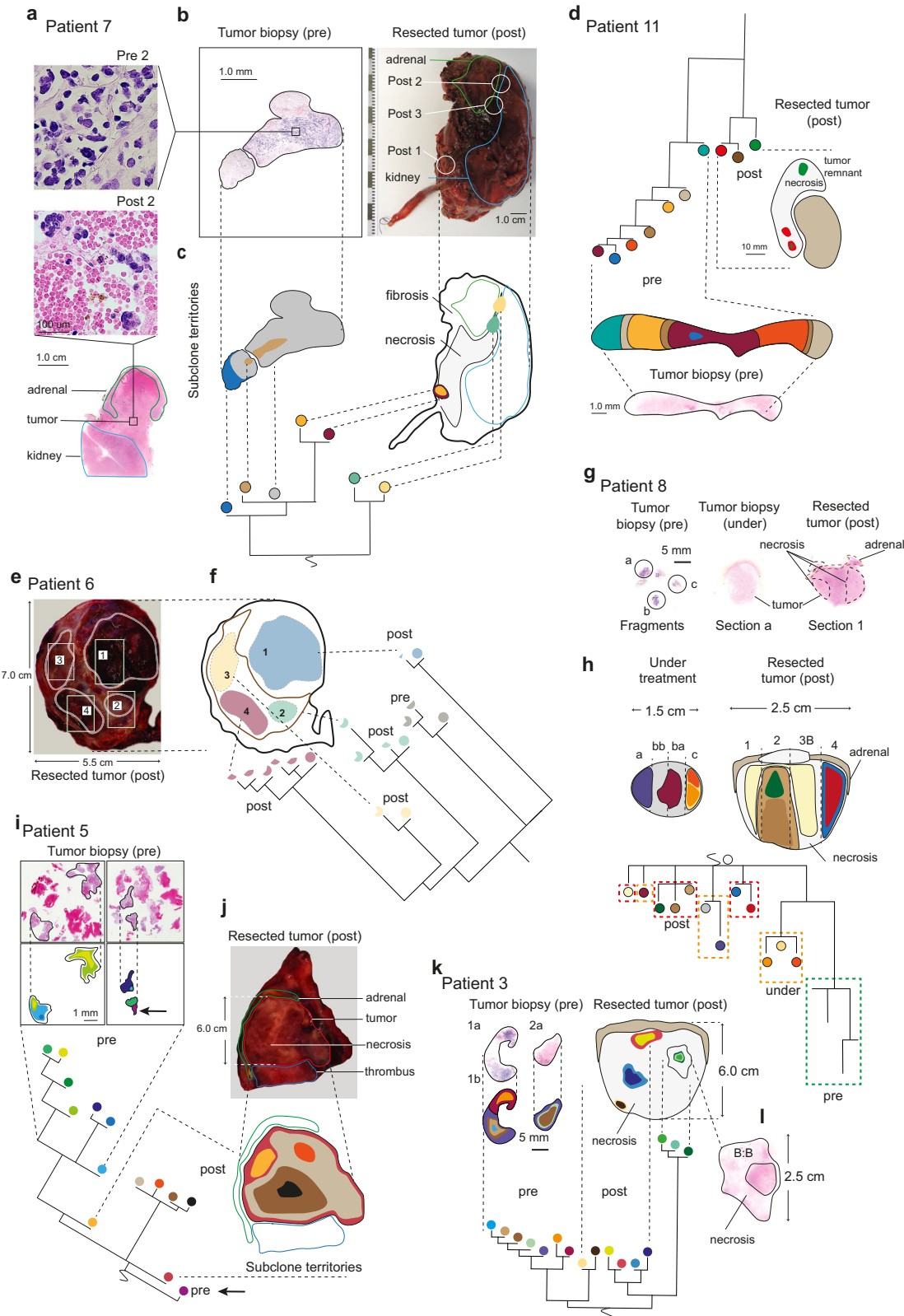

There was also a difference across clinical-genetic NB subtypes in the fitness cost associated with different chromosomal imbalances. As a proxy to the survivability following different types CNA, we quantified the proportion of progeny (PoP; average fraction of detected single cells) originating from branches with certain CNA profiles compared to branches with other profiles (Supplementary Fig. 7g:II–III). In low/intermediate risk NB, the PoP from lineages having undergone whole chromosome gains during preceding generations was significantly larger than the PoP from lineages with chromosome breaks or losses. In contrast, high-risk NBs showed a similar PoP resulting from branches dominated by whole chromosome breaks as by gains, while the proportion of progeny was lower for cells having undergone losses. To finally evaluate the timing of phylogenetic branching in NB compared to other tumors, phylogenies based on

**Fig. 6 | Subclone territories.** Images of histopathological specimens are combined with phylogenies to map subclone territories before (pre) and after (post) treatment. **a–c** Microscopy images, gross specimen, subclone phylogeny, and territories from Patient 7. **d** Phylogeny, ideogram of gross specimen and microscopic image of core needle biopsy from Patient 11. **e, f** Gross specimen with ideogram and phylogeny from Patient 6. **g, h** Microscopic images, territories and phylogeny from Patient 8. **i–j** and **k–l** corresponding images and ideograms from patients 5 and 3, respectively. The positions of subclone territories in relationship to phylogenetic trees and anatomic maps are linked by broken lines. Differently colored regions represent territories of specific subclones, with sizes (areas) inferred from frequency distributions of clones across biopsies from the same patients. Regions with black borders represent taxa whose precise anatomic locations are ascertained by multiregional sampling while regions without marked borders correspond to taxa whose locations are inferred from a specific cancer cell fraction in a single sample. In the phylogeny of Patient 8 (panel h, lower) clones detected before, under and after chemotherapy are marked by broken line boxes (green, orange and red, respectively). Tumor nodules surrounded by fibrous capsules are enhanced by gray lines in Patient 6 (**e**). While taxa are tightly variegated without anatomic borders before treatment (**c** left, **d** lower, **i**, **k**), they are distinctly segregated by necrosis (**c** right, **d** right, **h**, **l**) or encapsulating fibrosis (**e**, **f**) after treatment. Details of metastatic cases are presented in Supplementary Fig. 5.

scWGS for a reported reference group ($n = 4$)[11] of pediatric tumors less aggressive than NB were reconstructed. NB was found to exhibit extensive branching even prior to the outgrowth of the major sub-clone, while this was not the case for the more indolent tumors (Supplementary Fig. 7g:IV–V). In summary, scWGS confirmed that extensive phylogenetic branching is a consistent feature of NB over time that can emerge already at ancestral stages with low-level *MYCN* gain. High-risk NBs seem to have developed tolerance to chromosome breaks already at the time of first branching, resulting in complex patterns of sub-clone diversity that can form the substrate of CCR, through which new clones emerge to replace those killed off by chemotherapy.

## Discussion

The present study shows that genetic intratumor diversity in NB emerges by early phylogenetic branching, resulting in distinct sub-clone territories. This branched architecture sets the ground for clonal evolution under therapy, a process which we show is extensively determined by whether the subclones dominating the primary tumor are eradicated or not. If such dominant subclones survive che-motherapy, they seed direct descendants through linear evolution as disease progresses. If eradicated, they can leave room for replacement by a genetically diverse panorama of collateral relatives through CCR. In the present study, CCR was found with effective treatment irre-spective of whether patients had high-risk disease treated with very intensive chemotherapy (Patients 4–5, 7–11) or had low/intermediate risk disease treated with less intensity (Patients 3 and 6). CCR was also recapitulated when comparing COJEC-responsive PDX tumors to their local relapses, even though technical restrictions did not allow for an experimental setup perfectly mirroring the clinical patient scenario with pre-post sampling. Altogether, CCR is consistent with selection under chemotherapy of pre-existing NB cell populations with potential for treatment resistance, as has previously been described in a range of other malignancies[34–43]. The absence of CCR in cell culture by merely imposing a mechanical population bottleneck also supports Darwinian selection under chemotherapy as one underlying mechanism. On the other hand, linear evolution was observed at progressive growth irrespective of whether chemotherapy was given or not (patient 12, PDX3 C5, and PDX1 C1-C3 showed extensive linear radiation from detectable ancestral clones but were not exposed to chemotherapy).

Previous paired comparisons of tumor genomes from primary and relapsed NBs have identified a broad repertoire of driver muta-tions enriched at relapse, in particular RAS-MAPK pathway mutations[8–10,33], supporting some degree of selection under effective chemotherapy. However, the diversity among tumor genomes from different relapse patients and the notable absence of bona fide resis-tance mutations in NB in the present and previous studies, indicate that heritable factors other than DNA variants could be the main substrate of selection. Recent studies of chemotherapy resistance in NB have indeed indicated the presence of a spectrum of lineage development states based on super-enhancer-associated transcription factor networks[44–47], where cells with a mesenchymal transcriptional signature are enriched in post-chemotherapy and relapsed tumors compared to cells with an adrenergic signature. In fact, using the same PDX systems as in the present paper we have shown an increased expression of early developmental signatures, with features of Schwann cell precursors in treatment resistant and relapsed lineages, and also an increased mesenchymal signature in relapsed tumors versus an increased adrenal signature in PDXs cured by chemotherapy and surgery[27]. In this context, our data suggest that different genetic lineages present in the same patient have a diverse potential for attaining the mesenchymal, chemoresistant state, a hypothesis also supported by the in vitro data on SK-N-SH cells presented here, where state transition to chemotherapy resistance was accompanied by clear shifts in the clonal landscape. While a certain lineage dependence of such state-transition could be explained by epigenetic heritability, our finding of distinct subclonal territories in clinical samples suggest that microenvironmental factors may also contribute to priming NB cells towards attaining chemoresistance. Another alternative is that linea-ges dominating the primary tumor undergo negative selection at treatment because they are simply slower, albeit in the end not less capable, than minor collateral populations to reprogram into the mesenchymal state. The present study is not capable of distinguishing these alternatives for several resons. First, it is based largely on clinical paraffin-embedded samples, small in size, providing very limited opportunities for more comprehensive, phenotypic chatacterization. Second, our study is underpowered and undersampled (only 12 patients) and should thus be regarded as largely explorative. Future studies should ideally include larger patient cohorts with a broader access to fresh tumor material for phenotypic studies. An additional improvement for future studies would be a script-based, simulation-validated process that replaces the manual assignment of allele com-positions and clone size in the complex sublone scenarios often pre-sent in clinical NB samples.

The present study provides important clues on how to assess neuroblastoma patients for the purpose of precision oncology. Our results show that, at up-front progression new mutations are typically just added to pre-existing major clones, while regression followed by relapse implies a more radical shift in the clonal landscape through CCR, mandating resampling for additional sequencing. Furthermore, our tracing of lineages back to anatomic territories reveals the critical importance of taking spatial heterogeneity into account at precision oncology assessment. Subclones were densely packed in diagnostic biopsies and relapse samples, in a fashion where just a few mm$^3$ of nearby extra tissue in a biopsy would be rich in additional genetic information. In contrast, when evaluating the primary tumor after treatment, phylogenetically distant populations were found dispersed across small islands of surviving tumor cells. Thus, after therapy, care must be taken to sample several spatially distant tumor regions to capture the full repertoire of clones. This may prove practically chal-lenging because surviving tumor cell populations are typically nested in vast areas of necrosis and reactive host tissue. A future alternative for monitoring clonal evolution in NB could be liquid biopsy protocols. Several studies have shown great promise for using cell-free DNA to monitor the temporal dynamics of NB genomes, particularly in high-risk patients[48,49]. However, it remains an open question whether these methods will have sufficient sensitivity for regular clinical use. In summary, the present study shows that evolutionary branching early in NB pathogenesis sets the stage for clonal replacement under effective

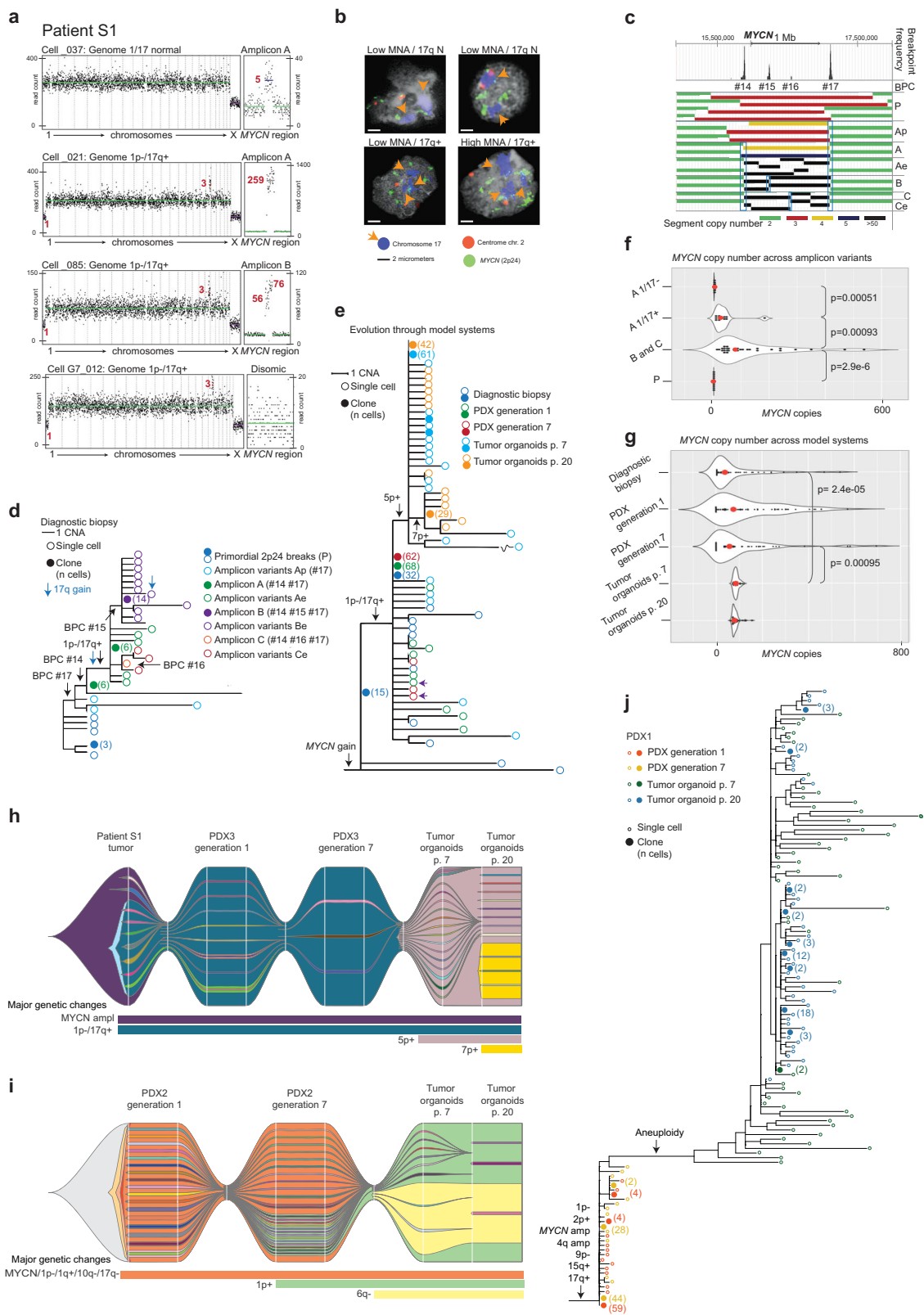

therapy, which has direct implications for how to sample these tumors for genomic profiling.

## Methods

### Ethics statement

The present study was in compliance with all relevant etica regulations: it was approved by the Regionala Etikprövningsnämnden (EPN) under permit numbers L289-11 genomic analyses; updated as L796-2017 and L605-05 (biobanking; updated as L883-2018). For data sharing, this was complemented by permit from the Swedish Ethical Review Authority (2023-01550-01). Written informed consent was obtained for all study subjects included from the children's parents or from legal guardians. For patients ≥15 years of age, direct consent was also obtained. Information about the study was given by a trained

**Fig. 7 | Evolution from ancestral *MYCN* copy number gain. a** Single cell whole genome sequencing (scWGS) copy number profiles of *MYCN*-amplified (MNA) neuroblastoma S1 and corresponding patient derived xenograft (PDX) generation 7 (G7). Left, low-resolution whole genome plot (1 Mb binning); right, high-resolution plot (40 kb binning) of the *MYCN* region. Red type, deviant copy numbers. Single cells _037 (ancestral stage), _21 and _085 show evolution from *MYCN* gain in absence of copy number alterations (CNAs), to cells with 1p deletion, 17q gain, and MNA (amplicon cassettes A and B; see panel c). Cell G7_012, loss of the amplicon in a PDX subpopulation. **b** Fluorescence in situ hybridization analysis (S1 biopsy) confirming cells with low-level MNA and normal 17q status (17q N) along with cells showing MNA and 17q gain (17q + ; 51-154 cells scored /tumor). **c** *MYCN* region breakpoint clusters (BPC; S1) defines: primordial cell (P) with heterogeneous breakpoints, population Ap with BPC#17 as common breakpoint, amplicon cassette A with BPCs #14 and #17, Ae with evolved variants of A in single cells, and clonal variants B and C defined by BPCs #15 and #16. **d** scWGS phylogeny of S1 biopsy based on BPCs and CNAs. Early branching (blue circles) results from *MYCN* diversity, followed by clonal expansion of amplicon A (6 cells; filled green circle), then 1p deletion and 17q gain (1p-/17q + ). **e** scWGS phylogeny of S1 biopsy, PDXs, and tumor organoids. **f, g** *MYCN* copy numbers by scWGS in the S1 biopsy, subdivided by clonal ancestry in biopsy (**f**, compare **d**), and origin from biopsy, and corresponding PDXs and tumor organoids (**g**). Red circles represent median; significance testing by Mann–Whitney U test (two-sided) using Benjamini-Hochberg correction. **h** scWGS CNA-based clonal dynamics (fish plot) over time in Patient S1 biopsy, and corresponding xenograft (PDX3) and tumor organoids. **i** scWGS CNA-based clonal dynamics in PDXs and tumor organoids from a different MNA patient (PDX2). **j** scWGS phylogeny of PDX cells and tumor organoids from a third patient (PDX1). Note aneuploidization at transition from PDX to organoids. Source data are provided as a Source Data file (**f, g**).

pediatrician and/or pediatric oncologist. There was no compensation to study participants. All animal procedures were conducted according to the guidelines from the Regional Ethics Committee for Animal Research (permit no. M11-15, 19012-19).

## Patient cohort

To study alterations in neuroblastoma genotype over the time of disease course, we reviewed pathology files of all patients <18 years of age diagnosed with and treated for a histopathologically verified neuroblastoma in the Southern Healthcare Region of Sweden from 1998-2018. From this cohort of approximately 50 patients, we selected those from whom there were samples with >50% viable tumor cells available for DNA-analysis, material from at least two different disease time points, and samples from at least two intratumoral locations from at least one of these time points. This resulted in totally 12 patients with material available from two or more of the following phases:

(1) At the diagnostic procedure before treatment was started (pre-chemotherapy). For most patients, core needle biopsies intersecting different tumor regions were obtained. Each biopsy core with viable tumor cells was analyzed as one or more separate samples; DNA from biopsies were not mixed.

(2) At the surgical resection of tumor after treatment (post-chemotherapy). Here, islands of surviving neuroblastoma cells were identified by light microscopy and isolated by microdissection of the corresponding paraffin blocks. Each island, or group of islands within the same 5 mm radius, was analyzed as a separate sample. One patient (Patient 8) was also sampled (by open biopsy) under ongoing chemotherapy.

(3) At the time of metastatic relapse. This material consisted of either core needle biopsies or open biopsy fragments, where each core needle biopsy/each fragment was analyzed as a separate sample.

(4) In the context of treatment resistant progression. This material consisted of open surgical biopsies at metastatic sites (Patient 1) or sampling of primary tumor and metastases at autopsy (Patient 2).

No specific analyses of sex and gender were performed as this was not within the scope of the study. Patients were enrolled based on the diagnosis and the availability of biological samples. The cohort contains children of both sexes as expected because the disease under study affects both sexes almost equally. There is a slight increased risk for boys and there is also a slight male predominance in our small study cohort. Sex was assigned and not self-reported as most of the study participants were young children. The study cohort is too small for meaningful subdivision according to sex and specifying it at publication may threaten the privacy of participants.

## In vivo chemotherapy experiments

The animal studies included correspond to previously published experiments by Mañas et al.[27] (Figs. 3, 4) and Braekeveldt et al.[25] (Fig. 7),

whereas the phylogenetic analyses are novel. All mice were monitored for weight loss and other signs of toxicity. According to the ethics protocol mice were sacrificed when tumors reached 1800 mm³ or at a humane end point due to overall health deterioration. Exceptions were not made from this protocol. Mice were housed in a controlled environment with ad libitum access to food and water. Temperature was set to between 20 and 24 °C to ensure optimal thermal comfort for the mice. Relative humidity (RH) was maintained between 45-55%. Lights were turned on from 6:00 to 18:00 (12 h light/dark cycle), including dusk and dawn periods.

NMRI nude were purchased from Taconic, while NSG mice were obtained from in-house breeding[25,27]. The number of animals used for each experiment is specified in Figs. 3a, e and 4a. For cisplatin experiments (Fig. 3a) female 8–12 weeks old NMRI nude mice were used. For COJEC experiments with progressive growth (PDX1; Fig. 3e) female 8–12 weeks old mice were used; controls and two COJEC-treated mice were NSG, the remaning COJEC treated were NMRI nude. For COJEC experiments with response (PDX3; Fig. 4a) female 8–12 weeks old NMRI nude mice were used. PDX cells were cultured as free-floating tumor organoids in serum-free stem cell medium[26,50]. Mycoplasma infection was tested for prior to the experiments, using standard methods, and was confirmed to be negative. PDX cells (2 ×10⁶) were suspended in a 100 μl mixture of the medium and matrigel (Corning, Cat No.354234) (3:1) and injected subcutaneously into the flanks of female mice. Tumor size was measured using a digital caliper and calculated with the formula V = (πls2)/6 mm³ (where l is the long side and s is the short side). Mice were randomly allocated to control, cisplatin, COJEC or resection groups once their tumor had reached approximately 500 mm³. The cisplatin group was treated with intraperitoneal injection of 4 mg/kg cisplatin (Selleckchem, Cat No.S1166) each Monday, Wednesday and Friday, and the control group was treated with equal amounts of saline. The COJEC group was treated with intraperitoneal injections of cisplatin (1 mg/kg; Selleckchem, Cat No.S1166) and vincristine (0.25 mg/kg; Santa Cruz, sc-201434) on Mondays, etoposide (4 mg/kg; Santa Cruz, sc-3512) and cyclophosphamide (75 mg/kg; Santa Cruz, sc-361165) on Wednesdays, and carboplatin (25 mg/kg; Santa Cruz, sc-202093A) on Fridays. When the tumor size had reduced to approximately 200 mm³, the tumor was surgically removed and mice were then observed for tumor regrowth. For the resection only group, >90% of the untreated tumor was surgically removed and the mice were then observed for tumor regrowth. PDXs established through orthotopic implantation of undissociated patient tumor fragments into the paraadrenal space of NSG mice using Matrigel, were subjected to serial in vivo orthotopic passaging up to eight generations for each model. Tumor samples from in vivo generations 1 and 7 were analyzed here.

## In vitro chemotherapy exposure

Two neuroblastoma cell lines were used; IMR-32 (ATCC CCL-127) was maintained in RPMI-1640 medium (Gibco, Cat No.21875-034) and SK-

N-SH (a gift from Dr. Daniel Bexell, Lund university) was maintained in MEM (Gibco, Cat No.31095029). Both media were supplemented with 10% fetal bovine serum and 1% penicillin-streptomycin and the cells were grown at 37 °C and 5% $CO_2$. The cells were confirmed to be mycoplasma negative prior to the experiments. For passaging and cell counting, cultured cells were washed by phosphate-buffered saline (PBS), disaggregated by trypsin (HyClone, Cat No.SH30236.01) and centrifuged at $300 \times g$ for 5 min. Cells were collected and saved as a frozen pellet at −80 °C until DNA was extracted. As reference, the cell cultures from which cells were seeded for each experiment (mother culture) were analyzed.

Six different experiments were performed with the IMR-32 cells (Supplementary Fig. 4), experiments 1–3 (samples A, B and C) were performed in one batch, followed by experiment 4 (samples SB), and lastly experiments 5 and 6 (samples MB cis and MB).

(1) **IMR-32 continuous growth without bottleneck** (Supplementary Fig. 4b: III samples A1-A3): One million cells were seeded, and the medium was replaced one day after seeding. One third of the cells were passaged to new T25 flasks twice per week. In total, the cells were passaged 11 times during 42 days.

(2) **IMR-32 long-term low-dose cisplatin exposure** (Supplementary Fig. 4b: III samples B1-B3): One million cells were seeded. The next day, the original medium was replaced with medium containing 0.1 μM cisplatin. The medium was changed twice per week to new cisplatin-containing medium. Cisplatin suppressed cell growth in the beginning but the population started to expand while exposed to cisplatin at the end of the experiment. Overall, the medium was changed 12 times and the cells were passaged twice during 42 days.

(3) **IMR-32 single bottleneck induced by high-dose cisplatin** (Supplementary Fig. 4b: III samples C1-C3): One million cells were seeded, and the cells were treated with 1.0 μM cisplatin for 72 h with start the day after seeding. This reduced the number of cells with 94% compared to cells grown in absence of cisplatin. Viable cells were cultured until confluent in a 6-well plate. This took 42 days which also became the endpoint for experiments (1) and (2) in order to remove time as confounder for experiments (1)-(3).

(4) **IMR-32 single bottleneck without cisplatin** (Supplementary Fig. 4c: III samples SB1-SB3): One million cells were seeded and new medium was added the following day. After 72 h the same number of cells as in group (3) was collected and re-seeded in order to create a bottleneck with similar cell number compared to cells treated with high-dose cisplatin. The cells were cultured until they reached confluence in a 6-well plate. The total experimental time was 14 days.

(5) **IMR-32 multiple bottlenecks induced by cisplatin** (Supplementary Fig. 4d: III samples MB cis1-MB cis3): One million cells were seeded, and the cells were treated with 0.75 μM cisplatin the day after seeding. 0.75 μM cisplatin was used because IMR-32 cells could not tolerate three times exposure of 1 μM as used for the single bottleneck experiment. Viable cells were counted after 72 h exposure to cisplatin and 540,000 cells (mean number alive cells of the three cultures) were cultured until confluence of a 6-well plate. This step was regarded as one bottleneck. One million cells were collected from the 6-well plate and re-seeded to a T25 flask. The same procedure was repeated two more times in order to create three bottlenecks. The number of viable cells increased after the second and third exposure to cisplatin compared to the first exposure due to cisplatin resistance as in experiment (2), but the same number of cells (540,000) were seeded at each round. Cells were cultured until they became confluent in a 6-well plate after the third bottleneck. The total experimental time were 53, 46 and 49 days for MBcis1-MBcis3, respectively.

(6) **IMR-32 multiple bottlenecks without cisplatin** (Supplementary Fig. 4d: III samples MB1-MB3): One million cells were seeded and

the medium was changed one day after seeding. As in experiment (5), 540,000 cells were reseeded after 72 h in order to create the same size of bottleneck as for MBcis1-MBcis3. These cells were cultured until they reached confluence in a 6-well plate. This step was regarded as one bottleneck. Two more bottlenecks were created and the cells were collected after reaching confluence in a 6-well plate. The total experimental time was 42 days.

To validate the findings in chemotherapy treated IMR-32 cells, two different experiments were performed with SK-N-SH cells in one batch (Supplementary Fig. 4f).

(1) **SK-N-SH bottlenecks induced by cisplatin** (Supplementary Fig. 4f samples cis1.1-5.1 and cis1.2-5.2): An experimental setup involving two consecutive bottlenecks, was performed. Cells to be used as controls of the original culture were collected at two time points, one at day zero (CC.1) and one three days later (CC.2). At day zero, one million cells were seeded in eight T25 flasks. At day one, media was changed in all eight cultures and cisplatin was added to five of the flasks to a final concentration of 4 μM. The cells were treated for 72 h resulting in approximately 80% cell death. Three days later (day 7), the cisplatin treated cells had started to proliferate and their morphology were changed to large substrate adherent cells, making the culture almost confluent. Cells from each flask (500000) were re-seeded and 4.0 μM cisplatin was added again the following day. The remaining cells from each flask (cis1.1 - cis5.1) were saved as frozen pellets to be analyzed by SNP-array. After the second hit with cisplatin it took 25 days for cells to become almost confluent and then, at day 33, these cells were pelleted and saved for analyses (cis1.2 - cis5.2).

(2) **SK-N-SH continuous growth without bottleneck** (Supplementary Fig. 5f: samples ctrl 1.1–3.1 and 1.2-3.2): Cells in three untreated cultures were continuously passaged at 1:3 approximately twice a week (in total eight times) and fractions of the cells were collected as time point controls (ctrl1.1- ctrl3.1 respectively ctrl1.2-ctrl 3.2) in parallel with the cisplatin treated cells. We did not get SNP-array data from cis3.2 due to insufficient DNA amount.

## DNA extractions

From fresh frozen tumor and blood samples, DNA was extracted using the DNeasy blood and tissue kit (Qiagen, Cat No. 69504). DNA from PDX snap frozen tumor pieces and from frozen IMR-32 and SK-N-SH cell pellets were extracted using the AllPrep DNA/RNA Mini Kit (Qiagen, Cat No.80204) with standard methods. For formalin fixed paraffin-embedded tissue (FFPE) DNA was extracted using the Allprep DNA/RNA FFPE kit (Qiagen, Cat No. 80234) with the following modification: during deparaffination, two 10-minute incubations of the samples with 1 ml xylene at RT were performed to ensure complete removal of paraffin. DNA concentration was measured using Qubit DS HS DNA (Invitrogen Cat No. Q32854).

## Overall approach to genomic profiling and phylogenetic analysis

The high proportion of CNAs compared to driver point mutations complicates evolutionary analyses of NB. Many types of chromosomal rearrangements violate the infinite sites hypothesis on which conventional phylogenetic analyses rest[51]. In particular, aneuploidy can affect only the limited number of chromosomes present in a cell and is thus frequently the subject of parallel evolution and back mutation. Applying phylogenetic methodology to NB cells thus requires adaptation of methods to incorporate features of chromosome-level evolution, for example by allowing parallel evolution of whole chromosome alterations and back mutation from whole chromosome gains while treating chromosomal breakpoints outside centromeric regions as unique events and loss of heterozygosity as irreversible[52]. We have recently reported a software tool (DEVOLUTION, https://

github.com/NatalieKAndersson/DEVOLUTION) that allows input of point mutations in parallel to CNA data, where the latter is curated according to the constraints of chromosomal evolution[22]. DEVOLUTION integrates sequencing data and CNA profiles by clustering genetic alterations based on their clone sizes (mutated clone fractions) across samples. These clusters, together with the criterion that the sum of parallel subclones must not exceed 100% in any biopsy is used to deduce the most likely subclone configurations across biopsies, including if subclones are nested within or parallel to each other. The algorithm generates an event matrix encompassing the detected subclones. This matrix was used as input for phylogenetic reconstruction to deduce their evolutionary relationship using the maximum parsimony method or maximum likelihood method employing the R package phangorn (v2.8.1)[53], and visualized using the ggplot2 (v.3.3.5) package. Phylogenies were rooted in a cell having no genetic alterations. The manual "How to build MP and ML trees from a segment file using DEVOLUTION" is available at Zenodo.

To perform high-resolution mapping of subclonal territories across anatomic tumor space in NB patients we used archived FFPE tumor tissue. This allowed us to make a comprehensive investigation of tumor cells left in primary tumors after chemotherapy, even in cases with good chemotherapy response where survivor populations are often limited to spatially dispersed tumor islets a few millimeters in size, surrounded by necrosis, hemorrhage and fibrosis. Areas of ~20 mm² containing >30% tumor cells were subjected to whole genome copy number profiling parallel to whole exome sequencing (WES). Mutations detected by WES were validated by targeted resequencing using a custom-made panel based on the WES findings. Analyses were done on FFPE tumor in all cases except three (Patient 6, Patient 9 and Patient 12) from which DNA was extracted from frozen tumor samples. Combining CNA and SNV/indel data resulted not only in an increased number of mutations being found in each sample, but also the number clones that could be deconvolved from each sample (Supplementary Fig. 1m, n). Tumor cell fraction had a slight impact on the number of SNVs/indels detected in each sample but no significant impact on the number of detected CNAs. An overview of the workflow from raw SNP array and sequencing data to phylogenetic trees can be found in Supplementary Fig. 1p.

### Copy number analyses by SNP array

CytoScan HD SNP array (Thermo Fisher/Affymetrix) was used and the copy number profiles were visualized with Chromosome Analysis Suite (version 3.3.0.139; Thermo Fisher/Affymetrix) to analyze DNA from fresh frozen samples and cell culture experiments. CEL files were normalized by the R package Rawcopy[54], and scatter plots whose x-axis and y-axis denotes log2 ratio and allelic imbalance, respectively, were generated through the R package TAPS (Tumor Aberration Prediction Suite)[55]. DNA extracted from FFPE samples were analyzed with the Oncoscan array platform at the Swegene Center for Integrative Biology (SCIBLU) according to the manufacturer's recommendations. The CEL-files were converted to OSCHP-files with the OncoScan Console software (version 1.3.0.39; Affymetrix). The Nexus Express software for Oncoscan (version 3.1) was used for visual inspection of the OSCHP-files. Three text-files containing information regarding probes, segments and SNPs were exported from the Nexus Express software and used as input to create TAPS plots. Only copy number segments larger than or equal to 0.1 Mb were included in downstream analyses. Constitutional segments were excluded via the Database of Genomic Variants[56].

Aberrations identified in the clinical samples were summarized in circos plots, one plot per patient with individual samples illustrated as concentric circles. These plots were generated with BioCircos (v0.3.4) in R (v4.1).

### Whole exome sequencing of PDX samples

Equal amounts of DNA from two samples from the same PDX were combined and analyzed with whole exome sequencing at the Center for Translational Genomics at Lund university as detailed in ref. 57. The Sure Select XT HS Library Prep (Agilent) was used and paired-end sequencing (2×150 cycles) was performed on a NextSeq500 (Illumina). bcl2fastq 2.20 (Illumina) was used to convert files to fastq-formatted files and paired end reads were mapped to the human reference genome (GRCh37 with decoys from the 1000 Genomes' Project) using BWA-MEM 0.7.15[58]. Duplicate reads were marked using sambamba 0.6.7[59]. Somatic variant calling was performed using freebayes (with the –pooled-continuous, --pooled-discrete and -F 0.03 flags; https://arxiv.org/abs/1207.3907v2) and strelka2 (using the --exome flag)[60]. Contaminating reads from the mouse genome were removed. A variant was excluded from the study if detected by less than 10 reads, had a variant allele frequency (VAF) below 0.1 in all samples, was present in more than 1% of the reads in the corresponding normal sample and/or showed inferior quality at visual inspection of the reads in Integrative genomics viewer[61]. The Polyphen-2 tool[62], was used to predict the impact of mutations on protein function.

### Whole exome sequencing and targeted sequencing of clinical samples

DNA repair was performed on DNA extracted from FFPE-material using the NEB PreCR kit (New England Biolabs, Cat.No. M0309S) following the manufacturer's recommendations, using 500 ng of DNA. The repaired DNA was purified using AMPure XP beads (Beckman Coulter, Cat No. A63881) at a 1.8X ratio and eluted in Illumina Truseq Exome kit RSB buffer (Illumina). Library preparation was performed using the Truseq Exome kit (Illumina, Cat No.20020614). DNA fragmentation was performed on a Covaris S220, and end repair, size selection, 3' end adenylation, adaptor ligation and DNA enrichment steps were performed according to the protocol recommendations. Libraries were quantified with the Qubit DS HS DNA kit and pooled in an equimolar manner. Exome capture, amplification (8 cycles) and purification steps were performed as recommended. Samples were quantified using Qubit dsDNA HS kit, and library quality was assessed on Bioanalyzer, using the Agilent High Sensitivity DNA kit (Agilent Cat No. 5067-4626) before sequencing. DNA samples were analysed either in-house with the NextSeq 500/550 high output kit v2.5, 300 cycles (Illumina, Cat No. 20024908) on a NextSeq500 (Illumina) or at the Center for Translational Genomics at Lund University using the NovaSeq 6000 S1 reagent kit v1, 300 cycles (Illumina, Cat No. 20028317) on the NovaSeq6000 (Illumina). Illumina paired-end reads were aligned to the human reference genome hg19 by BWA-MEM (https://arxiv.org/abs/1303.3997v2). Duplicate reads marking and local realignment were performed by GATK (version 4.0.11.0)[63]. Mutect (version 1.1.7)[64], GATK Mutect2 (version 4.0.11.0), and MuSE (version v1.0rc)[65] were used to identify somatic single nucleotide variants (SNVs) and small insertions/deletions (indels). The SNVs and indels called by Mutect2 were further filtered with the GATK FilterMutectCalls. The variant vcf files were converted into maf files by the vcf2maf package (https://github.com/mskcc/vcf2maf). DNA from fresh frozen clinical samples from Patient 6, Patient 9 and Patient 12 were analyzed in the same way as the PDX samples, but the variant calling was performed jointly for all tumor samples from each patient using Mutect2 (https://doi.org/10.1101/861054) with a panel consisting of 20 unrelated exome samples used as normals, followed by variant filtering with FilterMutectCalls including the read orientation filter[65], according to the best practice guideline for somatic short variant discovery[66]. Variants were excluded using the same criteria as for the sequencing of PDX samples but a VAF cut-off at 0.2 was used for the FFPE extracted samples. Very few mutations were detected in samples from Patient 3 and Patient 7, hence a VAF cut-off value at 0.1 was used for these samples.

Targeted re-sequencing of 454 mutation in total that remained after filtering of WES data as specified above was performed with at the Center for Translational Genomics at Lund University using a Twist Custom probe Capture panel with the Twist library preparation EF Kit (product no. 101058, Twist Bioscience). The samples were sequenced on a NovaSeq 6000 (Illumina). Raw reads were processed through the Sentieon® unique molecular indices (UMI) aware pipeline (https://support.sentieon.com/appnotes/umi/). Briefly, UMIs were extracted from raw fastqs and raw reads were aligned to the reference genome (GRCh37) using the sentieon implementation of bwa mem (https://arxiv.org/abs/1303.3997v2), followed by consensus fastq generation using the sentieon consensus tool (https://www.sentieon.com/products/). The consensus reads were then mapped to the reference genome (again using the sentieon implementation of bwa mem) and the resulting bam-files were processed through freebayes (https://arxiv.org/abs/1207.3907) with the following settings: --pooled-continuous, --pooled-discrete, --min-repeat-entropy 1, -F 0.03 on a per patient basis, i.e. all samples from each patient were called jointly. Variants were excluded if detected in >1% in the corresponding normal sample, had a VAF value below 0.1 in all samples and were covered <100 reads in all samples. The mean total coverage of the genomic location for mutations included in the study was 545x. All variants were annotated using the VEP tool (https://grch37.ensembl.org/Homo_sapiens/Tools/VEP). To identify relevant mutations to highlight in the phylogenetic trees (Figs. 1–2, Supplementary Fig. 1) and heat maps (Supplementary Data 1d), missense mutations flagged to have an impact by PolyPhen or SIFT in genes correlated to neuroblastoma, tumors in general or neuronal differentiation according to Pubmed (https://pubmed.ncbi.nlm.nih.gov) were selected. The transcript with the highest predictions score was shown at protein level.

## Clone size estimations based on SNP array data

A log2 ratio (log2R) value was obtained from SNP-array analyses for all identified aberrations. This is the ratio between the total number of alleles at the analyzed location and a standardized normal copy number which can be described as:

$$\log2R = \frac{MSF * Nt + (1 - MSF) * Nb}{Np} \tag{1}$$

The mutated sample fraction (MSF) is the portion of the entire sample, including normal cells, that harbors the aberration, Nt is the copy number of the aberration, Nb is the copy number of the background cells (either the number of alleles in a parallel clone or the ploidy of the tumor) and Np is the ploidy level the array is normalized against. The MSF can be calculated with:

$$MSF = \frac{Np * \log2R - Nb}{Nt - Nb} \tag{2}$$

For array data from Cytoscan HD, the log2 value needs to be divided by a correction factor of 0.53-0.6[67]. The MSF value can also be calculated using the mirrored B-allele frequency (mBAF) where the number of B-alleles is higher or equal to the number of A-alleles.

$$mBAF = \frac{Btot}{Atot + Btot} \tag{3}$$

The total number of B alleles, Btot, in a sample at a specific location is the sum of the B-alleles in the tumor ($N_B$) and the B-alleles in the normal cells, which is assumed to be one.

$$B_{tot} = B_{tumor} + B_{normal} = N_B \times MSF + 1 \times (1 - MSF) \tag{4}$$

The number of A-alleles can be described accordingly

$$A_{tot} = N_A \times MSF + 1 \times (1 - MSF) \tag{5}$$

If combining (3), (4), and (5), and simplifying the expression, MSF can be calculated with

$$MSF = \frac{1 - 2mBAF}{mBAF(N_A + N_B - 2) - N_B + 1} \tag{6}$$

The mBAF-value can be obtained from the TAPS plot since the allelic imbalance (AI) on the y-axis of these plots is defined as

$$AI = \frac{mBAF - 0.5}{0.5} \tag{7}$$

## Clonal deconvolution based on CNAs detected by SNP array in clinical samples

The SNP-array data were visualized in TAPS plots[55], where all chromosome segments in one sample are shown in the same plot with the log2R on the x-axis and the allelic imbalance on the y-axis. Thus, all normal chromosomal segments are displayed in the same cluster with the log2R and allelic imbalance equal to zero, while clonal CNAs appear in a grid like pattern where each junction represents a specific combination of A and B-alleles. Subclonal CNAs can be identified in the grid since their clusters deviate from these junctions. All CNAs were annotated with the lowest number of alleles possible, for example a CNA situated between the 1 + 2 and 1 + 3 junctions in the TAPS plot was annotated as a subclonal 1 + 3.

An aberration was called as "mixed" if present with two different allelic compositions in the same sample, as indicated by cluster positions in TAPS plots deviating from the grid. Events that were too small to be visualized in the TAPS plot were excluded from the study since the allelic imbalance could not be determined with certainty. The median of MSF values for aberrations that appeared to be clonal based on the TAPS plot were used to establish the tumor cell fraction (TCF) or purity of the sample. This was done both based on the log2R-values and on the allelic imbalance. Similar results (difference <approx. 0.1) from these calculations for the same aberrations was used as an indication of correct assessment of ploidy-level and allelic composition.

The mutated clone fraction MCF, describing the fraction of the tumor cells that contains an aberration were then calculated as

$$MCF = \frac{MSF}{TCF} \tag{8}$$

The MCF values based on the log2R was used for most aberrations, with the exception for CNNIs that don't affect the log2R -data, for rare occasions when the log2R was noisy while the data was stable based on mBAF-values or when the visual appearance of the TAPS plots reflected the AI based MCF-values the best. Amplifications and homozygous deletions were always scored as clonal. The MSF value for a "mixed" aberration was calculated for the largest of the subclones first and the value for the smaller subclone was calculated as 1-the MSF for the biggest. The MCF values were visualized in a heat map format and variants with similar MCF pattern over samples were grouped together (Supplementary Data 1d). An average of the MCF values per sample and group were calculated and that value rounded to the nearest 0.1 became the final clone size for all the variants in that group. When the appearance of the heat map indicated that a consistent phylogenetic tree could not be created, an event could be added as an "inferred stem" or "inferred private". For example if all samples derived from one patient demonstrated a gain with the same breakpoints but containing different number of alleles in different samples, the gain with the

lowest number of alleles was inferred as stem in all samples. Another example could be if a whole chromosome aberration was present in several samples, and it was impossible to visualize a logic order of events for the identified mutations, the whole chromosome events could be annotated as inferred private, i.e. separate events, since they are likely to appear in parallel. Such occasions are noted in Supplementary Data S1d. The information from the curated heat maps was used as input data for DEVOLUTION to construct phylogenetic trees. Two subclones containing the same aberration but with different allelic compositions (the scenario we called "mixed") cannot be nested events since that would indicate that cells in the nested population would have two different CNA states at the same time. In such cases, we provided the DEVOLUTION algorithm with a matrix indicating that these genetic alterations should not be nested into each other to avoid illicit evolutionary trajectories.

The maximum likelihood and the maximum parsimony trees obtained from DEVOLUTION were identical for all patients with the exception of the trees based on samples from Patient 6. Here mainly whole chromosome aberrations were detected and the absence of distinct breakpoints indicative of a relationship between subclones made it difficult to differentiate between parallel private and shared events. In this case, the tree containing the smallest number of back mutations (maximum parsimony) was chosen.

### Calculation of clone sizes based on integration of variant allele frequencies from exome sequencing with copy number from SNP-array data

The variant allele frequency (VAF) is derived from sequencing data and describes the number of reads detecting a specific mutation divided by all reads originating from tumor cells and diploid normal cells in the sample.

$$VAF = \frac{M \times MSF}{CN1 \times f1 + CN2 \times f2 + 2(1 - TCF)} \qquad (9)$$

where MSF is the mutated sample fraction, i.e. the proportion of the entire sample, including the normal cells, that contains the mutations. M is the number of mutated alleles, TCF is an average of the log2R and allelic imbalance values described above, and CN1 is the total number alleles in the main fraction of the tumor at the location of the mutation, defined by SNP-array, and f1 is the size of fraction 1. CN2 is the number of alleles in a background clone if present, and f2 is the size of CN2, and

$$TCF = f1 + f2 \qquad (10)$$

If Eqs. (8), (9) and (10) are rearranged and combined, the MCF can be calculated as

$$MCF = \frac{VAF \times (CN1 \times f1 + CN2 \times (TCF - f1) + 2(1 - TCF))}{M \times TCF} \qquad (11)$$

As a starting point, M was assumed to be 1 for diploid samples and 2 for tetraploid samples. For mutations located on a subclonal CNA or on a mixed background (two subclones with different CNAs), M could take different values. For example, if two different CNAs were present at the location of a specific mutation at for example $2 + 0$ at 30% and $1 + 2$ at 70%, M for that variant could be 0.7, 1.4 or 2, depending on the order of events. If several theoretical values for M were possible, the following rules were used to choose the M-value to be used for the MCF-calculation:

- If the MCF value became higher than 1.2, a higher value of M was chosen
- The choice of the M-value should not lead to a biologically impossible order of event, for example a completely lost (nullisomic) allele cannot be gained at a later stage.

- The generated MCF values should not violate the pigeon whole principle
- The order of events for a CNA and a SNV should be the same for all samples harboring identical aberrations

If more than one solution for M still remained, the value was chosen that introduced the fewest number of novel subclones so as not to overcall heterogeneity. The value of M chosen for calculation of MCFs for the specific variants are included in Supplementary Data S2b.

As benchmarking to our heuristic approach, we also re-estimated the MCF values with the phylogenetics tool DeCiFer v2.1.4[68]. It can notably quantify the proportion of cells which acquired an SNV or whose ancestors acquired the SNV inferring corresponding multiplicities across samples. We compared the MCFs estimated with DeCiFer, which extensively confirmed our calculations (see Supplementary Fig. 8 and its legend for details). DeCiFer uses density-based cluster assignment with variable epsilon boundary that can be defined with the n-sigma limit. By default, a fairly strict grouping is used (1-sigma limit), which means many data points from our dataset were not assigned to any cluster. In addition, we were unable to compare results of Patient 3, a pentaploid case due to state trees being unavailable in DeCiFer for the copy number alterations in question.

### Clonal deconvolution from SNP array data of cell lines and PDX cells

CNAs and allelic imbalances were classified as clonal or subclonal based their relative distribution in whole genome TAPS-plots. Subclonal events were subjected to a more precise clonal deconvolution. The MCF values were calculated for all aberrations using Np=2, and if the MCF value was over or equal to 90 %, the aberration was considered as clonal.

The final clonal deconvolution for phylogenetic analysis was based on the following rules:

- Because of an error margin of 10% at clone size estimation by SNP array[69], final clone sizes were rounded to the nearest 10%. The sum of subclone sizes was allowed to reach a maximum of 120%, because two concomitant subclones will yield a 20% error margin.
- Tumor cell concentration was normalized to 100% when the total sum of subclones reached over 100%.
- CNNI events $(2 + 0)$ in a sample against a background of trisomy $(1 + 2)$ was assumed to occur by losing an allele from the trisomy if the majority of the other samples from the same experiment displayed trisomies $(1 + 2)$.

For cell line experiments, a private aberration was defined as an aberration unique for each lineage, while for PDX experiments (two samples per tumor) a private aberration was defined as an aberration unique for each PDX tumor. All other aberrations were defined as shared or stem.

### Clonal deconvolution from sequencing data from PDX models

SNP array data from the two samples from the same PDX tumor were combined and segmental aberrations present in the main clone were used to calculate the mutated clone fraction (MCF) for variants. Because reads from murine DNA were removed at analyses of PDX samples, the tumor cell content was assumed to be 100 %. The majority (>90%) of the detected mutations were located on normal diploid alleles or on segments that were clonally deviant, hence f1 = 1 and the MCF was calculated by:

$$MCF = \frac{VAF \times CN}{M} \qquad (12)$$

Mutations with an MCF > 0.7 were considered to be clonal. A variant detected on an unbalanced segment $(1 + 2)$ was assumed to be

clonal if the same variant was judged as clonal in a balanced setting. This was achieved by choosing the M that produced an MCF close to 1 in the unbalanced setting. The cut-off value for clonality was lowered to MCF > 0.6 for a mutation detected on a 1 + 1 segment that was shared between several samples and clonal in the majority (90 %) of them. A shared variant present at a low frequency (VAF < 0.20) in a sample where a clonal sweep had taken place was considered as a technical artifact and removed.

#### Deconvolution of clinical samples after targeted re-sequencing

MCFs were calculated by using the average TCF (purity) values obtained by calculations from the log2-ratio and from the allelic imbalance values from the TAPS plots. The number of mutated alleles M was assumed to be 1 in a diploid setting but changed to 2 if the MCFs > 1.2 in the majority of the samples where the mutation was identified. Our inclusion criteria for variants (see above) could result in MCF-values above 1.2 in some samples. This could be due to for example low coverage in the specific sample or sublconal background poly-ploidization. These variants were kept in the study since they were clearly detected. Their MCF-values were rounded down to 1.0 and thus did not induce false heterogeneity. The final MCF values were obtained by using the same heat map approach as for the MCF calculations on SNP array data on clinical samples. After quality filtering and deconvolution 359 of the 454 variants from the TWIST design were retained for phylogeny construction by DEVOLUTION.

#### Quantitative analyses of genetic diversity and phylogenetic trees

The index of genomic diversity (IGD) was constructed to minimize the impact of differences in number of aberrations detected per sample and differences in the number of samples per patient, when comparing the diversity between sample types (i.e. samples taken pre or post treatment or from a metastasis). The lengths of the branches ($dS_i$) starting from the closest common node for all subclones (n) identified for a sample type, to all (n) the individual subclones ($dS_1$, $dS_2$, … $dS_n$), were annotated together with the distances ($DS_i$) from the start of the stem to the same subclones ($DS_1$, $DS_2$,..$DS_n$). IGD was calculated as the ratio between the sums of these two different distances:

$$IGD = (dS_1 + dS_2 + ..dS_n) / (DS_1 + DS_2 + ..DS_n) \qquad (13)$$

IGD varies between 0 and 1, where a higher IGD-score indicates more genetically diverse subclones.

Irregularity describes how the phylogenetic tree deviates from a symmetric star phylogeny. Branch lengths, i.e. the distances from the stem to the subclones in the tree are annotated and irregularity is calculated as the variance of all branch lengths.

#### Reconstructing subclone geography

Leveraging that the paraffin-embedded material was regularly sliced down to around 3 mm depth across biopsies and patients to obtain material for DNA extraction allowed us to approximate territories across a two-dimensional spaces, with surface areas corresponding to subclone sizes. Relative subclone sizes were calculated as MCFs as described above. While subclones detected in more than one tissue samples were assumed to be located around the interface between these biopsies, subclones ascertained only in one biopsy was placed in the remaining space. If found only in a single sample, nested subclones were placed in the geographic center of their mother clones.

#### Single cell WGS

Single cell copy number profiles from shallow WGS were analyzed for clonal identification used in phylogenetic analyses and fishplots[70], of PDX tumors and tumor organoids. Identified copy numbers encompassing less than five consecutive 1 Mb bins in the dataset, were excluded from further analysis, except for high-grade amplifications of the *MYCN* region in chromosome arm 2p and of the *MAML3* region in 4q28.3-q31.1, in which case two consecutive 1 Mb bins were considered true gains[11].

Single cell copy number profiles from shallow WGS were used as input to a custom-made algorithm, latest version found on github (https://github.com/NatalieKAndersson/SCEM). The input file is a matrix where each column is a single cell, each row a chromosomal segment, and the matrix elements are the copy numbers of chromo-somal segments in single cells. The algorithm identifies all genetic alterations present in each single cell while applying an optimized cut-off of five consecutive 1 Mb bins for identifying true imbalances, and combines the information from all analyzed single cells to identify consecutive events encompassing overlapping chromosomal seg-ments. The output was an event matrix that was used to perform phylogenetic reconstruction to deduce the evolutionary relationship between individual cells using the maximum parsimony method or maximum likelihood method employing the R package phangorn (v2.8.1)[53], and visualized using the ggplot2 (v.3.3.5) package. Phylo-genies were rooted in a cell having no genetic alterations.

#### Reporting summary

Further information on research design is available in the Nature Portfolio Reporting Summary linked to this article.

### Data availability

The datasets generated or analysed during the current study are available in curated raw data format in Supplementary Data 2. Raw data plots from whole genotyping of clinical samples, PDX tumors and cell lines are available at Zenodo (https://doi.org/10.5281/zenodo.8017767). Input files used for DEVOLUTION and the subclone anno-tation derived from running DEVOLUTION are available at Zenodo together with single cell WGS input files for SCEM (https://zenodo.org/records/10727558). Raw data have been deposited the European Genome-phenome Archive (https://ega-archive.org) under accession number EGAS00001007650, with Dataset IDs as follows: -Whole gen-ome genotyping/targeted sequencing: EGAD00001015012 -Whole exome sequencing: EGAD00001015413 -Single cell whole genome sequencing: EGAD00001015414. Access to raw genomic data will require assessment by a data access committee of the applicant's ethics permit as well as technical specifications of how the data should be stored to ensure compliance with ethics regulations of the study as well as GDPR. A data transfer agreement will have to be made and signed by legal representatives. Source data are provided with this paper.

### Code availability

The software DEVOLUTION was used for clustering and phylogenetic analyses of bulk sequencing data (https://doi.org/10.5281/zenodo.13304334)[71]. The full software including access links is published in Andersson et al.[22] The software SCEM used to construct phylogenies from single cell WGS is available online at https://github.com/NatalieKAndersson/SCEM.

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

## Acknowledgements

The present study was supported by grants from the Swedish Research Council, the Swedish Cancer Society (21 1383 Pj), the Swedish Childhood Cancer Foundation (PR2022-0026), Regional Clinical Research grants (ALF Projekt0067), the Royal Physiographic Society, and the LMK Foundation. We thank Center for Translational Genomics, Lund University and Clinical Genomics Lund, SciLifeLab for providing sequencing service. The illustration of a mouse used in Figs. 3a and e and 4a is a slight adaptation of "Vector diagram of a laboratory mouse (black and white)" by Gwilz/Wikimedia Commons/CC BY-SA 4.0.

## Author contributions

J.K., N.A., S.C., K.P, F.F., D.B., and D.G., designed the study. H.Y., A.M., K.H., K.A., C.J., G.D., N.R., J.K., and M.F. performed experimental work. J.K., H.Y., A.M., N.A., K.A., M.Y., N.R., S.C., D.S., F.F., A.V., and D.G. analyzed data. D.G. and J.K. selected patients from clinical material.

## Funding

## Competing interests

D.B. has received research funding from Healx, aPODD foundation, and Captor Therapeutics for unrelated work. The other authors declare no competing interests.
