## [Peer Review File · Nature Communications]

REVIEWER COMMENTS

Reviewer #1 (Remarks to the Author): Expert in brain cancers and preclinical models

The manuscript Early evolutionary branching across spatial domains predisposes to clonal replacement under chemotherapy in neuroblastoma by Karlsson et al. explores the temporospatial heterogeneity and clonal evolution of primary neuroblastoma (NB) through therapy, using clinical specimens and patient-derived xenograft (PDX) in vivo modeling in combination with precision in silico modeling of clonal phylogenies. They show that tumors responding to treatment comprise a unique clonal landscape at relapse, whereas those that do not respond possess ancestral subclones detectable at diagnosis.

The authors do an excellent job of addressing alternative hypotheses that might explain their data, and they refute these hypotheses where possible, while providing adequate explanation where not (i.e., effective spatial sampling of in vivo PDX models pre-treatment when the xenografts are not very large). Previous studies have compared primary and relapsed NB to identify drivers of relapse, but the dynamic changes of specific subclones with respect to treatment refractory or treatment-sensitive subpopulations through therapy have not been well understood. The manuscript is also generally well-written, and the figures are extremely well done. Hence, this is an important study that will be broadly appreciated by the cancer research readership.

The following points should be addressed by the authors.

Comments:

- The study is underpowered, and the authors could comment on how a broader sampling of patient tumors through therapy might change the findings. For example, line 106-109 draw conclusions from single tumors. This could be described in a limitations/future directions section.
- It is difficult to discern whether “low dose” or “high dose” cisplatin are effective representations of stable disease through treatment vs treatment sensitive. What effect does low dose have on the cell population? Are cells dying? Preserved clonality could simply reflect a lack of cytotoxicity at the dose used in vitro.
- Can the authors better explain/cite the tumor organoid model, line 263? It appears quite abruptly.
- How are the authors inferring the mesenchymal state from their findings? Is it because of resistance to chemotherapy? It is unclear from the cited literature whether the chemotherapy resistant state in the present study is fueled by mesenchymal characteristics. Perhaps the authors could use their organoid model to explore how mesenchymal characteristics are related to resistance to chemotherapy in this model.
- Karlsson et al., discuss that their study has implications for how to sample NB tumors from genomic profiling based on the evolutionary branching that is set early on in NB pathogenesis. Information regarding the feasibility and limitations should be included in the discussion. Also, the authors should discuss liquid biopsies as an alternative to multiple sampling to capture tumor heterogeneity. For

example, the following study characterises pediatric medulloblastoma tumors by analysing the cerebrospinal fluid circulating tumor DNA revealing intra-tumour genomic heterogeneity (See DOI: 10.1038/s41467-020-19175-0)

Minor points:

- Ensure all abbreviations are defined, for example MNA NB line 178
- Consider minor typos in the text (examples below)
 - o Abstract, line 20 "...due to its propensity to resist treatment"
 - o Line 40 "...there are now a broad range..."
- Abstract, line 23 "...under effective treatment". Effective isn't a clear term here. Consider using terms like treatment refractory, treatment sensitive etc.
- Line 243 "figure 6" should specify which panel
- Supplemental figures are very long, can a summary/table be provided?

Reviewer #2 (Remarks to the Author): Expert in neuroblastoma genomics, therapy and clinical research

In this well written manuscript Karlsson and colleagues from the Gisselsson group study the clonal evolution of neuroblastomas under the selective pressure of chemotherapy, an area which remains poorly understood. They study multi-region samples at diagnosis, post therapy and relapse from 12 patients with neuroblastomas, most but not all, with high-risk disease. They use PDXs models and a cell line to validate their findings. They then apply single cell DNA sequencing to map subclone geographies before and after treatment. The authors conclude that patients with progressive disease harbor de novo mutations that confer up-front resistance to chemotherapy, resulting in predominant resistant clone that evolve in a linear fashion. On the other hand, tumors from patients responding to chemotherapy undergo collateral clonal replacement – a process by which the founding mutations are replaced by new mutations.

The work is timely and could provide an important and clinically relevant observation to the literature. However, while there are important data in this manuscript, there are several concerns that should be addressed to merit publication in Nature Communications.

Major concerns

- The tumors have been collected over a period of 20 years and are apparently an admixture of frozen and formalin-fixed paraffin embedded samples. There is little discussion of data quality, and the data are not available for review (not is it mentioned where it will be made available post-publication). Copy number analysis in particular is sensitive to noisy data. This needs to be addressed and data made

available to the reviewers.

- Variant calling, especially in aneuploid tumors can be challenging and results can have high false positive rates. To fully assess the findings a set of orthogonal variant callers should be used.
- Of the mutations called in the post therapy samples that undergo collateral clonal replacement (CCR) – it is not clear whether these mutations are driver mutations conferring survival advantages or if these are passenger mutations, and this should be clarified.
- . Figure 1: Since the author conclude that there are two patterns of clonal evolution: linear and collateral, it would be useful to have a panel summarizing the main conclusion of the manuscript, according to which densely packed territories of closely related subclones present at diagnosis are replaced under effective treatment by islands of distantly related survivor subclones, by a mechanism of CCR; while in tumors that progressed under treatment, ancestors of subclones dominating later in disease are present already at diagnosis. As currently structured, Figure 1 presents the phylogenies of individual patients and does not convey the main message in a direct graphical manner. Rather, it forces the readers to look at the description of each individual patient to understand the timeline of his/her tumor samples. Consider including an Oncoprint or other summarization of the overall data.
- Figure 2/3: the experiments shown in this figure attempt to model refractory disease. The authors use a cell line derived xenograft (not a PDX) treated with cisplatin and observe linear evolution. The authors use the exact same model in Figure 3, but with multiagent chemotherapy COJEC. Thus, the same CDX responds to an intensive therapy but does not respond to single agent therapy. To make a fair comparison to what is seen in the clinic, the authors should use a CDX or PDX from a refractory tumor (such as one harboring P53 mutation) and treat the PDX with COJEC instead of cisplatin as all patients receive the same induction therapy. Currently, this reviewer does not see what Figure 2 is adding to the overall story.
- Figure 3: the results in this figure suggest that mutations in PTPRD and TCF4 drive resistance in relapse, but it is not clear if this is a reproducible finding or if these are “one-offs” in this model. This needs to be clarified. Do the genes in panel f-g appear in both mice that relapsed or only one?
- Figure 4: The authors attempt to show CCR in vitro. The results should be validated in more than one cell line.
- Figure 6: The findings are interesting. However, it is not clear how these results relate to the entire story of clonal evolution under chemotherapy resistance.

Minor comments

- For the two patients who progressed under therapy, we suspect that this is due to MDM2 amplification in patient 1 and NF1 loss in patient 2. However, this is not discussed in the text and it is unclear what is the author’s hypothesis as to why these patients progressed.
- The authors use ratios of surviving cells in the tumor to document anti-tumor response. While valid, it would be good to know the overall end induction response to chemotherapy for each subject.
- The rationale for including low- and intermediate-risk patients who do not convert to high-risk needs to be carefully stated and how including these subjects in the overall analysis explained.
- You previously mention that Patient 12 was treated with up-front surgery only, but here you say that Patients 4 and 12 were the ones with least chemotherapy-induced regression. Please clarify.
- Line 140. Please specify the meaning of “same” as in “same PDX”.
- 142. Please define the acronym “COJEC” when first reported in the manuscript.
- 161-163. The fact that the finding of CCR under therapy could not be validated in the CDX/PDX model

is a major limitation of the study. Please comment on this limitation in the manuscript.

- 171-172. How can you conclude that the PDX model supports the association between effective chemotherapy response and CCR if the pre-treatment sampling and genetic analysis of PDX was not performed? Please explain.
- 178. Please define the acronym “MNA” when first reported in the manuscript.
- 238-266. In this paragraph you derive your conclusions mostly from the findings in 3 MNA tumors. Please comment on the findings in the other six non-MNA tumors.
- 278. Please define “numerically low” when first reported in the manuscript.
- 329-331. How can you conclude that “in this context, our data suggest that different genetic lineages present in the same patient have different potential for the mesenchymal, chemoresistant state” without analyzing any mesenchymal signature gene? If you have done so, please specify in the manuscript.
- 389. Here you say that one of the spatiotemporal comparison was “pre vs MR”, while the figure 1g says “pre/post vs MR”. Please explain the discrepancy.
- 405. Why does the IGD was calculated on 10 phylogenetic trees only, instead of 11 (12 total patients minus Patient 12 who had up-front surgery). Please explain.
- 415. Please specify how many “yes” and how many “no”.
- 426. Here you say that the PDXs where all MYCN-amplified while in the main text (line 121) you solely mention Stage III NB. Please explain the discrepancy.
- 496. Where is the letter “l” in figure 5?
- 530. I believe “BPC” is misspelled here for “PBC”. Please fix accordingly.

Reviewer #3 (Remarks to the Author): Expert in cancer evolution, phylogenetics, and single-cell genomics

In this work, Karlsson et al. study clonal evolution of Neuroblastoma under various conditions such is treated vs. non-treated, as well as across time and space.

Unfortunately I had a great difficulty deciphering large number of figures presented in this work, especially the phylogenetic trees. Overall, the results do not seem impressive and the analyzed cohort is of a very limited size to derive conclusions with broader implications. I am also reserved about the correctness of the provided trees as relatively low number of bulk-sequenced samples per patient yields limited signal and power to unambiguously resolve phylogenetic trees.

The Authors are referring to their previous tool, DEVOLUTION, that was used in this work to find most of the presented trees. In order to better understand this work, I tried to read DEVOLUTION paper but I faced some challenges that I could not resolve. For example, in that paper it is stated that phylogenetic tree of patient NB5 shown in Figure 2a has polyclonal seeding, with one seeding clone originating from a subclone harboring stem events only and one seeding clone sharing the same copy number profile as Subclone A (this is stated in the caption of Figure 2). If Subclone A does not contain any additional somatic mutations or aberrations compared to the branching point at the right end of the edge labeled with “++19 ++22”, then the two other subclones in the metastatic sample, which are E and C, are having the same profile as A, which contradicts the above statement. Therefore I conclude that A has some

additional events compared to this branching point. But then the variants specific (private) to A appear at frequency approx. 80% (based on the pie chart), whereas variants specific to Subclone C are at approx 50%, which adds up to 130% for two subclones on separate branches, violating Pigeonhole principle (see 10.1101/cshperspect.a026625).

In summary, I find this way of depicting phylogenetic history and clonal and subclonal frequencies not just confusing, but also very difficult to interpret. Could the Authors please provide broadly adopted "clonal trees" where proportions of subclones in different samples are clearly indicated instead of giving frequencies of variants specific to clones? We are primarily interested in clonal dynamics and seeing which subclones die out and which ones emerge and/or take over. It would be much simpler and easier to interpret figure if this information is directly given, instead of having to derive it by subtracting children from parent frequencies.

A solution that I recommend is to provide figures that are combination of what is shown in panels d and e in the figure shown at <https://github.com/hdng/clonevol>. For example, a sample S can be associated with, say, 100 small circles of the same color (the number can be less than 100 but should be the same for all samples). If it is estimated that 25% of cells from S belong to subclone 4, then subclone 4 (that is, tree node 4) can be assigned 25 dots of the color of S (better contrasts between colors or some other way to better distinguish samples will be needed - see below). Mutations can be assigned along the edges of the clonal tree and potential metastatic seeding events can be clearly marked by pointing to the edges along which migration happened.

I also have several other questions listed below. However, I underline that I was not able to interpret some important results so in the future I might have additional major comments and questions about the tree inference and the other parts of the manuscript.

How was the following conclusion derived "CCR in the primary tumor under therapy was thus associated with the emergence of a more diverse clonal landscape."?

Assuming that all cells in a tumor originate from the same founder cell, then all cells are having some shared ancestry. So the term "ancestral relationship" (line 78) is vague and needs to be precisely defined. It is currently not clear what does it mean that "strikingly similar ancestral relationship between subpopulations" were observed.

In line 126 parallel evolution in RAS/MAPK is mentioned and Figure 2B referenced. Where is this evolution shown in this figure?

In Figure 2b, there are six branches stemming from STEM node. Do these correspond to 6 mice treated with cisplatin and each represents evolution of the corresponding mouse? If true, then two of the six branches (branches having mutations in CCNI2 and EXT1 in the mother clones) are showing non-linear pattern of tumor evolution. Overall, I do not clearly see how the data supports claims of linear evolution when there are multiple co-evolving branches in cisplati-progressive cases Cisp4 and Cisp7.

Caption in Figure 2c mentions samples A and B. What are these samples and are they mentioned in the main text? I do not see them mentioned in the section starting at line 117. Please describe these clearly in the main text.

The coloring scheme employed makes it sometimes very difficult to distinguish between some populations, especially for populations from the samples that are colored in dark nuances of distinct colors. This would be especially challenging when reading a printed copy of the paper.

In the main text, it is stated that Patient 12 had a single small tumor in the neck and was treated with up-front surgery only. Then in Figure 1 (a)-(d) phylogenetic tree for this patient is shown, which includes MR samples. Please clarify this.

Is there a misspelling in "replicates" and "shows" in the sentence "Chemotherapy treated xenografts and cell culture models replicates these two contrasting scenarios and shows branching evolution to be a constant feature of proliferating NB cells." ?

Please revisit the first sentence in Figure 2a caption (form + formed).

Line 128, insert "Fig." in the reference.

Line 137, direction -> directions ?

RESPONSE TO REVIEWERS

Early evolutionary branching across spatial domains predisposes to clonal replacement under chemotherapy in neuroblastoma

In the following text, reviewer comments are copied verbatim in italics (bold type added by us) while our response and revisions are in plain text.

POINT-BY-POINT REVIEWER RESPONSE

Reviewer #1 (Remarks to the Author): Expert in brain cancers and preclinical models

"The manuscript Early evolutionary branching across spatial domains predisposes to clonal replacement under chemotherapy in neuroblastoma by Karlsson et al. explores the temporospatial heterogeneity and clonal evolution of primary neuroblastoma (NB) through therapy, using clinical specimens and patient-derived xenograft (PDX) in vivo modeling in combination with precision in silico modeling of clonal phylogenies. They show that tumors responding to treatment comprise a unique clonal landscape at relapse, whereas those that do not respond possess ancestral subclones detectable at diagnosis.

The authors do an excellent job of addressing alternative hypotheses that might explain their data, and they refute these hypotheses where possible, while providing adequate explanation where not (i.e., effective spatial sampling of in vivo PDX models pre-treatment when the xenografts are not very large). Previous studies have compared primary and relapsed NB to identify drivers of relapse, but the dynamic changes of specific subclones with respect to treatment refractory or treatment-sensitive subpopulations through therapy have not been well understood. The manuscript is also generally well-written, and the figures are extremely well done. Hence, this is an important study that will be broadly appreciated by the cancer research readership.

The following points should be addressed by the authors.

Comments:

• The study is underpowered, and the authors could comment on how a broader sampling of patient tumors through therapy might change the findings. For example, line 106-109 draw conclusions from single tumors. This could be described in a limitations/future directions section."

Response: The reviewer raised a relevant point.

Response: This has now been elaborated on in the Discussion (p. 17) where we also take the opportunity to highlight that future studies are needed to validate our suggestion on the heritability of the capacity to transit from an adrenergic to a mesenchymal phenotype.

"• It is difficult to discern whether "low dose" or "high dose" cisplatin are effective representations of stable disease through treatment vs treatment sensitive. What effect does low dose have on the cell population? Are cells dying? Preserved clonality could simply reflect a lack of cytotoxicity at the dose used in vitro."

Response: We had written in the manuscript on p. 9 that, "high-dose (1.0 μ M) cisplatin exposure with a 94% reduction in population size, ..." (i.e. cell death). Also in online methods (paragraph 2, samples B1-B3) it is described that low dose cisplatin initially suppressed cell growth, but the cells started to expand at the end of the experiment under exposure to cisplatin, but in contrast, the high-dose experiment reduced the number of cells with 94% compared to untreated control cells (paragraph 3, samples C1-C3).

Revision: We have added a sentence on p.9 in the manuscript explaining the lack of a long-time effects on cell proliferation of low dose cisplatin. We hope this highlights the differences between the two treatments.

"• Can the authors better explain/cite the tumor organoid model, line 263? It appears quite abruptly."

Response: The reviewer raises a good point.

Revision: We have now explained in Results that this is a free-floating three-dimensional tumor organoid system (p. 14) and also provided a reference to a recently published paper where the system is described (Mañas et al. Sci Adv 8:43 2022).

"• How are the authors inferring the mesenchymal state from their findings? Is it because of resistance to chemotherapy? It is unclear from the cited literature whether the chemotherapy resistant state in the present

study is fueled by mesenchymal characteristics. Perhaps the authors could use their organoid model to explore how mesenchymal characteristics are related to resistance to chemotherapy in this model.”

Response: We do not infer the mesenchymal state in our manuscript, we only mention it in the Discussion to put our findings in context of recent findings on the biology behind treatment resistance in neuroblastoma (see references in Discussion). The aforementioned PDX and animal models were recently used in a paper (co-authored by us) confirming the importance of transcriptional cell states for chemotherapy resistance also in model systems for neuroblastoma (Mañas et al. Sci Adv 8:43 2022). As response to the reviewer’s remark, we have now clarified this connection in Discussion.

Revision: we have clarified in Discussion (p. 17) that there is a now a published paper establishing the link that the reviewer requested (that paper was still in review when the present paper was submitted).

”• Karlsson et al., discuss that their study has implications for how to sample NB tumors from genomic profiling based on the evolutionary branching that is set early on in NB pathogenesis. Information regarding the feasibility and limitations should be included in the discussion. Also, the authors should discuss liquid biopsies as an alternative to multiple sampling to capture tumor heterogeneity. For example, the following study characterises pediatric medulloblastoma tumors by analysing the cerebrospinal fluid circulating tumor DNA revealing intra-tumour genomic heterogeneity (See DOI: 10.1038/s41467-020-19175-0)”

Response: The reviewer raises a valid point and we have now pointed out the challenges with multiregional sampling post chemotherapy while indicating that liquid biopsy could be a viable alternative.

Revision: Text has been added to Discussion (p. 18) regarding the practical limitations of multiregional sampling while citing two early studies (van Roy et al. Clin Cancer Res 2017; Chicard et al. Clin Cancer Res 2018) showing the feasibility of plasma cfDNA WES and low-pass WGS, respectively, to monitor clonal dynamics in neuroblastoma. However, we also add a sentence stating that the clinical value of these methods remains an open question: sensitivity seems to remain a problem in low-stage disease and since the two major studies referenced, there have not been any more publications on larger cohorts of neuroblastoma patients followed by liquid biopsy protocols.

Minor comment:

”• Ensure all abbreviations are defined, for example MNA NB line 178

• Consider minor typos in the text (examples below)

o Abstract, line 20 “...due to its propensity to resist treatment”

o Line 40 “...there are now a broad range...”

• Abstract, line 23 “...under effective treatment”. Effective isn’t a clear term here. Consider using terms like treatment refractory, treatment sensitive etc.

• Line 243 “figure 6” should specify which panel”

Revision: All minor comments above are addressed, however we decided to keep “there **is** a broad range”. We also kept “effective treatment” since we couldn’t find any better term to describe what we want to convey, ie a description of if the tumor responded to treatment in a certain moment, meaning that the treatment gave a certain **effect**. Using treatment sensitive is difficult in our context since this implies that the tumor has certain constant properties, while it is in fact often some change in treatment response over time; we tried using “sensitive” in a previous version of the paper it created confusion among the co-authors. We have added “Fig 6 **d, e, and j**”

”• Supplemental figures are very long, can a summary/table be provided?”

Response: Yes, we do agree with the Reviewer, the supplement is very long. We are not sure how to interpret the comment but have tried our best to improve the manuscript.

Revision: A summary of the major findings from the clinical neuroblastoma cases has been added to Figure 1 (new panel, Fig. 1p) and we have also added much more details to “Table of content” on the first page of Supplementary data legends”.

Reviewer #2 (Remarks to the Author): Expert in neuroblastoma genomics, therapy and clinical research

"In this well written manuscript Karlsson and colleagues from the Gisselsson group study the clonal evolution of neuroblastomas under the selective pressure of chemotherapy, an area which remains poorly understood. They study multi-region samples at diagnosis, post therapy and relapse from 12 patients with neuroblastomas, most but not all, with high-risk disease. They use PDXs models and a cell line to validate their findings. They then apply single cell DNA sequencing to map subclone geographies before and after treatment. The authors conclude that patients with progressive disease harbor de novo mutations that confer up-front resistance to chemotherapy, resulting in predominant resistant clone that evolve in a linear fashion. On the other hand, tumors from patients responding to chemotherapy undergo collateral clonal replacement – a process by which the founding mutations are replaced by new mutations.

The work is timely and could provide an important and clinically relevant observation to the literature. However, while there are important data in this manuscript, there are several concerns that should be addressed to merit publication in Nature Communications.

Major concerns

• The tumors have been collected over a period of 20 years and are apparently an admixture of frozen and formalin-fixed paraffin embedded samples. There is little discussion of data quality, and the data are not available for review (not is it mentioned where it will be made available post-publication). Copy number analysis in particular is sensitive to noisy data. This needs to be addressed and data made available to the reviewers."

Response: We thank the Reviewer for addressing this important point. Our response can be divided according to the two issues raised:

(1) Sharing genomic raw data: We have done the following to accommodate the reviewer's concerns

- A.** Zenodo archive deposition: Zenodo is an open-access repository operated by CERN (European Organization for Nuclear Research). As described in detail below we have put raw data plots from all SNP array profiles here to openly show the quality of the data. A link to this data is provided in the revised version of the paper.
- B.** We are currently transferring raw data to the EGA repository. Not that downloading this data will require a specific ethics permission, high requirements for documentation of information security, and has to be granted by a Lund University Data Access Committee.

(2) Sharing for quality control of data: When planning the study, we were of course aware of the problems associated with analyzing FFPE extracted DNA and we chose techniques and approaches thoroughly to ensure robust data. For SNP array analyzes we used the Oncoscan SNP-array platform, a technique especially constructed to analyze FFPE extracted DNA, and for WES analyses we confirmed our findings with targeted re-sequencing using the Twist-technology recommended for FFPE-extracted DNA. In addition, we used more stringent cut-off values for inclusion in the study when analyzing FFPE-extracted DNA compared to DNA extracted from cell lines or PDX- samples.

Revision: To enable evaluation of the quality of the SNP-array data we have uploaded TAPS plots for all samples analyzed to Zenodo. <https://doi.org/10.5281/zenodo.8017767>

TAPS was used in the manuscript to evaluate allelic composition, and there is a reference to the method in the manuscript. In the TAPS overview plots uploaded to Zenodo, exemplified by a sample from Patient_1 below, the SNP data is illustrated in two different plots (next page):

On top, all segments are visualized with log2 ratio on the x-axis and allelic imbalance on the y-axis with one plot per chromosome and below is a whole genome view illustrating both the log ratio and the allelic frequencies. We believe that these plot types efficiently and reliably reflect the quality of the data.

” Variant calling, especially in aneuploid tumors can be challenging and results can have high false positive rates. To fully assess the findings a set of orthogonal variant callers should be used.”

Response: We did use **three different variant callers** in the manuscript as stated in the paragraph “Whole exome sequencing and targeted sequencing of clinical samples”, but the rationale for doing so was actually to make sure we picked up all variants present. As described above, we created a custom-made panel based on another technology (Twist) that was used for targeted re-sequencing that enabled us to exclude false positives. The variants identified with both WES and targeted re-sequencing that passed the strict filtering steps described in the Methods section, were also visually inspected in IGV where the appearance of the sequences from the tumor samples were compared with the sequences obtained from the related normal samples.

Revisions: None as different variant callers were already used, and additional steps were taken to eliminate false positives

” Of the mutations called in the post therapy samples that undergo collateral clonal replacement (CCR) – it is not clear whether these mutations are driver mutations conferring survival advantages or if these are passenger mutations, and this should be clarified. ”

Response: This is a very interesting comment. It is not obvious how to distinguish between driver and passenger mutations in a tumor mostly driven by copy number aberrations, but if we focus on exonic mutations found in genes commonly mutated specifically in neuroblastoma but also in other tumors, we find few driver candidates globally in our cohort. We have identified some specifically interesting genes that could potentially act as drivers, and those are featured in the phylogenetic trees in Figure 1 and Figure S1.

Revision: None as the requested information was already provided.

” Figure 1: Since the author conclude that there are two patterns of clonal evolution: linear and collateral, it would be useful to have a panel summarizing the main conclusion of the manuscript, according to which densely packed territories of closely related subclones present at diagnosis are replaced under effective treatment by islands of distantly related survivor subclones, by a mechanism of CCR; while in tumors that progressed under treatment, ancestors of subclones dominating later in disease are present already at diagnosis. As currently structured, Figure 1 presents the phylogenies of individual patients and does not convey the main message in a direct graphical manner. Rather, it forces the readers to look at the description of each individual patient to

understand the timeline of his/her tumor samples. Consider including an Oncoprint or other summarization of the overall data.”

Response: Thank you for a very good comment!

Revision: We have now added a heatmap where we have summarized patient data, sample type and the main findings. It is added to Figure 1 (panel Fig. 1p).

”• Figure 2/3: the experiments shown in this figure attempt to model refractory disease. The authors use a cell line derived xenograft (not a PDX) treated with cisplatin and observe linear evolution. The authors use the exact same model in Figure 3, but with multiagent chemotherapy COJEC. Thus, the same CDX responds to an intensive therapy but does not respond to single agent therapy. To make a fair comparison to what is seen in the clinic, the authors should use a CDX or PDX from a refractory tumor (such as one harboring P53 mutation) and treat the PDX with COJEC instead of cisplatin as all patients receive the same induction therapy. Currently, this reviewer does not see what Figure 2 is adding to the overall story.”

Response: The PDX used in the previous version was in fact directly derived from a patient but had a short intermediate period in cell culture prior to injection into mice for a new set of PDXs.

The reviewer has a very valid point and we have now added data from another PDX system (PDX1) resistant to COJEC treatment, allowed to progress under COJEC treatment in the same type of mice as the previous data from PDX3 (sensitive to COJEC). These new data show the same evolutionary pattern of linear radiation from the founder clones (cell culture) to PDXs that grew under COJEC treatment.

We finally want to point out that it is not so much the treatment modality as the tumor’s history of progression per se that we argue determine the pattern of linear evolution. In fact, Patient 12, only received surgical treatment (was never exposed to chemo) but also exhibited linear evolution. Also several untreated PDXs exhibited this evolutionary pattern (PDX3 C5, PDX1 C1, C2 and C3).

Revision: New data have been added in Results (p. 7) and to Fig. 2 with substantial revision of the latter to increase its information value. More detailed data from the new analyses are also added to Supplementary Fig 2. Legends to Figures and Supplementary Data have been extended to incorporate the new data. We have also now clarified in the Discussion (p.15) that it is progression per se/immediate lack of chemotherapy response, not the treatment modality under progression (if any) that determines the pattern.

”• Figure 3: the results in this figure suggest that mutations in PTPRD and TCF4 drive resistance in relapse, but it is not clear if this is a reproducible finding or if these are “one-offs” in this model. This needs to be clarified. Do the genes in panel f-g appear in both mice that relapsed or only one?”

Response: These are valid questions and the manuscript has been revised accordingly

Revision: We have added a sentence in Results (p. 9) clarifying that these deletions are not “one-offs” but in fact recurrent features of neuroblastoma PDX systems under continuous growth in vivo, while not having a direct link to resistance and relapse. We have also cited a recent paper that clarified this in detail in the same PDX system. The legend to Fig 3f-g has been revised to clarify that the “genes” (i.e. mutations) occurred in both mice/PDX tumors.

• Figure 4: The authors attempt to show CCR in vitro. The results should be validated in more than one cell line.

Response: We have made a new experiment with the aim to explore changes in the clonal landscape in the neuroblastoma cell line SK-N-SH after induction of cell death with cisplatin. We chose it specifically to investigate our finding, from the previous version, that sensitive-resistant state transition is to some degree dependent on genetic lineage, albeit not usually on specific mutations.

The cell line encompasses cells that can transiently differentiate between two cell states, S-type (substrate adherent) and N-type (neuroblastic). The bigger flat S-type cells are known from previous studies to correspond to a mesenchymal more drug resistant phenotype, while the neuronal cells correspond to adrenergic chemo sensitive cells. Cell cultures (n=5) were treated with 4 μ M cisplatin at two succeeding time points for 3 days each. Samples were also taken in parallel from untreated cells (n=3) to be used as control and from the cell culture (n=2) at the start of the experiment.

In the original cell culture, we detected a subclone at 80% containing two nested parallel daughter clones at 30% each. This landscape was largely stable over time in untreated controls until a novel subclone at 30%, was detected in the control at the end of the experiment. Cells with only stem aberrations were present in all mother and control cultures. After cisplatin induced death (80%) and regrowth, there was clear shift in the clonal landscape, uniformly affecting all treated cultures, where the clone having only stem aberrations uniformly disappeared/was replaced by a subclone that underwent a clonal sweep, while a nested subclone increased to 70-80%; at the same time, one of the largest subclones in the mother cultures, decreased down towards detection limits (10%).

These new data support our findings that effective chemotherapy treatment is associated with a change in the genetic landscape, even in a model system where there is a known phenotypic state-transition taking place. The changes had clear features of CCR, as some clones expanded and others diminished at treatment in a pattern reciprocal to control cultures. However, we note that there were no novel clones appearing after treatment, making CCR incomplete. This contrasts to the scenario we observed in chemo-sensitive tumors, PDXs and the IMR-32 cell line and is probably explained by the fact that there is a rather large pre-existing population of resistant cells in SK-N-SH that rapidly overtook the cultures. Nevertheless, we sought it most scientifically prudent to perform additional experiments in a cell line that was prone to state transition as it would challenge our hypothesis – instead of cherry picking a cell line, similar to IMR-32, that did not show this capacity. Should the reviewer disagree with our choice we could make additional experiments in a more drug-sensitive cell line, but this will require substantial time. Also, we would like to stress that CCR is a uniform finding in the clinical cohort and it is questionable how much validation in vitro systems actually provide in that context.

Revision: The experiment and the novel data are described in the Result section (p 10-11), the supplementary figures S4 f-g, in the online method section, and the aberrated segments are demonstrated in Table S2 I.

”• Figure 6: The findings are interesting. However, it is not clear how these results relate to the entire story of clonal evolution under chemotherapy resistance.”

These results in fact explain why we see patterns of CCR after therapy. Based on finding CCR by analysis of bulk samples obtained before and after therapy it was obvious that NBs had developed highly branched phylogenies after treatment. However, there could be two alternative explanations to this:

- (1) Prior to therapy branching emerges only from the most common recent ancestor to the clones detected at that stage. Under treatment, as these clones are killed, new branches sprout from the surviving ancestor
- (2) Extensive branching is present already at an early stage, prior to treatment, meaning the major clones observed after treatment emerge from close ancestors already there.

We have attempted to outline these two competing scenarios in the figure below:

The data in Figure 6 show that it is the second alternative (early branching) which best explains CCR, as single cell analysis could trace branching back to the putatively earliest mutations occurring in neuroblastoma, i.e. *MYCN* gain. As we already have a large number of figures, and the other two reviewers did not raise this point, we have currently tried to better explain in the text why the data in Figure 6 provides a critical explanation to CCR.

Revision: Two explanatory sentences have been added in Results (p. 12) to explain both the *MYCN* amplicon diversity analysis on bulk samples and the single cell analyses in Figure 6. If the reviewer or editor prefers us to also include the figure above as an additional supplement, we will be happy to do so in the next revision.

Minor comments

• *For the two patients who progressed under therapy, we suspect that this is due to *MDM2* amplification in patient 1 and *NF1* loss in patient 2. However, this is not discussed in the text and it is unclear what is the author's hypothesis as to why these patients progressed.*

Response: this is a good point and we have revised accordingly, highlighting with appropriate references the clinical significance of *MDM2* amplification and *NF1* loss, respectively.

Revision: A sentence highlighting the fact that Patients 1 and 2 had these high-risk anomalies, has now been added to the description of these patients in Results (p. 4).

• *The authors use ratios of surviving cells in the tumor to document anti-tumor response. While valid, it would be good to know the overall end induction response to chemotherapy for each subject.*

Response: this is a good point and we have revised accordingly; still it should be noted that end volumes are poor proxies of response as neuroblastomas often respond to treatment with maturation and fibrosis, something that does not necessarily result in shrinkage of the tumor volume. We therefore used histopathology to measure response in the present paper.

Revision: end induction response classification according to the revised (2017) International Neuroblastoma Response Criteria (INRC; Park et al. (*J Clin Oncol.* 2017;35(22):2580–2587) have been added in Supplementary Table 1A

• *The rationale for including low- and intermediate-risk patients who do not convert to high-risk needs to be carefully stated and how including these subjects in the overall analysis explained.*

Response: The benefit of including these patients was to see whether also less aggressive disease, treated with a milder protocol than COJEC, showed the same evolutionary reaction pattern to treatment as COJEC-treated high-risk patients.

Revision: We have now clarified the rationale for including these patients in Discussion (p. 16).

• *You previously mention that Patient 12 was treated with up-front surgery only, but here you say that Patients 4 and 12 were the ones with least chemotherapy-induced regression. Please clarify.*

Response: We apologize for being unclear. In fact, there were two unclaritys here that we have now addressed.

Revisions:

- (1) The sentence mentioned (Results, p. 6) has been changed to “treatment-induced regression” not to risk interpretation that this patient was chemotherapy-exposed before relapse – even though this was in fact clarified in a subsequent sentence in the previous version of the paper
- (2) In Supplementary Table 1A, we have now imposed a system of footnotes to clarify that in Patient 4 there was merely 20% necrosis while there was 60% ganglioneuroma-like maturation

• *Line 140. Please specify the meaning of “same” as in “same PDX”.*

Revision: This sentence is re-written, and “same PDX” is removed.

• *142. Please define the acronym “COJEC” when first reported in the manuscript.*

Revision: This has now been done (p. 7).

"• 161-163. The fact that the finding of CCR under therapy could not be validated in the CDX/PDX model is a major limitation of the study. Please comment on this limitation in the manuscript."

Response: We indeed wrote in p. 8 that "Because PDX sizes were feared to be too small for sampling prior to treatment without inducing populations bottlenecks, the finding of CCR under therapy from the clinical cohort could not be validated by sampling PDXs before and after therapy." Here we meant to stress that we could not use **exactly the same sampling setup** as in the patient material due to the small size of PDXs (larger ones would violate the ethics permit). However, we do see the same CCR pattern in the PDX system (Figure 3) as we did in the clinical situation when a chemo- and surgery treated tumor relapses, **validating CCR in that situation**. Finally, we want to note that comparison of COJEC vs. saline treated PDXs allowed us to validate increased subclone diversity emerging with treatment (Fig. 3d).

Revision: To avoid further misunderstanding we have now commented, as requested by the referee, in the Discussion that we were not able to show CCR in PDXs using exactly the same setup (p. 16) of pre-post sampling as in patients but that we were able to validate the CCR seen when metastases appeared after treatment, as was described in p. 5. Here we write "CCR was also observed for the majority of subclones in the primary tumor when compared to those in a later metastatic relapse (n=5; Patients 4, 9-12)." Also, to avoid confusion the title of the Results section describing PDX COJEC results has been changed to "*Subclone replacement at relapse after effective chemotherapy in patient-derived xenografts*".

"• 171-172. How can you conclude that the PDX model supports the association between effective chemotherapy response and CCR if the pre-treatment sampling and genetic analysis of PDX was not performed? Please explain."

Response: As discussed in the point above, we were not able to analyze the samples before treatment since they were too small, and we were worried that sampling would induce a bottle neck. Instead, we analyzed the correlation between the samples post treatment and samples taken from the PDXes that relapsed locally. This is now explained in the manuscript p. 8 and 16.

Revision: See the point immediately above.

"• 178. Please define the acronym "MNA" when first reported in the manuscript."

Revision: This has now been done (p. 6).

"• 238-266. In this paragraph you derive your conclusions mostly from the findings in 3 MNA tumors. Please comment on the findings in the other six non-MNA tumors."

Response: We focused in the text on MNA tumors as MYCN gain is a bona fide very early event in neuroblastoma pathogenesis. However, also non-MNA tumors showed extensive branching after the stem events, and this information is already stated on p. 13.

Revision: none

• 278. Please define "numerically low" when first reported in the manuscript.

Response: We can't find "numerically low" in the manuscript, but we have written; "...numerical only" low intermediate risk NB .." twice p.14 and p.15.

Revision: none

"• 329-331. How can you conclude that "in this context, our data suggest that different genetic lineages present in the same patient have different potential for the mesenchymal, chemoresistant state" without analyzing any mesenchymal signature gene? If you have done so, please specify in the manuscript."

Response: Please see our response to this issue when raised by Reviewer 1. In summary, we do not make any strong statements about this but only try to put our data in the context of the very lively debate on this issue.

Revision: We have substantially revised the Discussion text in p.17, putting more context to our arguments by explaining the link between our data and a recent PDX study of the shift between mesenchymal and adrenergic states under COJEC treatment.

"• 389. Here you say that one of the spatiotemporal comparison was "pre vs MR", while the figure 1g says "pre/post vs MR". Please explain the discrepancy."

Response: It should be primary instead of pre/post vs MR just as we had written in the text prior to the parenthesis.

Revision: We have changed pre/post in figure 1g to Primary and made changes accordingly in the figure legend.

"• 405. Why does the IGD was calculated on 10 phylogenetic trees only, instead of 11 (12 total patients minus Patient 12 who had up-front surgery). Please explain."

Response: We lack WES-data for two of the patients, hence we only have 10 phylogenetic trees based on both CNA and WES analyses. IGD was calculated on these 10 trees. This information is written in the legend to fig 1g ("IGD was calculated on the ten phylogenetic trees available based on combined CNA and SNV/indel data").

Revision: none

"• 415. Please specify how many "yes" and how many "no"."

Response: There are four "yes" and six "no".

Revision: This information has been added to the legend to fig 1-l.

"• 426. Here you say that the PDXs where all MYCN-amplified while in the main text (line 121) you solely mention Stage III NB. Please explain the discrepancy."

Response: We apologize for this confusion!

Revision: We have replaced "Stage III" with "MYCN-amplified".

• 496. Where is the letter "l" in figure 5?

Response: It is down to the left in the figure above "Patient 5".

Revision: none

• 530. I believe "BPC" is misspelled here for "PBC". Please fix accordingly.

Response: BPC is an acronym for breakpoint cluster. This is stated in the legend to Fig 6.

Revision: none

Reviewer #3 (Remarks to the Author): Expert in cancer evolution, phylogenetics, and single-cell genomics

"In this work, Karlsson et al. study clonal evolution of Neuroblastoma under various conditions such is treated vs. non-treated, as well as across time and space.

Unfortunately I had a great difficulty deciphering large number of figures presented in this work, especially the phylogenetic trees. Overall, the results do not seem impressive and the analyzed cohort is of a very limited size to derive conclusions with broader implications. I am also reserved about the correctness of the provided trees as relatively low number of bulk-sequenced samples per patient yields limited signal and power to unambiguously resolve phylogenetic trees. "

Response: We agree that the study is limited in power, which has now been commented on in the Discussion according to recommendations from another reviewer. However, it should be noted that obtaining such a rich material over time and space in neuroblastoma is exceedingly rare, especially since identification and sampling the few remaining islands of neuroblastoma cells typically left after chemotherapy is painstaking histopathological and technical work. The fact that this was successfully done made us capable of monitoring shifts in the genetic landscape over treatment time, albeit we admit the power is still limited.

Revision: Limitations have been elaborated on in Discussion (p. 18) according to recommendations from another reviewer.

"The Authors are referring to their previous tool, DEVOLUTION, that was used in this work to find most of the presented trees. In order to better understand this work, I tried to read DEVOLUTION paper but I faced some challenges that I could not resolve. For example, in that paper it is stated that phylogenetic tree of patient NB5 shown in Figure 2a has polyclonal seeding, with one seeding clone originating from a subclone harboring stem events only and one seeding clone sharing the same copy number profile as Subclone A (this is stated in the caption of Figure 2). If Subclone A does not contain any additional somatic mutations or aberrations compared to the branching point at the right end of the edge labeled with "+19 ++22", then the two other subclones in the metastatic sample, which are E and C, are having the same profile as A, which contradicts the above statement. Therefore I conclude that A has some additional events compared to this branching point. But then the variants specific (private) to A appear at frequency approx. 80% (based on the pie chart), whereas variants specific to Subclone C are at approx 50%, which adds up to 130% for two subclones on separate branches, violating Pigeonhole principle (see 10.1101/cshperspect.a026625).

In summary, I find this way of depicting phylogenetic history and clonal and subclonal frequencies not just confusing, but also very difficult to interpret. Could the Authors please provide broadly adopted "clonal trees" where proportions of subclones in different samples are clearly indicated instead of giving frequencies of variants specific to clones?

We are primarily interested in clonal dynamics and seeing which subclones die out and which ones emerge and/or take over. It would be much simpler and easier to interpret figure if this information is directly given, instead of having to derive it by subtracting children from parent frequencies.

A solution that I recommend is to provide figures that are combination of what is shown in panels d and e in the figure shown at <https://github.com/hdng/clonevol>. For example, a sample S can be associated with, say, 100 small circles of the same color (the number can be less than 100 but should be the same for all samples). If it is estimated that 25% of cells from S belong to subclone 4, then subclone 4 (that is, tree node 4) can be assigned 25 dots of the color of S (better contrasts between colors or some other way to better distinguish samples will be needed - see below). Mutations can be assigned along the edges of the clonal tree and potential metastatic seeding events can be clearly marked by pointing to the edges along which migration happened."

Response: We note that the reviewer chooses to comment on our previously peer-reviewed and published methods paper here. The criticism evidently arises from our way of annotating clone sizes as pie charts whose sizes correspond directly to the genetic markers for a certain subclone. The subclone to be "fit" into the "pigeonhole" thus do not immediately correspond to the pie sizes but can easily be derived from comparing pies following each other in subclone lineages nested in each other as per the figure below:

However, we realize that this way of annotating clone size may appear confusing, and we thank the Reviewer for a very good suggestion! Therefore, we have now made a variant of the recommended plot type for the revised manuscript. We believe these plots add a new valuable layer of information, since they convey the clone sizes in an illustrative way.

Revision: We have made new plots accompanying all phylogenetic trees based on the clinical cases (Fig S1). Each sample were symbolized by a circle containing 20 small circles, where each small circle represents a clone size of 5%. The clone sizes were calculated by subtracting the sizes of a pie chart representing a downstream population from an upstream population (child from parent) as described in the figure above. We have also clarified in the legends to Fig 1 and supplementary Fig 1 that the Venn diagrams/pie charts illustrate the size of a population of cells that share the same mutation or set of mutations instead of clone sizes as previously stated.

"I also have several other questions listed below. However, I underline that I was not able to interpret some

important results so in the future I might have additional major comments and questions about the tree inference and the other parts of the manuscript."

Response: We have addressed all questions listed below and hope that this, in conjunction with amendments as specified above regarding tree construction, will have alleviated the need for additional major comments.

"How was the following conclusion derived "CCR in the primary tumor under therapy was thus associated with the emergence of a more diverse clonal landscape."?"

Response: This rests on the finding detailed in the previous sentences detailing a higher index of genomic diversity (IGD) in the treated primary tumors compared to the corresponding diagnostic samples. To avoid further lack of clarity, the text has been revised.

Revision: We have rephrased the text in Results (p. 5) to clearly state that the increase of diversity after treatment rests on the IGD measurements and also moved the point of reference to the figure panel (Fig. 1j) where IGD is shown.

"Assuming that all cells in a tumor originate from the same founder cell, then all cells are having some shared ancestry. So the term "ancestral relationship" (line 78) is vague and needs to be precisely defined. It is currently not clear what does it mean that "strikingly similar ancestral relationship between subpopulations" were observed."

Response: We agree with the reviewer that the term "ancestral" has been used in a confusing manner in the referred passage, as we were really referring to the overall phylogenetic relationship.

Revision: the term "ancestral relationship" has been replaced with "phylogenetic relationship".

"In line 126 parallel evolution in RAS/MAPK is mentioned and Figure 2B referenced. Where is this evolution shown in this figure?"

Response: We thank the Reviewer for pointing out that we forgot to specify which genes we were referring to.

Revision: We have now added "BRAF and MAP3K4" in the legend to figure 2d.

"In Figure 2b, there are six branches stemming from STEM node. Do these correspond to 6 mice treated with cisplatin and each represents evolution of the corresponding mouse? If true, then two of the six branches (branches having mutations in CCN12 and EXT1 in the mother clones) are showing non-linear pattern of tumor evolution. Overall, I do not clearly see how the data supports claims of linear evolution when there are multiple co-evolving branches in cisplatin-progressive cases Cisp4 and Cisp7. "

Response: The six branches originating from the STEM node are illustrating the mother clones. This is explained in the legend to Figure 2b (Figure 2d in the revised version). We drew our conclusions from the linear radiation originating from the stem population via the mother clones. This pattern is described in the newly added text on p 7.

Revision: Newly added text on p 7.

"Caption in Figure 2c mentions samples A and B. What are these samples and are they mentioned in the main text? I do not see them mentioned in the section starting at line 117. Please describe these clearly in the main text."

Response: We took two samples from each tumor (sample A and sample B). This was described in the Methods section, and in the figure legend before and is now also added in the main text at p. 6 to avoid confusion.

Revision: Added explanation in the main text at p. 6 to avoid confusion.

"The coloring scheme employed makes it sometimes very difficult to distinguish between some populations, especially for populations from the samples that are colored in dark nuances of distinct colors. This would be especially challenging when reading a printed copy of the paper. "

Response: We have been aware of this problem, and we have tried to pick the colors cleverly, but in some cases the number of clones or what we need to illustrate are just many more than the colors available. We have been careful to pick illustrative colors when it's extra important to be able to track specific samples for example pre vs post.

Revision: We have been careful in choosing clear color schemes when extensively revising Figure S1 (the new subclone charts), where we think the major problem arises.

"In the main text, it is stated that Patient 12 had a single small tumor in the neck and was treated with up-front surgery only. Then in Figure 1 (a)-(d) phylogenetic tree for this patient is shown, which includes MR samples. Please clarify this. "

Response: MR stands for metastatic relapse which appeared after the treatment. This is written in the legend to figure 1.

Revision: To avoid confusion we have re-written the referred sentence in the main text and included that the patient later developed a metastatic relapse.

"Is there a misspelling in "replicates" and "shows" in the sentence "Chemotherapy treated xenografts and cell culture models replicates these two contrasting scenarios and shows branching evolution to be a constant feature of proliferating NB cells." ?"

Response: Thank you for noticing this.

Revision: We have corrected the sentence.

"Please revisit the first sentence in Figure 2a caption (form + formed)."

Revision: This sentence is now re-written.

"Line 128, insert "Fig." in the reference."

Revision: This sentence is now re-written

"Line 137, direction -> directions ?"

Revision: We have changed to "directions".

REVIEWER COMMENTS

Reviewer #1 (Remarks to the Author):

The authors have addressed all the questions and minor points we raised regarding their manuscript “Early evolutionary branching across spatial domains predisposes to clonal replacement under chemotherapy in neuroblastoma”. We have no further questions.

Reviewer #3 (Remarks to the Author):

1. The Authors have not addressed my major comment about the phylogenetic trees representation, which I find essential to be able to easily interpret their phylogenetic analysis and results. I strongly insist on this being addressed as per the instructions provided in the first round of review. I agree that clonal frequencies might be inferred from the given figures, but in this case it is impractical.
2. It is not clear how one can reproduce the presented results or test whether the same results would be obtained using tools other than DEVOLUTION. Are all data and code required to reproduce results presented in this work available and, if yes, where can I find them?
3. I did not understand the observation starting on Line 107 and ending on Line 110. I assume that the authors are referring to Panels VII from each of the referenced figures, but I do not see how the observations follow from these phylogenies.

Minor:

The manuscript needs very thorough revision in presentation. I will illustrate this point just on Figure 1 where I found a number of ambiguities, typos, inconsistencies etc. Although they are predominantly minor, their total number by far exceeds what I usually observe when reviewing other submissions to this journal. Below is the list of my comments (all are related to Figure 1):

In panel f, blue, orange etc. circles are used without being defined (their meaning is provided only in later panels). Furthermore, the use of the labels P1, P2, P3 in this panel as well as in panel d, with different meanings, can be confusing.

What is the grey filled circle present in panel g (in Primary vs. MR) representing and why is its definition not provided?

In the legend given for Pre samples in lower-left part in panel c, please provide labels Pre1, Pre2 and Pre3. Also, what is the difference between Pre1 (used in the text in caption) and P1 (used in panel d)?

It is not clear what the dots in the plot shown in panel k represent.

In the caption, it is not clear what is shown in panel m. First, in Line 484 it is mentioned "m-o", but then in the next line (Line 485) only "n-o" is mentioned, leaving it unclear where m falls.

In the caption, Line 436, there is a linguistic error in "with information SNV/indels".

In the caption, Line 447, "of" is missing in "gain chromosome 15".

In the caption, Line 443, should "allow" be replaced with "allows" (or "comparison" in Line 442 replaced with "comparisons")?

In the caption, Line 447, defining abbreviations using "=" in "chromothripsis like = CHL" is non-standard and some readers can find it ambiguous.

In the caption, Line 456, "." missing after the first use of "vs". Similarly, "." missing at the end of sentence in Line 470.

In the caption, inconsistent use of "breakpoints" vs. "break points".

In the caption, Line 483, for an MR sample, "was present" or "was available" might be more appropriate than "occurred".

In the caption, Line 490, "An heatmap" -> "A heatmap".

In the caption, Line 496, "at same site" -> "at the same site".

Reviewer #4 (Remarks to the Author): Expert in neuroblastoma genomics and clinical research; replaces Reviewer #2

For the most part the authors have addressed my concerns and made improvements.

RESPONSE TO REVIEWERS

Early evolutionary branching across spatial domains predisposes to clonal replacement under chemotherapy in neuroblastoma

In the following text, reviewer comments are copied verbatim in italics (bold type added by us) while our response and revisions are in plain text.

POINT-BY-POINT REVIEWER RESPONSE

Reviewer #1 (Remarks to the Author):

"The authors have addressed all the questions and minor points we raised regarding their manuscript "Early evolutionary branching across spatial domains predisposes to clonal replacement under chemotherapy in neuroblastoma". We have no further questions. "

Response: We are grateful that the reviewer(s) find that the manuscript has improved and we note that there are no other recommendations for improvement.

Revision: None.

Reviewer #3 (Remarks to the Author):

1. The Authors have not addressed my major comment about the phylogenetic trees representation, which I find essential to be able to easily interpret their phylogenetic analysis and results. I strongly insist on this being addressed as per the instructions provided in the first round of review. I agree that clonal frequencies might be inferred from the given figures, but in this case it is impractical.

Response: We are surprised by this comment for three reasons

1. We did make a substantial changes in our previous revision according to our interpretation of the Reviewer's statement. Please see our response at previous revision, enclosed as an Addendum at the end of this letter. In summary, we used the suggested system of annotation (d in the github-link provided by the referee of the "cloneevol" software) in our revision of Figure S1 and our revised Figure 2. We found it difficult to use in Figure 1 as the suggested system requires considerable space to show details because our abundance of a samples for each time point. We considered using the suggested flow chart (e in the github link) but we believe it makes us lose too much information, in particular phylogenetic distances. The latter are important as they are discussed in the main text and also presented in summary in Fig. 1k. Furthermore, it would also be cluttered with information, again because of the many samples per time point in our study. However, to make further amendments in line with the reviewer comment we have been inspired by the cloneevol output to provide a simplified phylogenetic overview in Fig. 1 to make it easier to interpret the results as requested by the reviewer. Here we took particular care to visualize subclone and clone status and not mutation frequencies in line with the reviewer's request (see revisions below).
2. In our previous revision, we added a set of panels in Fig. S1, matching "d" in the github link in order to accommodate the Reviewer's criticism. The reviewer did not at all comment on the revised Fig. S1 and we are worried that our major revision of this according to her/his instructions were in fact not noticed.
3. Our way of subclone annotation or variants thereof has been used in several previous publications without similar criticism. Examples are Rastegar et al. Clin Cancer Res 2023 (29: 2668–2677), Andersson et al. Genes Chromosomes Cancer 2023 (62:93-100), and Commun Biol 2021 (4:1103), and Karlsson et al. Nat Genet 2018 (50:944-950). We also note that none of the other three reviewers have reported any issues with our annotation system. We would also like to again stress that the examples provided from "cloneevol" are from phylogenies based on relatively few samples, while our dataset is based on a very high number of samples per patient, making phylogenetic presentation inherently much more complex and warranting other types of figures for illustration.

Considering this, we interpret the Reviewer's criticism as a recommendation to use his/her suggested system also for Figure 1, as this figure is also pointed out below as seems to be the major focus of the review.

Revision: Figure 1 has again been substantially revised, and we have tried to accommodate the system of presentation from the github link of "cloneevol" by

- (1) Clarifying presentation of subclone composition in Fig. 1c-d, thereby providing a direct explanation to the further comprehensive “clonevol”-type presentation provided in Fig. S1a-l for all cases (panels at the bottom of each figure in Fig. S1).
- (2) Removing pie charts from Fig. 1 altogether as the reviewer thinks “clonal frequencies” are impractical. Instead we have opted for graphically simplified phylogenetic trees without clonal frequencies/subclone sizes but where we still retain all nodes and with phylogenetic distances preserved. As information on precise subclone sizes would be overwhelming in this figure, we have chosen to just present subclones as open circles and populations found to clonal in all detected locations as filled circles. The exact sizes of subclones detected in each sample are still presented in Fig. S1 just as in our prior revision, in line with the Reviewer’s recommendations.

2. It is not clear how one can reproduce the presented results or test whether the same results would be obtained using tools other than DEVOLUTION. Are all data and code required to reproduce results presented in this work available and, if yes, where can I find them?

Response: we apologize if we were unclear on this point. The requested information was available in “Online Methods” as per journal standard under the headings **Data Availability** and **Code Availability** (page 22). In summary, the following sources are given for independent reproduction of the data:

- **Table S2.** Curated raw data files on mutation frequencies. These can be used to test other phylogenetic tools than DEVOLUTION
- **The Zenodo archive:** here are deposited raw data plots from whole genotyping of clinical samples, PDX tumors and cell lines (<https://doi.org/10.5281/zenodo.8017767>). These are primarily deposited as a quality control to raw data curation/interpretation.
- **The European Genome-phenome Archive:** Raw data from whole genome genotyping and sequencing have been deposited in EGA (<https://ega-archive.org>) under accession number **EGAS00001007650**.
- **DEVOLUTION:** has been published in a peer-reviewed original article Andersson, N., Chattopadhyay, S., Valind, A., Karlsson, J. & Gisselsson, D. DEVOLUTION-A method for phylogenetic reconstruction of aneuploid cancers based on multiregional genotyping data. *Commun Biol* **4**, 1103 (2021). In this original paper are links to the software and its updates
- **SCEM**, the software used to construct phylogenies from single cell WGS is available online at <https://github.com/NatalieKAndersson/SCEM>.

Revision: Data is now uploaded to EGA. We have also now explicitly stated in Results (p. 3-4, lines 56-59) that information regarding raw data is given in the Online Methods to avoid further confusion.

3. I did not understand the observation starting on Line 107 and ending on Line 110. I assume that the authors are referring to Panels VII from each of the referenced figures, but I do not see how the observations follow from these phylogenies.

Response: In our copy, lines 107-110 contain the following text” Minor but significant exceptions from this were Patients 4 and 12. In Patient 4 (Fig. S1d), a regional population corresponding the common ancestor of relapsing subclones was detected in the primary tumor after therapy, while in Patient 12 (Fig. S1l) it was detected prior to therapy.” We apologise for not being sufficiently specific in our references to the Fig. S1. Panels S1d:VII and S1l:VIII were indeed the most important points of references, although the entire figures should be viewed for the context to be understood.

In both the above mentioned panels, it can be clearly seen that the relapse subclones are descendants from a population detected in the primary tumor, i.e. population A in S1d:VII and population H in S1l:VIII.

Revision: To clarify these relationships, the specific panels and subclones have now been pointed out in the revised manuscript text (page 6, lines 114-115).

Minor:

The manuscript needs very thorough revision in presentation. I will illustrate this point just on Figure 1 where I found a number of ambiguities, typos, inconsistencies etc. Although they are predominantly minor, their total number by far exceeds what I usually observe when reviewing other submissions to this journal. Below is the list of my comments (all are related to Figure 1):

Response: We apologize for this and thank the reviewer for pointing out points for improvement. We have

addressed the issues below and also done yet another round of correction on the language and format of the entire manuscript. See details below on each point, now corrected. We are very grateful to the reviewer for pointing out these points for improvement!

In panel f, blue, orange etc. circles are used without being defined (their meaning is provided only in later panels). Furthermore, the use of the labels P1, P2, P3 in this panel as well as in panel d, with different meanings, can be confusing.

Revision: the definitions have now been moved in adjacently to Fig. 1f to avoid further confusion.

What is the grey filled circle present in panel g (in Primary vs. MR) representing and why is its definition not provided?

Revision: the grey circle was meant to refer to primary either before or after treatment (as stated under the ideogram of the child). As this created confusion, we have replaced the grey circle with green and orange ones as to follow the definitions.

In the legend given for Pre samples in lower-left part in panel c, please provide labels Pre1, Pre2 and Pre3. Also, what is the difference between Pre1 (used in the text in caption) and P1 (used in panel d)?

Revision: we apologise for this, and have now inserted subclone maps of Pre1-3 and MR in c to clarify that it is three different samples and also changed P to Pre in d.

It is not clear what the dots in the plot shown in panel k represent.

Revision: It refers to phylogenetic tree branch length as stated on the y-axis. To clarify further, this has now been described in more detail also in the legend (p. 20, line 538).

In the caption, it is not clear what is shown in panel m. First, in Line 484 it is mentioned "m-o", but then in the next line (Line 485) only "n-o" is mentioned, leaving it unclear where m falls.

Revision: it has now been clarified in the legend (p. 21, lines 551-552) that m is metastatic relapse and n-o progression under treatment

In the caption, Line 436, there is a linguistic error in "with information SNV/indels".

Revision: now corrected to "information on SNV/indels" (p. 19, line 482)

In the caption, Line 447, "of" is missing in "gain chromosome 15".

Revision: now corrected as "gain of chromosome 15" (p. 19, line 484)

In the caption, Line 443, should "allow" be replaced with "allows" (or "comparison" in Line 442 replaced with "comparisons")?

Revision: "Comparison" has been changed to "Comparisons" (p. 19, line 490)

In the caption, Line 447, defining abbreviations using "=" in "chromothripsis like = CHL" is non-standard and some readers can find it ambiguous.

Revision: we agree it is a non-standard definition, and have attempted a further clarification by putting CHL in square brackets (p. 19, line 495)

In the caption, Line 456, "." missing after the first use of "vs".

Revision: This has been corrected (p. 20, line 519)

Similarly, "." missing at the end of sentence in Line 470.

Revision: This has been corrected (p. 20, line 531)

In the caption, inconsistent use of "breakpoints" vs. "break points".

Revision: This has been corrected (p. 20, line 528; p. 21, line 554)

In the caption, Line 483, for an MR sample, "was present" or "was available" might be more appropriate than "occurred".

Revision: "occurred" has been changed to "was analyzed" (p. 21, line 549)

In the caption, Line 490, "An heatmap" -> "A heatmap".

Revision: This has been corrected (p. 21, line 557)

In the caption, Line 496, "at same site" -> "at the same site".

Revision: This has been corrected (p. 21, line 563)

Based on the overall criticism regarding language and style, we have also thoroughly reviewed the text and made a number of corrections/edits, viewable in the enclosed "track-changes" manuscript file.

Reviewer #4 (Remarks to the Author): Expert in neuroblastoma genomics and clinical research; replaces Reviewer #2

For the most part the authors have addressed my concerns and made improvements.

Response: We are grateful that the reviewer(s) find that the manuscript has improved and we note that there are no other recommendations for improvement. This is especially rewarding to us as the previous Reviewer #2 raised a very high number of points for improvement, all of which we addressed in the first revision.

Revision: None.

Addendum: Previous Reviewer #3 criticism on phylogenetic trees and our response at first revision

Reviewer #3 (Remarks to the Author): Expert in cancer evolution, phylogenetics, and single-cell genomics

"In this work, Karlsson et al. study clonal evolution of Neuroblastoma under various conditions such as treated vs. non-treated, as well as across time and space.

Unfortunately I had a great difficulty deciphering large number of figures presented in this work, especially the phylogenetic trees. Overall, the results do not seem impressive and the analyzed cohort is of a very limited size to derive conclusions with broader implications. I am also reserved about the correctness of the provided trees as relatively low number of bulk-sequenced samples per patient yields limited signal and power to unambiguously resolve phylogenetic trees."

Response: We agree that the study is limited in power, which has now been commented on in the Discussion according to recommendations from another reviewer. However, it should be noted that obtaining such a rich material over time and space in neuroblastoma is exceedingly rare, especially since identification and sampling the few remaining islands of neuroblastoma cells typically left after chemotherapy is painstaking histopathological and technical work. The fact that this was successfully done made us capable of monitoring shifts in the genetic landscape over treatment time, albeit we admit the power is still limited.

Revision: Limitations have been elaborated on in Discussion (p. 18) according to recommendations from another reviewer.

"The Authors are referring to their previous tool, DEVOLUTION, that was used in this work to find most of the presented trees. In order to better understand this work, I tried to read DEVOLUTION paper but I faced some challenges that I could not resolve. For example, in that paper it is stated that phylogenetic tree of patient NB5 shown in Figure 2a has polyclonal seeding, with one seeding clone originating from a subclone harboring stem events only and one seeding clone sharing the same copy number profile as Subclone A (this is stated in the caption of Figure 2). If Subclone A does not contain any additional somatic mutations or aberrations compared to the branching point at the right end of the edge labeled with "+19 +22", then the two other subclones in the metastatic sample, which are E and C, are having the same profile as A, which contradicts the above statement. Therefore I conclude that A has some additional events compared to this branching point. But then the variants specific (private) to A appear at frequency approx. 80% (based on the pie chart), whereas variants specific to Subclone C are at approx 50%, which adds up to 130% for two subclones on separate branches, violating Pigeonhole principle (see 10.1101/cshperspect.a026625).

In summary, I find this way of depicting phylogenetic history and clonal and subclonal frequencies not just confusing, but also very difficult to interpret. Could the Authors please provide broadly adopted "clonal trees" where proportions of subclones in different samples are clearly indicated instead of giving frequencies of variants specific to clones?

We are primarily interested in clonal dynamics and seeing which subclones die out and which ones emerge and/or take over. It would be much simpler and easier to interpret figure if this information is directly given, instead of having to derive it by subtracting children from parent frequencies.

A solution that I recommend is to provide figures that are combination of what is shown in panels d and e in the figure shown at <https://github.com/hdng/clonevol>. For example, a sample S can be associated with, say, 100 small circles of the same color (the number can be less than 100 but should be the same for all samples). If it is estimated that 25% of cells from S belong to subclone 4, then subclone 4 (that is, tree node 4) can be assigned 25 dots of the color of S (better contrasts between colors or some other way to better distinguish samples will be needed - see below). Mutations can be assigned along the edges of the clonal tree and potential metastatic seeding events can be clearly marked by pointing to the edges along which migration happened."

Response: We note that the reviewer chooses to comment on our previously peer-reviewed and published methods paper here. The criticism evidently arises from our way of annotating clone sizes as pie charts whose sizes correspond directly to the genetic markers for a certain subclone. The subclone to be "fit" into the "pigeonhole" thus do not immediately correspond to the pie sizes but can easily be derived from comparing pies following each other in subclone lineages nested in each other as per the figure below:

However, we realize that this way of annotating clone size may appear confusing, and we thank the Reviewer for a very good suggestion! Therefore, we have now made a variant of the recommended plot type for the revised manuscript. We believe these plots add a new valuable layer of information, since they convey the clone sizes in an illustrative way.

Revision: We have made new plots accompanying all phylogenetic trees based on the clinical cases (Fig S1). Each sample were symbolized by a circle containing 20 small circles, where each small circle represents a clone size of 5%. The clone sizes were calculated by subtracting the sizes of a pie chart representing a downstream population from an upstream population (child from parent) as described in the figure above. We have also clarified in the legends to Fig 1 and supplementary Fig 1 that the Venn diagrams/pie charts illustrate the size of a population of cells that share the same mutation or set of mutations instead of clone sizes as previously stated.

REVIEWER COMMENTS

Reviewer #3 (Remarks to the Author):

I thank the authors for revising the manuscript. I dislike that they attempt to use the following as some sort of argument: "We also note that none of the other three reviewers have reported any issues with our annotation system." Different reviewers usually have different backgrounds and assess papers from different angles raising different concerns. Nevertheless, I appreciate the other arguments provided related to the figures and find them reasonable.

However, I noticed that my concerns about reproducibility and data analysis have still not been completely addressed. I will do my best to provide some more detailed examples about the parts/types of data that I could not find, as well as parts of the analysis that look problematic or hard to reproduce. Given that a lot of data analysis is done in this work, it is essential that it is easy to reproduce the results and that reasonable assumptions are made and appropriate methods used for data analysis.

1. It is stated that "The TAPS plots were visually inspected, and an allelic composition was assigned to each aberration". How was this done and if someone tries to do this "visual inspection" from scratch tomorrow how likely is it that they will obtain the same results?
2. I could not understand the following sentence: "When the appearance of the heat map indicated that a consistent phylogenetic tree could not be created, an event could be added as an "inferred stem" or "inferred private"."
3. It is stated that "The information from the curated heat maps was used as input data for DEVOLUTION to construct phylogenetic trees." Where can these input files used to run DEVOLUTION be found and, if available, why no reference to their location is provided in this sentence?
4. Where can one find full details of the output trees, such is subclone frequencies, branches where mutations are placed etc., in a format that is convenient to parse and use for automated comparisons, different visualisations etc. I noticed that in one of the tables some information is provided in actually quite nice format. However, I had difficulties interpreting those. For example, in sheet d1 there is SNV that can be identified by 6:29394620_OR11A1. When looking into SNV tree it has frequency 0.2 in Pre3 sample, whereas in SNV+CNA tree the same SNV has frequency 0.3. Why do we have this difference and where can detailed descriptions of the content of this and other tables be found?
5. In Section "Calculation of clone sizes based on integration of variant allele frequencies from exome sequencing with copy number from SNP-array data" the calculation of variant allele frequencies does not account for possible presence of multiple copy number amplifications/deletions overlapping with SNV. It is problematic to assume that mutation has constant multiplicity among cancer cells, as there might exist distinct subclones having different numbers of mutated copies. In fact, SNV can occur on one branch of phylogeny and one or multiple copy number changes of the region containing mutation can happen on another branch, which makes calculation of MCF values complex and doing MCF estimation independent

of phylogenetic analysis can be inappropriate for SNVs overlapping with gains and losses. This is well known and has been discussed in the literature (see for example <https://genomebiology.biomedcentral.com/articles/10.1186/s13059-015-0602-8>).

6. Related to the previous point, the provided description also looks incomplete. For example, it is not explained how the value of M used in the formula is found.

7. I do not see how one could reproduce what is attempted to be explained by the sentence "Scoring was performed according to the optimized cut-off of five consecutive 1 Mb bins for identifying true imbalances."

8. I found it difficult to follow definition of IGD in "Quantitative analyses of genetic diversity and phylogenetic trees". Definitions of dSi are lacking. What does it mean that subclones belong to the same sample type? What is difference between n and N? Please expand this whole section to provide full and precise details of what exactly is done here.

9. In the data table (the one containing worksheets a-i), in sheet b clone size percentage is as high as 176 percent (Patient_1 MP3 MP 19 7927142 C T EVI5L), which is counter intuitive. How is this explained?

10. As for DEVOLUTION method, which plays important role in data analysis in this paper, while the work introducing DEVOLUTION went through peer review, the manuscript describing it seems to be containing mistakes at some key places and I could not follow description of algorithm. To give a simple example, looking at the sentence "It has the dimensions $K \times N$, where K is the number of genetic alterations in the cluster and N is the cluster number." and the whole paragraph containing that sentence and Equation (2), I do not understand why the cluster number N would be used as a dimension (cluster number is just a unique identifier and it is not obvious why it would serve as a dimension). Maybe it is meant that N is the number of clusters, but if that was the case then "where K is the number of genetic alterations in the cluster" does not fit as it is not clear to what cluster one is referring.

11. Similar to some of the previous points, where can aberration calls and phylogenies from single cell data be found?

To summarize, I understand that it might not be possible for the authors to release raw sequencing data to make it publicly available for immediate download by anyone without asking for access permissions. However, starting from SNV and copy number calls, I believe that all the steps of the analysis, including the scripts used etc., should be made available and clearly documented. Having Python Jupyter Notebook containing all the analyses could be one possible way of sharing these. I emphasize that I would like to go through all the steps of analyses and check robustness of the methodology used, possibly also including testing the pipeline on some artificial manually generated phylogenies to assess how well it can capture true phylogenies.

MINOR:

Since there is a lot of data analysis in this study, the paper would benefit from having a detailed table with all details of what sequencing data has been generated, what is the coverage for each WES sample, how many single cells were sequenced in other cases etc.

Main paper, line 144: change "of" to "or" in "either after free growth (n=3) of after progression under".

In line 213, "turmed" is used instead of "turned".

The sentence on lines 262 and 263 looks grammatically incorrect.

The first sentence of abstract is missing "to" and should be "Neuroblastoma (NB) is one of the most lethal childhood cancers due to its propensity to become treatment resistant".

In Online Methods, some words are missing in "where each core needle biopsy/each fragment analyzed as a separate sample".

was -> were in "Because reads from murine DNA was removed".

Reviewer Comments and Response 2024:

In the following text, reviewer comments are copied verbatim in italics, followed by our comments and revisions in plain text.

Reviewer #3 (Remarks to the Author)

I thank the authors for revising the manuscript. I dislike that they attempt to use the following as some sort of argument: "We also note that none of the other three reviewers have reported any issues with our annotation system." Different reviewers usually have different backgrounds and assess papers from different angles raising different concerns. Nevertheless, I appreciate the other arguments provided related to the figures and find them reasonable.

Response: We thank the author for appreciation our extensively revised figures!

However, I noticed that my concerns about reproducibility and data analysis have still not been completely addressed. I will do my best to provide some more detailed examples about the parts/types of data that I could not find, as well as parts of the analysis that look problematic or hard to reproduce. Given that a lot of data analysis is done in this work, it is essential that it is easy to reproduce the results and that reasonable assumptions are made, and appropriate methods used for data analysis.

Response: We are very grateful for coming back to us with concrete suggestions!

1. It is stated that "The TAPS plots were visually inspected, and an allelic composition was assigned to each aberration". How was this done and if someone tries to do this "visual inspection" from scratch tomorrow how likely is it that they will obtain the same results?

Response: TAPS is a very well-established method since more than a decade and we have found it to be superior in interpersonal reproducibility when it comes to complex allelic imbalances as it provides a robust way to concomitantly visualize allelic status, copy number and clonality/subclonality. It has been used by us in a number of previous publications (e.g. Karlsson et al. Nat Genet 2018 PMID: 29867221, Mañas et al. Sci Adv 2022 PMID: 36306349; Rastegar et al. Clin Cancer Res 2023 PMID: 37140929; Woodward et al. Nat Commun 2023 PMID: 3696613). Over the years we have tried to robustly call copy number segments by automatic tools such as GISTIC and Rawcopy but have found that they require a very high amount of manual curation to harmonize breakpoints across samples, making it considerably more efficient to use TAPS in combination with the segmentation softwares (ChAS, the companion of CytoScan HD, and Nexus express, the companion to Oncoscan).

We take this opportunity to try to explain how it works and how situations that are difficult to interpret (potentially resulting in lower reproducibility), were resolved:

The plot below from the original TAPS manuscript (Rasmussen et al. Genome Biol 2011 PMID: 22023820) shows a tetraploid tumor. All segments having four copies are centered around log₂ ratio zero. Gains will have log₂ratio > 0 and losses < 0 relative to this. A balanced CNA will have a low allelic imbalance, while a LOH-event will receive a high ratio.

For example, 2+2 and 3+1 both have a log2ratio of zero, but 3+1 has an increased allelic imbalance and will thus appear higher up in the plot. 3+2 on the other hand will have a higher log2ratio compared to 2+2 and since the alleles are less balanced, they will have a higher allelic imbalance ratio. Hence clonal CNAs will end up at parallel lines forming a grid in the TAPS plots making the assessment of allelic compositions to the different CNA straightforward.

If there are few CNAs present in a sample or if several aberrations are subclonal (i.e. no clear grid is formed or the CNA deviates from the “grid”) it can be more complicated to assess the allelic composition. In such cases we rely on the comparison of the MSF values calculated based on log2ratio values with MSF-values calculated with mBAF values obtained from the allelic imbalance ratio from the TAPS plot. Only a correct assessment of the allelic ration will generate similar MSF-values. This is described in the Online method section of the paper *“Similar results (difference < approx. 0.1) from these calculations for the same aberrations was used as an indication of correct assessment of ploidy-level and allelic composition.”*

All TAPS-plots are available at the Zenodo archive linked to the manuscript and the allelic compositions for all CNA are annotated in the segment files (Table S2a).

Revision: None at present. The TAPS methodology is very well explained in the original publication. We could of course add an example as above to the manuscript but its content would largely duplicate information in the original publication.

2. I could not understand the following sentence: *“When the appearance of the heat map indicated that a consistent phylogenetic tree could not be created, an event could be added as an “inferred stem” or “inferred private”.”*

Response: In some rare occasions (see examples below) it became clear that we needed to annotate a CNA differently to avoid producing a tree with, for example, faulty excessive branching or to illustrate that a common breakpoint was present in several samples although the CNA differed in the number of alleles between the samples (see example 1 below). In these cases, alterations were inferred into the input files for DEVOLUTION to resolve controversies in a biologically sound way. If such alteration was made it is indicated to the left of the heatmap in the “comment” column (Table S1d).

Example1: In Patient_1 all samples contained a 17q gain with the same breakpoints (**marked in blue** in the figure) but with different allelic compositions. Since it is unlikely that the exact same breakpoint should appear in parallel, a 17q (1+2) gain was inferred as a stem event although the four Pre-samples demonstrated this CNA with other allelic compositions (2+2, and 2+4 in parallel clones).

Patient_1												
ploidy: 2n												
CNA												
segment	chr/cytoband	allelic composition	comment	Pre1	Pre2	Pre3	Pre4	MP1	MP2	MP3	MP4	subclone
chr11:68257059-71047824	11q13q13	amp	CCND1/FGF19 amp	1	1	1	1	1	1	1	1	stem
chr12:69074905-70814518	12q15q15	amp	MDM2 amp	1	1	1	1	1	1	1	1	stem
chr12:57924714-58460909	12q13q14	amp	CDK4 amp	1	1	1	1	1	1	1	1	stem
chr12:25937301-26681930	12p12p11	amp		1	1	1	1	1	1	1	1	stem
chr12:121565870-122204084	12q24q24	amp		1	1	1	1	1	1	1	1	stem
chr12:12047618-12410964	12p13p13	1+0		1	1	1	1	1	1	1	1	stem
chr5:122889793	8p23p21	1+0		1	1	1	1	1	1	1	1	stem
chr17:39488988-80263427	17q21q25	1+2	inferred stem-common breakpoint17q+ (1+2)	1	1	1	1	1	1	1	1	stem
chr17:39488988-80263427	17q21q25	2+2	mixed 17q+ (2+2)	0,70	0,65	0,70	0,65	0	0	0	0	C
chr5:1-180915260	5 whole	1+3		0,40	0,40	0,40	0,40	0	0	0	0	D
chr6:1-171115067	6 whole	2+0	mixed +6 (2+0)	0,40	0,40	0,40	0,40	0	0	0	0	D
chr17:39488988-80263427	17q21q25	2+4	mixed 17q+ (2+4)	0,30	0,25	0,30	0,25	0	0	0	0	E
chr6:1-171115067	6 whole	1+2	mixed +6 (1+2)	0,30	0,25	0,30	0,25	0	0	0	0	E
chr4:52691175-190986238	4q11q35	1+3		0	0	0,20	0,30	0	0	0	0	F
chr8:23293013-146364022	8p21q24	1+2		0	0,1	0	0,1	1	1	1	1	G
chr5:1-180915260	5 whole	1+2		0	0	0	0	1	1	1	1	B
chr17:39488988-80263427	17q21q25	1+3		0	0	0	0	1	1	1	1	B
chr13:19084823-115103150	13 whole	1+2		0	0	0	0	1	1	0	1	A

Example 2: A whole 7+ (trisomy 7, denoted "7 whole" 1+2 in the table below) was detected in two samples in Patient_2, marked in green. In sample LP it was detected in all cells while it was only detected in 30% of the cells in sample MP3. A clonal aberration (subclone A) was detected in MP3 and ought to be an upstream event of +7 in this sample. Since subclone A is absent in sample LP it is likely that the gains of chromosome 7 in LP and MP3 are parallel events.

Patient_2															
ploidy: 2n															
CNA															
segment	chr/cytoband	allelic composition	comment	Pre1	Pre2	Pre3	Pre4	LP	MP1	MP2	MP3	MP4	MP5	MP6	subclone
chr2:3694994-4449917	2p25p25	amp		1	1	1	1	1	1	1	1	1	1	1	stem
chr2:15871375-16528803	2p24q24	amp	MYCN amp	1	1	1	1	1	1	1	1	1	1	1	stem
chr17:46042327-80263427	17q21q25	1+2		1	1	1	1	1	1	1	1	1	1	1	stem
chr6:1-39275282	6p25p21	1+2		1	1	1	1	1	1	1	1	1	1	1	stem
chr17:27740706-29578360	17q11q11	1+0		1	1	1	1	1	1	1	1	1	1	1	stem
chr17:29796197-31918109	17q11q12	1+0		1	1	1	1	1	1	1	1	1	1	1	stem
chr19:1-4206770	19p13p13	1+0		1	1	1	1	1	1	1	1	1	1	1	stem
chr17:29602610-29785370	17q11q11	0+0	NF1 homozygous del	1	1	1	1	1	1	1	1	1	1	1	stem
chr1:1-111434711	1p36p13	1+0	inferred stem /mixed/1p (1+0)	1	1	1	1	1	1	1	1	1	1	1	stem
chr1:1-111434711	1p36p13	2+0	mixed 1p (2+0)	0,6	0,2	0,3	0,4	1	1	1	1	1	1	1	F
chr1:111459847-249212878	1p13q44	1+2		0,6	0,2	0,3	0,4	1	1	1	1	1	1	1	F
chr11:100371319-124313722	11q22q24	1+0		0	0	0	0	0	0	0	1	0	0	1	A
chr7:1-159138663	7 whole	1+2	parallel 7+	0	0	0	0	1	0	0	0	0	0	0	B
chr7:1-159138663	7 whole	1+2	parallel 7+	0	0	0	0	0	0	0,3	0	0	0	0	J

Revision: We have added clarifying examples to the sentence Reviewer 3 commented on in Online Methods.

3. It is stated that "The information from the curated heat maps was used as input data for DEVOLUTION to construct phylogenetic trees." Where can these input files used to run DEVOLUTION be found and, if available, why no reference to their location is provided in this sentence?

Response: We had chosen to not include the data in the exact format that was used for input for DEVOLUTION in the supplement, since all information needed to create the input file was already present in the heatmaps (Table S1d). The input-file for DEVOLUTION is the same data as depicted in the heatmap but converted to a long table format.

An example of an input data set for devolution, Med LogR, VAF, Type, Method, Cytoband/Gene Clone size (%) can be replaced by NA is inserted below:

```
> head(datasegment)
  Tumor ID Samples Chr Start End Med LogR VAF (TRS) Type Method Cytoband/ Gene Clone size (%)
1 Tumor1 ALL 1 0 247249719 NA NA GAIN SNP Array WHOLE 100
2 Tumor1 ALL 3 63411 197852564 0.35 NA GAIN SNP array WHOLE 100
3 Tumor1 ALL 4 40421567 41888869 -0.65 NA LOSS SNP array 4p14p13 100
4 Tumor1 ALL 6 156974 170919481 0.1 NA GAIN SNP array WHOLE 100
5 Tumor1 B1 2 21494 45575110 NA NA GAIN SNP array 2p25p22 100
6 Tumor1 B1 4 69404 43444897 NA NA LOSS SNP array 4p16p15 80
```

Revision: To facilitate for the reader who wants to re-create the phylogenetic trees included in this paper, we have now rearranged our tables to directly match the input file format for Devolution. Since these new tables contain the same information as in Table S1d we didn't add these files as a supplement to the manuscript instead they can be found at Zenodo (<https://zenodo.org/records/10727558>).

*4. Where can one find full details of the output trees, such is subclone frequencies, branches where mutations are placed etc., in a format that is convenient to parse and use for automated comparisons, different visualizations etc. I noticed that in **one of the tables** some information is provided in actually quite nice format. However, I had difficulties interpreting those. For example, in sheet d1 there is SNV that can be identified by 6:29394620_OR11A1. When looking into SNV tree it has frequency 0.2 in Pre3 sample, whereas in SNV+CNA tree the same SNV has frequency 0.3. Why do we have this difference and where can detailed descriptions of the content of this and other tables be found?*

Response: All information that Reviewer 3 is asking for is included in the manuscript but not in the same file, because we found no pedagogic way to merge all output data in the same table. The allocation of mutations to specific branches is included in a R-element in the output of Devolution but the final frequencies of the different mutations and in which subclone the mutations end up in are available in the heatmap format Table S1d, the raw data (log2 ratios and allelic imbalance values from TAPS plots) in the segment files Table S2a and subclone sizes are visualized in the recently added Fig. S1a-i V-VII (IV; Patients 2 and 9).

To increase the stability of the data, we don't base the final clone sizes on just one mutation (if applicable). Instead, we first visualize the global pattern of the mutations in a patient in the heatmap format and then calculate the final mutated clone fractions (MCF). When the resolution of the clonal landscape increases while going from SNV-data only to CNA+SNV data, the final MCF -values can change slightly as in the example the Reviewer brought up. VAF-values and the preliminary MCF values are provided in Table S2b, and the final MCF values in the heatmaps in Table S1d.

The calculation of the final frequencies is described in the Online Methods, paragraph "Clonal deconvolution based on CNAs detected by SNP array in clinical samples": "The MCF values were visualized in a heat map format and variants with similar MCF pattern over samples were grouped together (Table S1d). An average of the MCF values per sample and group were calculated and that value rounded to the nearest 0.1 became the final clone size for all the variants in that group."

Revision: We have made a long-table format of the information also present in the heatmap Table S1d and uploaded this data to Zenodo (<https://zenodo.org/records/10727558>). These files can be used for an overview of mutation frequencies and which branch and subclone each mutation ends up in. In addition, we have made a new supplementary figure describing our workflow from raw data to phylogenetic trees (Fig S1p) where the location of all input and output data are showed together with an overview of which softwares and methods we have used in this study.

5. In Section "Calculation of clone sizes based on integration of variant allele frequencies from exome sequencing with copy number from SNP-array data" the calculation of variant allele frequencies does not account for possible presence of multiple copy number amplifications/deletions overlapping with SNV. It is problematic to assume that mutation has constant multiplicity among cancer cells, as there

might exist distinct subclones having different numbers of mutated copies. In fact, SNV can occur on one branch of phylogeny and one or multiple copy number changes of the region containing mutation can happen on another branch, which makes calculation of MCF values complex and doing MCF estimation independent of phylogenetic analysis can be inappropriate for SNVs overlapping with gains and losses. This is well known and has been discussed in the literature (see for example <https://genomebiology.biomedcentral.com/articles/10.1186/s13059-015-0602-8>).

Response: We fully agree with the reviewer on the point that copy number must be taken into account in clonal deconvolution of SNVs – this is particularly important in our field (pediatric cancer) where driver mutations are often outnumbered by copy number aberrations. In fact, our pipeline was specifically tailored to handle this issue and therefore we respectfully disagree on the point that our calculation of variant allele frequencies did not account for this.

To further clarify, we have taken the CNA-state (number of alleles and subclonal frequency (shown in Table S2 column O-Q) into account when calculating clone sizes of SNVs. The presence of a CNA aberration with different allelic imbalances generating/ending up in parallel clones is also accounted for in this table. The M-values used for the individual calculations are presented in Table S2b (column N) and in Table S2d (column V).

There are of course situations that cannot be resolved by bulk sequencing, for example an SNV and a copy number calculated to have similar and very small frequencies, making them fit into many places in the subclone landscape; resolving this would probably need single cell sequencing, which cannot robustly be performed on FFPE material to gain SNV and copy number information at the same time. But such cases should be very few as it requires an accidental convergence in clone size; all studies of cancer phylogenies from bulk sequencing data should contain this minor source of error.

The PhyloWGS pointed out in the reference seems to be tailored for WGS-data which cannot currently be obtained from FFPE material with high quality.

Revision: We have already provided the data the Reviewer is asking for. The only improvement we foresee here is better illustration of our source data – see above and the Summary comment.

6. Related to the previous point, the provided description also looks incomplete. For example, it is not explained how the value of M used in the formula is found.

Response: We always start by assuming that $M=1$, but if this gives rise to a clone size above 1.2 we increase the value as a clone cannot be larger than 100% of the tumor cell population. If the SNV is placed on a gained segment our approach is to avoid overcalling of heterogeneity, i.e. we use a value of M that results in a clonal mutation or if not applicable give rise to a mutation with the same clone frequency as other identified mutations in that sample.

How we pick the value of M is described in Online Methods, but unfortunately not in the first paragraph where M is mentioned in the paragraph “Calculation of clone sizes based on integration of variant allele frequencies from exome sequencing with copy number from SNP-array data”. Instead, this was mistakenly described in the section “clonal deconvolution from sequencing data from PDX models”.

Revision: We have moved the sentence where we describe how we pick the value of M to the paragraph “Calculation of clone sizes based on integration of variant allele frequencies from exome sequencing with copy number from SNP-array data” where M is first mentioned.

7. I do not see how one could reproduce what is attempted to be explained by the sentence "Scoring was performed according to the optimized cut-off of five consecutive 1 Mb bins for identifying true imbalances."

Response: We understand that this sentence can cause confusion and duly apologize.

Revision: We have re-written this sentence to "Identified copy numbers encompassing less than five consecutive 1 Mb bins in the dataset, were excluded from further analysis, except for high-grade amplifications of the MYCN region in chromosome arm 2p and of the MAML3 region in 4q28.3-q31.1, in which case two consecutive 1 Mb bins were considered true gains."

8. I found it difficult to follow definition of IGD in "Quantitative analyses of genetic diversity and phylogenetic trees". Definitions of dS_i are lacking. What does it mean that subclones belong to the same sample type? What is difference between n and N ? Please expand this whole section to provide full and precise details of what exactly is done here.

Response: We came up with IGD to be able to compare the heterogeneity between sample types (if the sample is taken before (Pre) or after therapy (Post) or from a metastasis) regardless of the sample number differed between the groups. In our material IGD truly reflected the visual appearance of the trees, thus we found it to be very useful. In the previous version of the manuscript, we described the calculation of IGD in what we believed to be a simplified version of the formula, but we now, thanks to Reviewer 3, realize that it might be more pedagogic to describe the formula in another way.

To calculate IGD, we annotate two distances for all (n) subclones detected in the sample type we would like to calculate IGD for: dS_i defined as the distance from the closest common node for the specific sample type to be investigated to the subclone i , and DS_i defined as the distance from the root of the stem to the subclone i . Thus, in the example below for subclone A (S_A) belonging to the pre samples illustrated in green, $dS_A=0$, while $DS_A=19$.

IGD is the ratio between the sum of all the distances from the closest common node to the individual subclones $dS_1+dS_2+..+dS_n$ and the distances $DS_1+DS_2+..+DS_n$ from the root of the stem to the same subclones:

$$IGD=(dS_1+dS_2+..+dS_n)/(DS_1+DS_2+..+DS_n)$$

IGD calculated on subclones detected in the Pre samples (A, E and G in green). Here the number of common aberrations for the Pre- samples, the common branch length l , is equal to $13+6=19$, $dS1=0$, $dS2=2$ and $dS3=3$, and $N=3$.

$$IGD(\text{pre samples, green}) = \frac{0+2+3}{19+21+22} = 0.08$$

while IGD for the post samples (red subclones, B, C, D, F) will be

$$IGD(\text{post samples, red}) = \frac{10+9+16+18}{23+22+29+31} = 0.5$$

The higher IGD-value retrieved for the subclones identified in the post samples compared with the pre samples is reflecting the visual appearance of the tree, ie the subclones identified in the Pre samples are more homogenous than the subclones found in the Post samples.

In the previous version of the manuscript, we had described the formula for IGD as:

$$IGD = \frac{dS1 + dS2 + \dots + dSn}{dS1 + dS2 + \dots + dSn + l \times N}$$

N is the number of subclones detected in the sample type, and the length of the stem (l).

Revision: We have expanded the section describing the IGD-calculations in the Online Methods and used the hopefully more pedagogic formula. We have also changed the numbers 1, 2, to n to lowercase to highlight that these numbers are indexes and explained what we refer to when we write “sample type”.

9. In the data table (the one containing worksheets a-i), in sheet b clone size percentage is as high as 176 percent (Patient_1 MP3 MP 19 7927142 C T EVI5L), which is counter intuitive. How is this explained?

Response: We had the following inclusion criteria for a mutation “Variants were excluded if detected in $>1\%$ in the corresponding normal sample, had a VAF value below 0.1 in all samples and were covered <100 reads in all samples.” This can result in a variant (EVI5L in the example) being included in the study although the specific location in a specific sample was only covered 48 times giving rise to unstable data in a specific sample. In such cases we were helped by our approach to call final clone sizes based on the mean of a group of variants with similar clone sizes.

One can argue that these variants should have been removed from the study, but we believe that it is more transparent, and lead to less data distortion, to include them with their deviant clone sizes since they are clearly present in the samples. There is the possibility of a subclonal tetraploid background in these tumors explaining this, but we can’t claim that with certainty since we haven’t analyzed all samples with single cell analyses. Since these high clone sizes are rounded to 100%, they don’t induce false heterogeneity in the data.

Revision: We have added a section in Online Methods “Deconvolution of clinical samples after targeted re-sequencing” where we have acknowledged this problem, described our rationale for keeping these variants and write that we round the clone sizes to 1.0.

10. As for DEVOLUTION method, which plays important role in data analysis in this paper, while the work introducing DEVOLUTION went through peer review, the manuscript describing it seems to be containing mistakes at some key places and I could not follow description of algorithm. To give a simple example, looking at the sentence “It has the dimensions $K \times N$, where K is the number of genetic alterations in the cluster and N is the cluster number.” And the whole paragraph containing that sentence and Equation (2), I do not understand why the cluster number N would be used as a dimension (cluster number is just a unique identifier and it is not obvious why it would serve as a dimension). Maybe it is meant that N is the number of clusters, but if that was the case then “where K is the number of genetic alterations in the cluster” does not fit as it is not clear to what cluster one is referring.

Response: We first need to clarify that this refers to our paper published 2.5 years ago in another journal and is not part of the current manuscript) <https://www.nature.com/articles/s42003-021-02637-6>. We are sorry that this paper was perceived unclear and have thus made the effort to clarify the issue here:

In the section pointed out in the DEVOLUTION paper, we merely explain one of the steps in which a matrix is created and saved. We created a *matrix which itself had the dimensions $K \times N$* . The matrix has K rows and N columns. Each column represents a cluster. The rows in a specific column lists the genetic alterations present in that cluster. It is simply a table illustrating which genetic alterations are encompassed by each cluster. These clusters are nested according to the pigeon-hole principle in the subsequent parts of the algorithm.

The matrix can for example look like this, where the first line is just saved as column names in R.

Cluster 1	Cluster 2	Cluster 3	Cluster 4	Cluster 5	Cluster 6
CNV 1	CNV 5	CNV 4	CNV 7	CNV 10	CNV 13
CNV 3	CNV 2	CNV 6	CNV 9	CNV 11	CNV 14
0	CNV 8	0	0	CNV 12	0

Here the mutation space contains the alterations (CNV 1, CNV 2, CNV 3, CNV 4, CNV 5, CNV 6, CNV 7, CNV 8, CNV 9, CNV 10, CNV 11, CNV 12, CNV 13, CNV 14). The matrix has the *dimensions 3×6* . No element is the same as another one unless it is a zero.

Revision: It is difficult to take action since the question concerns an already published paper and the sentences referred to are not part of this manuscript. If recommended by the reviewer or editor, we could add an explanatory note on DEVOLUTION in the supplements, but it may not add very much to the original publication.

11. Similar to some of the previous points, where can aberration calls and phylogenies from single cell data be found?

Response: All phylogenies can be found in Figure 6d-e and Figure S7 as mentioned in the manuscript.

Revision: Input files for SCEM displaying aberration calls per 1 MB bins for the single cell data have now also been added to Zenodo (<https://zenodo.org/records/10727558>).

*To summarize, I understand that it might not be possible for the authors to release raw sequencing data to make it publicly available for immediate download by anyone without asking for access permissions. However, starting from SNV and copy number calls, I believe that all the steps of the analysis, including the scripts used etc., should be made available and clearly documented. Having **Python Jupyter Notebook** containing all the analyses could be one possible way of sharing these. I emphasize that I would like to go through all the steps of analyses and check robustness of the methodology used, possibly also including testing the pipeline on some artificial manually generated phylogenies to assess how well it can capture true phylogenies.*

Response: Besides GDPR-protected raw data (deposited at EGA), all the data needed to repeat our analyses of bulk samples are already available in the Supplement section. We have in this revised version also added the single cell data and changed the layout supplementary table S1d so it can be directly used as input files for DEVOLUTION. The reliability of the segment data and heatmaps can be validated by visual comparison to the uploaded TAPS-plots that also contain whole genome views. The methods and bioinformatic tools are publicly available and versions etc are described in the manuscript.

The analyses of copy numbers done in this study are based on visual interpretation and could not be explained in code (see above regarding TAPS). For some steps we have used published scripts or software which are referred to. Hence, we don't find it beneficial for the reader to summarize the steps in for example Python Jupyter Notebook, especially since the DEVOLUTION script can't be used in that environment since Jupyter Notebook doesn't support newer versions of R.

Most of Reviewer 3 concerns seem to arise from DEVOLUTION. We want to stress that this is a peer-reviewed published method. In the original DEVOLUTION paper, we thoroughly evaluated the method and compared it to simulated bulk whole genome genotyping data. This allowed us to compare the identified subclones and phylogenetic trees generated with DEVOLUTION and the true subclones and phylogeny, obtained from the simulation and it showed a robust performance, especially for subclones > 10 % in size as well as events present in more than one sample. The method performs subclonal reconstruction from the data in a form of pigeon-hole principle method, while integrating information from all samples provided, while trying to minimize occurrences of parallel evolution and convergent evolution, a method which is not especially controversial. This is followed by phylogenetic reconstruction based on the identified subclones using the maximum likelihood method and maximum parsimony method, both well-known and established methods used for decades for phylogenetic reconstruction of the evolutionary relationship between biological entities.

We finally would like to note that in all publications we have studied in over the past years in cancer phylogenetics, there is an element of subjective choice regarding which models to use and how to clonally deconvolve complex situations. Having ourselves downloaded raw data, we have observed multiple situations, typically affecting distal branches of trees, that could be interpreted in a way different from what was presented. This did not make us disagree with the important conclusions in the paper. We are confident that the same principle could be applied here, i.e. there can be variants of interpretation of details affecting details in the phylogenetic trees, which could be discussed at length, but which would not affect the overall results. The only solution to this is transparency what choices were made, and we will keep doing our best to ensure this.

Revision: For reasons explained above, we have found that the best way to describe our stepwise analysis from start to finish is by adding a novel figure to the supplement, Fig S1p where we have created a flowchart which describe in detail all steps, we have followed when we analyze the data,

which bioinformatic methods we have used, and where the input and output data can be found. We have the input data for DEVOLUTION to Zenodo (<https://zenodo.org/records/10727558>) to facilitate independent validation of tree construction. We have also added a manual to Zenodo describing in detail how to use DEVOLUTION to facilitate the comparison of this method to other,

MINOR:

Since there is a lot of data analysis in this study, the paper would benefit from having a detailed table with all details of what sequencing data has been generated, what is the coverage for each WES sample, how many single cells were sequenced in other cases etc.

Response and revision: We have added a figure describing the workflow as described above. The number of cells sequenced has been added to Table S1. The mean coverage and cut-offs are already presented in Methods. The number of reads per SNV is already presented in the supplements (Table S1b). In summary, the data is already presented in the paper.

Main paper, line 144: change "of" to "or" in "either after free growth (n=3) of after progression under".

In line 213, "turmed" is used instead of "turned".

The sentence on lines 262 and 263 looks grammatically incorrect.

The first sentence of abstract is missing "to" and should be "Neuroblastoma (NB) is one of the most lethal childhood cancers due to its propensity to become treatment resistant".

In Online Methods, some words are missing in "where each core needle biopsy/each fragment analyzed as a separate sample".

was -> were in "Because reads from murine DNA was removed".

Response and revision: We are grateful to the reviewer for pointing out these errors. They are now corrected.

REVIEWER COMMENTS

Reviewer #3 (Remarks to the Author):

1. Neuroblastomas are well known as having a large presence of copy number changes. This is also stated by the authors in the sentence "The high proportion of CNAs compared to driver point mutations complicates evolutionary analyses of NB." Therefore, if single nucleotide mutations are used in the analysis, it is of a great importance to perform correct adjustments for the influence of copy number changes on mutated and non-mutated reads. I am concerned that this is not done properly in this work and the related issue that I brought up in the previous round of review has not been addressed. More precisely, it is assumed that a given mutation has a constant number of mutated copies in the cells where the mutation is present. In other words, it is assumed that there exist two groups of cells: (a) cells without the mutation and (b) cells having M copies of the mutation. What I am referring to is described in section "Calculation of clone sizes based on integration of variant allele frequencies from exome sequencing with copy number from SNP-array data". In tumors having a lot of copy number changes this assumption is problematic as the region containing mutation can undergo multiple gains or losses over time. I pointed to PhyloWGS just to as one of the papers describing some of these problems. That paper is not the only one describing this and there are more recent ones discussing how improper handling of copy number changes can result in phylogenies that are unrealistic. Some other examples of this can be found in the paper "DeCiFering the elusive cancer cell fraction in tumor heterogeneity and evolution".

What I found to also be problematic is the following. In their analysis the authors state that "As a starting point, M was assumed to be 1, but changed to 2 if MCF became higher than 1.20 and if more than two alleles were present at the location of the mutation." Consider a variant that occurs at the branch that in some sample comprises 30% of tumor cells, and assume that there is a gain of the copy with the mutation and that this gain is affecting 75% of these cells containing the mutation. In other words, 30% of all tumor cells have the mutation, and among these 30%, 75% of them have two copies of the mutation and the remaining 25% have one copy. This gives us that there are 3 groups of cells in the tumor: (a) those not containing mutation, (b) those having 1 copy of it and (c) those having 2 copies. Not just that this case is not covered, but the presence of multiple mutated copies would not be detected because MCF cutoff is set high and could work only for variants affected by gains in high proportion of cells.

2. Related to the previous point, it is surprising that large majority of copy number changes and SNVs are present at clone size of 100% (column clone size in DEVOLUTION input). This raises concerns about the sensitivity of this methodology in detecting copy number changes and mutations that are not present in all cancer cells. This is especially concerning in light of the recent discoveries, made using single cell analysis, of extensive subclonal copy number changes in some tumor types (<https://pubmed.ncbi.nlm.nih.gov/33762732/>) and the presence of parallel haplotype-specific variation (<https://www.nature.com/articles/s41586-022-05249-0>). Is neuroblastoma known to have most of its copy number changes and mutations at 100% frequency?

3. Possibility of having more than one phylogeny that work equally well for a given input is known when searching for the best phylogeny of tumor (see <https://journals.plos.org/ploscompbiol/article?id=10.1371/journal.pcbi.1008400>). How is this addressed in this work? Do alternative, equally good, phylogenies exist and do they support the main biological claims made here? If they exist, how can we obtain them?

4. I could not reproduce the results. I followed Microsoft Word document provided at Zenodo "How to build a tree.docx" (<https://zenodo.org/records/10727558>).

From the same Zenodo archive I downloaded "Input Devolution.xlsx" file, stored it into my directory and followed all the instructions from the document, using R version 4.2.2 as suggested,

but when executing the command

```
datasegment <- splitdata(data,name=x)
```

I got the following error

```
"Error in `[<-`(`*tmp*`, s, 1, value = k) : subscript out of bounds"
```

Then I suspected that x should be set to some value but no guidance on what that value should be is provided. There are some comments that I believe are supposed to serve as a guidance: "Extracting the beginning and end position of each sample in the segment file. #Declare which tumor you want to analyze. Specified by the first column in your data set." but they never mention x and I could not follow them.

Reviewer #5 (Remarks to the Author): Arbitrating reviewer; expert in computational cancer genomics, cancer evolution, and phylogenetic analysis

As reviewer #3 noted, some of the assumptions made when using DEVOLUTION, particularly in relation to the MCF calculation (e.g. described in the Methods section starting on l. 309), have indeed been shown to lead to erroneous reconstructions. In addition, there are tools that attempt to correct these problems, such as PhyloWGS (which may not be applicable here as it was developed for WGS data) and DeCiFer, which the reviewer also pointed out. Unfortunately, there seems to be only a benchmark against one other algorithm (MAGOS) in Andersson et al. and this is not based on predictions of ground truth from simulations, but applied to real tumor data, which prevents a definitive assessment of the performance and accuracy of DEVOLUTION.

Since the main conclusions of the current paper rely on accurate phylogenetic reconstructions, I think it would be important to re-run the data through DeCiFer (which I think should be possible since DeCiFer

allows integration of SNV and CNV data across multiple samples) to check the robustness of the inferred phylogenies. Alternatively, the authors should at least use MCF estimates from another algorithm (like DeCiFer) as input for their density-based clustering scheme, as they already mention as a possibility in Andersson et al.: "There are multiple dedicated tools that can infer MCFs from sequencing data such as the clustering algorithms PyClone, SciClone, and MAGOS, and recently also DeCiFer and for structural variants SVclone which could be used as an input for DEVOLUTION."

I also agree that the stability of the trees should be discussed (it is discussed in Andersson et al.) and that the selection criteria for the trees shown should be clearly stated.

Reviewer Comments and Response:

In the following text, reviewer comments are copied verbatim in italics, followed by our comments and revisions in plain text.

Reviewer #3 (Remarks to the Author)

Reviewer #3 (Remarks to the Author):

1. Neuroblastomas are well known as having a large presence of copy number changes. This is also stated by the authors in the sentence "The high proportion of CNAs compared to driver point mutations complicates evolutionary analyses of NB." Therefore, if single nucleotide mutations are used in the analysis, it is of a great importance to perform correct adjustments for the influence of copy number changes on mutated and non-mutated reads. I am concerned that this is not done properly in this work and the related issue that I brought up in the previous round of review has not been addressed. More precisely, it is assumed that a given mutation has a constant number of mutated copies in the cells where the mutation is present. In other words, it is assumed that there exist two groups of cells: (a) cells without the mutation and (b) cells having M copies of the mutation. What I am referring to is described in section "Calculation of clone sizes based on integration of variant allele frequencies from exome sequencing with copy number from SNP-array data". In tumors having a lot of copy number changes this assumption is problematic as the region containing mutation can undergo multiple gains or losses over time. I pointed to PhyloWGS just to as one of the papers describing some of these problems. That paper is not the only one describing this and there are more recent ones discussing how improper handling of copy number changes can result in phylogenies that are unrealistic. Some other examples of this can be found in the paper "DeCiFering the elusive cancer cell fraction in tumor heterogeneity and evolution".

What I found to also be problematic is the following. In their analysis the authors state that "As a starting point, M was assumed to be 1, but changed to 2 if MCF became higher than 1.20 and if more than two alleles were present at the location of the mutation." Consider a variant that occurs at the branch that in some sample comprises 30% of tumor cells, and assume that there is a gain of the copy with the mutation and that this gain is affecting 75% of these cells containing the mutation. In other words, 30% of all tumor cells have the mutation, and among these 30%, 75% of them have two copies of the mutation and the remaining 25% have one copy. This gives us that there are 3 groups of cells in the tumor: (a) those not containing mutation, (b) those having 1 copy of it and (c) those having 2 copies. Not just that this case is not covered, but the presence of multiple mutated copies would not be detected because MCF cutoff is set high and could work only for variants affected by gains in high proportion of cells.

Response: We thank the reviewer for this thorough explanation and agree with the reviewer that that there is a risk of oversimplification when one does not take further copy number alterations of mutated alleles into account. However, we would like to point out that

- (1) We have relatively few sequence mutations in comparison to copy number aberrations, making the possible situation of having a mutation located to a mixed-copy-number segment overall low.
- (2) These special cases will mostly affect details in the branches of the trees since the scenario outlined entails the presence of several small clones

(3) However, **most importantly**, we were aware of the problem when handling the data and have closely monitored situations where the region containing mutation can undergo multiple gains or losses over time. Mixed populations of this type are detected and delineated by the TAPS view. Thus, we would in fact detect a situation where there are parallel subclones with different copy numbers of a mutated locus. Moreover, this is accounted for (variables f1 and f2) in the formulas present in Online Methods (pages 18-19). However, the next step – fitting the mutated allele copy numbers across this clonal landscape is complicated and “ground truth” sometimes not possible to reach without single cell sequencing, which is not possible on FFPE-derived DNA. In our case, we used a heuristic approach, i.e. trying combinations that would create most consistency across the data, i.e. for variants on a complicated copy-number background we chose mutated allele copy numbers guided by MCF values in other samples from the same patient, having less complicated backgrounds. One example where we used our heuristics to resolve the subclone landscape is given below (Patient 12).

		Pre1	Pre2	Pre3	MR1	
VAF	17:6381992	0,8	0,8	0,8	0,3	All samples (1+0) M=1
	14:102431252	0,5	0,5	0,4	0,3	Pre2 (2+0) 20% M= 1,2, MR1 (2+0) 40% M= 1,4
	4:77926641	0,8	0,6	0,9	0,4	Pre 1 and Pre2 (2+0/1+2), Pre3 (2+0), MR (1+2), M=2 in all samples
	4:110765234	0,4	0,3	0,4	0,2	Pre 1 and Pre2 (2+0/1+2), Pre3 (2+0), MR1 (1+2), M=1 in all samples
	13:23912863	0,3	0,3	0,3	0,3	All Pre samples (2+3) M=1, MR1 (2+0/3+0) M=1.9
Original MCFs	17:6381992	1,1	1,3	1,2	1,0	
	14:102431252	1,1	1,0	1,0	1,0	
	4:77926641	1,0	0,9	1,1	0,9	
	4:110765234	1,0	1,0	1,1	1,0	
	13:23912863	1,0	1,0	1,1	0,9	
DeCiFer MCF	17:6381992	0,9	0,9	0,9	0,7	
	14:102431252	1,0	1,0	0,9	1,3	
	4:77926641	1,0	0,9	1,0	1,1	
	4:110765234	1,0	0,9	1,0	1,2	
	13:23912863	1,0	1,0	1,1	1,0	

Legend to figure above: Here genome positions of mutations are given to the left of the table and VAFs/MCFs inside the table – made as a heat map to illustrate values. To the right are given copy numbers of diverse subclones underlying the VAF and MCF values. M correspond to the number of copies for the mutated allele. Note the high agreement to the original mutated clone fractions MCFs resulting from our heuristic approach and those resulting from the DeCiFer software, now used for benchmarking in the revised version.

We acknowledge that a software-based approach may be advantageous, although we did not find any suitable program. Still, we agree that some alternative method of reference to our heuristic approach would be pertinent. We thank the reviewer for pointing to DeCiFer as an alternative, which proved useful to us in the benchmarking now done in the latest revision of the paper. We were aware of DeCiFer before but (as becomes clear below) it has certain limitations when dealing with tumors having a high number of copy number changes and polyploidy and few mutations.

Revisions:

- (1) We have now tried to describe our heuristic approach in detail in Online Methods (pages 19-20), hoping this will clarify that we already from the beginning tackled the problem pointed out by the reviewer.
- (2) As far as possible, we have now also benchmarked clone size calculations with DeCiFer as follows in detail below.

2. Related to the previous point, it is surprising that large majority of copy number changes and SNVs are present at clone size of 100% (column clone size in DEVOLUTION input). This raises concerns about the sensitivity of this methodology in detecting copy number changes and mutations that are not present in all cancer cells. This is especially concerning in light of the recent discoveries, made using single cell analysis, of extensive subclonal copy number changes in some tumor types (<https://pubmed.ncbi.nlm.nih.gov/33762732/>) and the presence of parallel haplotype-specific variation (<https://www.nature.com/articles/s41586-022-05249-0>). Is neuroblastoma known to have most of its copy number changes and mutations at 100% frequency?

Response: The SNP-array based copy number detection approach is considered to have a sensitivity down to around 10% -- we agree with the reviewer that below this limit we cannot claim detection. Our single cell data in the paper in fact show just the type of extensive variation at single cell level in some cases that was pointed out by the reviewer. However, it is important to note that

- (1) These minor clones (<10%) will affect distal branches of the trees, not affecting the overall patterns and our conclusions regarding collateral clonal replacement
- (2) The 100% (=clonal) in our heat maps often refers to single 3-5 mm sized samples, not the entire population. Thus, it implies local clonality as opposed to stem aberrations. The data fit very well with our previous studies, showing that neuroblastomas often have clonal sweeps across spatial domains creating a high number of locally clonal events (Karlsson et al. Nat Genet 2018 PMID: 29867221).

Revision: None, as overall results will not be affected by this, and single cell methods of the type mentioned cannot be applied to FFPE samples. Furthermore, the frequent regional clonality is expected from previous studies.

3. Possibility of having more than one phylogeny that work equally well for a given input is known when searching for the best phylogeny of tumor (see <https://journals.plos.org/ploscompbiol/article?id=10.1371/journal.pcbi.1008400>). How is this addressed in this work? Do alternative, equally good, phylogenies exist and do they support the main biological claims made here? If they exist, how can we obtain them?

Response: The phylogenies based on bulk DNA data as well as single cell copy number data are generated by employing the maximum parsimony (MP) method. All phylogenies presented in the manuscript are the most parsimonious solution based on the MP algorithm. Moreover, when we generate the phylogeny, we employ the pratchet algorithm with 2000 iterations in combination with the TBR algorithm, making it unlikely to end up in a local optimum during the tree search iteration. The phangorn R package, in which the MP algorithm is implemented, includes information on how to obtain alternative phylogenies. In these cases, these are less parsimonious and also involves inclusion of additional back mutations or parallel mutations. The DEVOLUTION algorithm itself does, however, allow acquisition of alternative *nesting* solutions, if there are any, but these are also less parsimonious.

Revision: None, as we believe the issue has been addressed. Not also, that we have also already stated in Online Methods (p. 18) that the most parsimonious tree was chosen.

4. *I could not reproduce the results. I followed Microsoft Word document provided at Zenodo "How to build a tree.docx" (<https://zenodo.org/records/10727558>).*

From the same Zenodo archive I downloaded "Input Devolution.xlsx" file, stored it into my directory and followed all the instructions from the document, using R version 4.2.2 as suggested,

but when executing the command

```
datasegment <- splitdata(data,name=x)
```

I got the following error

```
"Error in `[<-` (*tmp`, s, 1, value = k) : subscript out of bounds"
```

Then I suspected that x should be set to some value but no guidance on what that value should be is provided. There are some comments that I believe are supposed to serve as a guidance: "Extracting the beginning and end position of each sample in the segment file. #Declare which tumor you want to analyze. Specified by the first column in your data set." but they never mention x and I could not follow them.

Response: We apologize for the confusion. The variable "x" is supposed to be the tumor name, as indicated by the first column of the segment file of interest. In this example, to run the code, enter the following:

```
Datasegment <- splitdata(data,name="Tumor1")
```

We refer to the github page for more information:

<https://github.com/NatalieKAndersson/DEVOLUTION>

Revisions: None as we believe the issue has been resolved above.

Reviewer #5 (Remarks to the Author): Arbitrating reviewer; expert in computational cancer genomics, cancer evolution, and phylogenetic analysis

1. *As reviewer #3 noted, some of the assumptions made when using DEVOLUTION, particularly in relation to the MCF calculation (e.g. described in the Methods section starting on l. 309), have indeed been shown to lead to erroneous reconstructions. In addition, there are tools that attempt to correct these problems, such as PhyloWGS (which may not be applicable here as it was developed for WGS data) and DeCiFer, which the reviewer also pointed out. Unfortunately, there seems to be only a benchmark against one other algorithm (MAGOS) in Andersson et al. and this is not based on predictions of ground truth from simulations, but applied to real tumor data, which prevents a definitive assessment of the performance and accuracy of DEVOLUTION.*

Since the main conclusions of the current paper rely on accurate phylogenetic reconstructions, I think it would be important to re-run the data through DeCiFer (which I think should be possible since DeCiFer allows integration of SNV and CNV data across multiple samples) to check the robustness of the inferred phylogenies. Alternatively, the authors should at least use MCF estimates from another algorithm (like DeCiFer) as input for their density-based clustering scheme, as they already mention as a possibility in Andersson et al.: "There are multiple dedicated tools that can infer MCFs from sequencing data such as the clustering algorithms PyClone, SciClone, and MAGOS, and recently also DeCiFer and for structural variants SVclone which could be used as an input for DEVOLUTION."

Response: We have now processed the samples with the DeCiFer (version 2.1.4) with Python 3.18 under GCC 11.3.0 (to be noted: the tool was built in Python 2.7 architecture and since then, has not been ported. We reached out to the maintainer to install a precompiled binary but eventually were able to use it with an updated kernel. DeCiFer requires the input information in a sample-wide consensus format where each variant is accompanied with corresponding allele specific copy number status across samples alongside estimated sample purity. Script for this can be found at (https://github.com/Subhayan18/DeCiFer-scripts-for-NB_Mets-NatComm24).

We have re-calculated MCF values for the SNVs with DeCiFer, and the results are presented in the novel Figure S8. We decided to use default settings in DeCiFer as the primary goal was to calculate clone sizes. The `cmm_CCF` estimates were used for further comparison with previously reported estimates

We believe DeCiFer to be a useful tool, but mainly for SNVs on CNA with a low number of alleles and for tumors containing many mutations that can avoid sample sample bias, since the most likely order of events when it comes to CNA and SNVs is difficult to elucidate when the measure points are few. Unfortunately, DeCiFer still fails to resolve a number of more complex copy number issues in our dataset containing a number of tetraploid and one pentaploid tumor. In our opinion using it less optimal than the heuristic approach we have described in Online Methods (p. 19-20):

-DeCiFer uses precomputed likelihood statistics of CNA changes across sample and the prebuilt statistics only include diploid and some elementary hyperdiploid tree states.

-DeCiFer uses density-based cluster assignment with variable epsilon boundary that can be defined with the n-sigma limit. The cluster assignments can be manipulated with widening the limit but with default 1-sigma boundary many variants were not assigned to a cluster (marked in orange in the new Fig S8 and summarized in panels marked V in that figure). Since changing the limit would seem highly subjective and not fulfil the task of benchmarking, we stayed with the default settings in the benchmarking analysis performed.

-DeCiFer infers copy number changes in a chronological manner. It chooses the ploidy based on an assumed order of the CNAs. We think this led to questionable ploidy choices in a few instances (see details below). In addition, copy number states varying across space (as in our dataset) can arise through two or more parallel chronological sequences, which DeCiFer seems unable to accommodate. Moreover, for higher-order, complex copy number changes (e.g. 1+1 → 3+2), DeCiFer needs likelihood estimates provided for corresponding samples. These can be estimated with a DeCiFer plug-in *generatetree* where the tree states can change based on the copy number change order. We did not implement this solution as we think the order

introduces as high degree of subjectivity. Furthermore, DeCiFer use case and the patient samples in our study are not compatible. Thus, we were unable to compare results of Patient 3, a pentaploid case due to state trees being unavailable in DeciFer for the copy number alterations in question.

Revision: We have now benchmarked our clone size calculations against DeCiFer as far as possible, considering some of the obstacles outlined above. In the novel Figure S8, we show VAFs for all variants and we have marked all variants present on CNAs (Panel I), the original MCFs calculated by us and how we sorted the variants into different clusters before calculating the final clone sizes. MCFs obtained by DeCiFer with variants not assigned to a cluster marked in orange are given (Panel III), a cleaned version of the original phylogenetic tree (Panel IV), and an overview of the number of SNVs detected originally, the number of variants not assigned to a cluster by DeCiFer, and the number of variants that did not receive an MCF value by DeCiFer since it was situated on a complex CNA background (summarized in Panel V).

The main purpose of Figure S8, however, is to show the striking similarities between the original clone sizes and the DeCiFer-generated (compare heatmaps II with heatmaps III). The individual MCFs are very similar, and the global patterns almost identical with a few exceptions seen for Patient 1 (Fig S8a) and Patient 5 (Fig S8d). For Patient 1, the DeCiFer-generated MCFs makes it impossible to generate a logical phylogenetic tree and DeCiFer-generated biologically unlikely values for Patient 5 (see legends to Figure S8 for details). It was very assuring to see how similar the MCF values became even when the variants were situated on CNAs, for example see variants marked by black borders for Patient4 (Fig S8c), Patient5 except for the variants discussed above (Fig S8d), Patient10 (Fig S8h), and Patient12 (Fig S8j).

Although it was not possible to generate MCF values for all variants with DeCiFer we hope that Figure S8 demonstrating both the VAF values and the result of the manual clustering with the original MCF values will convince the reviewers that the overall structure of the phylogenetic tree matches the appearance of the VAFs and that our heuristic approach is sound and clear, also providing a way to analyze cases with complex copy number profiles. Especially we would like to point out that the further application of DeCiFer to create trees from patients 1 and 5 would force us to make a high number of subjective choices specifically for these cases, while we could use our heuristic approach consistently across all cases.

2. I also agree that the stability of the trees should be discussed (it is discussed in Andersson et al.) and that the selection criteria for the trees shown should be clearly stated.

Response: See the point 3 from Reviewer 3.

REVIEWERS' COMMENTS

Reviewer #3 (Remarks to the Author):

The authors have addressed my main concerns and I have no additional comments.

Reviewer #5 (Remarks to the Author):

The authors have used an alternative algorithm (Decifer) to determine MCF values, which in turn serve as input for the reconstruction of the phylogenetic trees. This led to qualitatively similar results that support the heuristic procedure used to generate the original MCFs. The new Supplementary Figures 8 are an informative addition to the paper. It would also have been useful to see the results if a less restrictive distance measure had been used, which would have allowed Decifer to assign more variants to clusters. Although it is expected that this would not change anything qualitatively. In addition, it would be good if the procedure for estimating M and MCF were automated and validated against simulation data at some future time point.

Furthermore, I recommend that the authors clean up the formulas in the "Online methods" section with regard to the definitions of MSF and MCF: The equation in l.429 is just a rearrangement of the equation in l.419, but instead of MSF it says MCF and the definition of f_2 in l.435 is wrong. It might also be useful to describe in more detail how the information in the TAPS plots was used (currently primarily referred to by citing Rasmussen et al., 2011), as much of the conclusions are based on their interpretation.

Reviewer Comments and Response:

In the following text, reviewer comments are copied verbatim in italics, followed by our comments and revisions in plain text.

Reviewer #3 (Remarks to the Author)

The authors have addressed my main concerns and I have no additional comments.

Response: We are immensely happy to read this!

Reviewer #5 (Remarks to the Author):

The authors have used an alternative algorithm (Decifer) to determine MCF values, which in turn serve as input for the reconstruction of the phylogenetic trees. This led to qualitatively similar results that support the heuristic procedure used to generate the original MCFs. The new Supplementary Figures 8 are an informative addition to the paper. It would also have been useful to see the results if a less restrictive distance measure had been used, which would have allowed Decifer to assign more variants to clusters. Although it is expected that this would not change anything qualitatively. In addition, it would be good if the procedure for estimating M and MCF were automated and validated against simulation data at some future time point.

Response: We did in fact try several Decifer settings but realized that changing these settings required a series of highly subjective decisions that would severely impact data reproducibility. We agree that it would be wonderful to have a future automated M and MCF assignment to formalize our manual heuristic and have incorporated this idea in the revised Discussion.

Revision: We have now added in Discussion (p. 18, l. 414-417) the following sentence: "An additional improvement for future studies would be a script-based, simulation-validated process that replaces the manual assignment of allele compositions and clone size in the complex subclone scenarios often present in clinical NB samples. "

Furthermore, I recommend that the authors clean up the formulas in the "Online methods" section with regard to the definitions of MSF and MCF: The equation in l.429 is just a rearrangement of the equation in l.419, but instead of MSF it says MCF and the definition of f2 in l.435 is wrong. It might also be useful to describe in more detail how the information in the TAPS plots was used (currently primarily referred to by citing Rasmussen et al., 2011), as much of the conclusions are based on their interpretation.

Response: We thank the reviewer for finding these mistakes and information gaps.

Revision: The Method section is revised according to the instructions; equation steps have been removed and the errors corrected. We have also added a section detailing the structure of the TAPS plots and how these plots were used to retrieve information regarding subclonality and allelic composition for identified aberrations. This information is added to the Methods paragraph "Clonal deconvolution based on CNAs detected by SNP array in clinical samples".